# Mechanically derived short-range order and its impact on the multi-principal-element alloys

Jae Bok Seol [1,9,10] ✉, Won-Seok Ko[2,9], Seok Su Sohn [3], Min Young Na[4], Hye Jung Chang[4], Yoon-Uk Heo [5], Jung Gi Kim[1], Hyokyung Sung[6], Zhiming Li [7], Elena Pereloma [8] & Hyoung Seop Kim [5,10] ✉

Chemical short-range order in disordered solid solutions often emerges with specific heat treatments. Unlike thermally activated ordering, mechanically derived short-range order (MSRO) in a multi-principal-element $Fe_{40}Mn_{40}Cr_{10}Co_{10}$ (at%) alloy originates from tensile deformation at 77 K, and its degree/extent can be tailored by adjusting the loading rates under quasi-static conditions. The mechanical response and multi-length-scale characterisation pointed to the minor contribution of MSRO formation to yield strength, mechanical twinning, and deformation-induced displacive transformation. Scanning and high-resolution transmission electron microscopy and the anlaysis of electron diffraction patterns revealed the microstructural features responsible for MSRO and the dependence of the ordering degree/extent on the applied strain rates. Here, we show that underpinned by molecular dynamics, MSRO in the alloys with low stacking-fault energies forms when loaded at 77 K, and these systems that offer different perspectives on the process of strain-induced ordering transition are driven by crystalline lattice defects (dislocations and stacking faults).

Future industrial development requires sustainable metallic alloys with an outstanding synergy between strength and ductility[1–3], particularly under extreme conditions. Multi-principal-element alloys (MPEAs), the so-called medium- or high-entropy alloys (MEAs or HEAs, respectively), have received considerable interest. At the core of the interest lies their potential for achieving the requirements, for example, near-infinite alloy compositions at elevated temperatures[4–13]. Unlike traditional metallic alloys, MPEAs are concentrated solid solutions comprising mixtures of elements in which the atoms of principal elements tend to be distributed randomly. However, several studies have indicated that the large-enthalpy interactions among the constituent species can drive a few MPEA systems towards inherent chemical ordering at nanometric scales, as suggested by the results obtained using the computational simulations[14,15] and experimental methods[16–21]. When compositionally homogeneous structures have specific low-coordination-number clusters, the preferential local ordering of principal elements generally dominates over the spatial order of a few nearest-neighbour spacings, i.e. short-range order (SRO), often called chemical SRO (CSRO)[22,23].

[1]Department of Materials Engineering and Convergence Technology, Center for K-metal & Microscopy, Gyeongsang National University, Jinju 52828, South Korea. [2]Department of Materials Science and Engineering, Inha University, Incheon 22212, South Korea. [3]Department of Material Science and Engineering, Korea University, Seoul 02841, South Korea. [4]Advanced Analysis and Data Center, Korea Institute of Science and Technology, Seoul 25451, South Korea. [5]Graduate Institute of Ferrous and Energy Materials Technology, Pohang University of Science and Technology, Pohang 37673, South Korea. [6]School of Materials Science and Engineering, Kookmin University, Seoul 02707, South Korea. [7]Department of Materials Science, Central South University, Changsha 410083, China. [8]School of Mechanical, Materials, Mechatronic and Biomedical Engineering & UOW Electron Microscopy Centre, University of Wollongong, Wollongong, NSW 2522, Australia. [9]These authors contributed equally: Jae Bok Seol, Won-Seok Ko. [10]These authors jointly supervised this work: Jae Bok Seol, Hyoung Seop Kim. ✉e-mail: jb.seol@gnu.ac.kr; hskim@postech.ac.kr

The formation of diffusion-mediated CSRO domains in fully disordered alloys belongs to the category of thermally activated isostructural disorder-to-order transition at short ranges. Such ordering influences the dynamics of structural defects (primarily, dislocations) as well as the macroscopic mechanical, thermal, and functional properties of concentrated solid solutions including most MPEAs[24–30]. It has been widely conjectured that localised planar slip and leading dislocations would destroy the pre-existing CSRO domains in a face-centred-cubic (fcc) phase upon loading, which corresponds to the so-called glide plane softening[31,32]. This effect drives the non-random system towards ideal random, referred to as the 'deformation-derived CSRO-to-disorder transition'. Meanwhile, increasing the degree and spatial extent of the CSROs can strengthen fcc solid solutions by activating the planarity of dislocation slip and by modifying the strain hardening capacity[16,23,27].

Extensive efforts to monitor the ordering transition in MPEAs have provided insights into the thermally induced CSROs via simulations[14,15,26] and X-ray absorption[30]. However, X-ray-based measurements provide relatively low analytical capabilities, restricting the experimental observation of CSROs in the individual grains of the materials, as they are averaged over a comparatively large volume of material[16]. The transmission electron microscopy (TEM) and atom probe tomography techniques were considerably enhanced so as to record the extremely small CSROs within individual fcc grains for metallic alloys[16–21]. For instance, TEM dark-field images, formed with CSRO-induced additional diffuse scattering in reciprocal-space selected-area electron diffraction patterns (EDPs) with specific zone axes, enable quantitative measurements of the CSRO degree/extent in NiCrCo and VNiCo MEAs[16–18] and $Fe_{50}Mn_{30}Cr_{10}Co_{10}$ (at%) HEA[19] with the similar atomic sizes of constituent species. This approach can also be used to study binary Ti–6Al concentrated solution[33] and high-Mn (25 at%) steel sample[34]. Overall, CSRO sizes ranged from ~0.7 to ~2.0 nm for a given fcc-structured alloy composition[34,35]. Recently, medium-range orders (1–5 nm) in a dual-phase $Al_{9.5}CrCoNi$ (at%) MEA system have been temporarily defined as the next-level order structure beyond CSRO[21].

In contrast to the superlattice reflections attributable to long-range-ordered nanoprecipitates, elastic diffuse scattering between normal fcc Bragg reflections in TEM-EDPs is ascribed to structural deviations (e.g. vacancies, dislocations, stacking faults, and SROs) from a periodic crystalline lattice[36,37]. Among these defects, the detection of extra diffuse discs at ½{311} locations in the [$\bar{1}$12] TEM-EDPs is often used to evidence the presence of randomly distributed CSROs within the individual grains in most fcc-structured alloys, including MPEAs and high-Mn steel. Thus, stacking faults (SFs) in most fcc structures often cause clear streaking along the {111} directions between common fcc Bragg reflections in the EDPs under the [110] zone axis. Recently, ref. 16 made an intriguing discovery that the TEM dark-field images accompanied by 'SF-caused streaking', provide irrefutable evidence of CSROs in a furnace-cooled NiCrCo MEA. They further revealed that increasing the amount of CSRO increased the stacking-fault energy (SFE) and hardness. The experimental observations of the work were consistent with those of the previous modelling-based studies, where tuning the local chemical order at the nanoscale altered the SFEs and mechanical properties of some MPEAs[14,26]. However, there are some points of disagreement about the effect of CSROs. Other TEM-based studies suggested that SFEs of MPEAs do not influence the degree of chemical ordering[24,25]. Therefore, the following fundamental questions remain: (i) whether SFs are phenomenologically intertwined with the clustered ordering; (ii) whether ordering can affect mechanical twinning and deformation-induced martensitic transformation[12]; and (iii) the mechanism and reason for the 'deformation-derived disorder-to-SRO transition', suggested based on experimental observations[35] and theoretical calculations[38].

In this work, we address these crucial questions using the aforementioned TEM suite of tools for an interstitially hardened $Fe_{40}Mn_{40}Cr_{10}Co_{10}$ (at%) HEA with boron (B) ingress. The reasons for choosing this non-equiatomic HEA composition are explained in the Methods section. With the aim of naturally extending the fundamental knowledge on strain-induced disorder–order transition space relevant to the mechanical behaviour of MPEAs, we primarily characterised both SROs, i.e. diffusion-mediated CSRO in the initial microstructure of B-doped $Fe_{40}Mn_{40}Cr_{10}Co_{10}$ HEA (before tensile deformation) and the newly formed SRO during tensile deformation at 77 K. Hereafter, such strain-induced ordering is denoted as mechanically derived SRO (MSRO) for convenience. We then elucidate the evolution of the microstructure with different loading rates under quasistatic conditions across multiple lengths scales to investigate the vital effect of MSRO on the mechanical twins and deformation-induced martensitic transformation at 77 K. To phenomenologically underpin the TEM results obtained by studying the fcc single-grains, we further conducted atomistic Monte Carlo (MC) simulations, molecular dynamics (MD) simulations, and MD-based virtual diffraction analyses in the 77 K-deformed structure of an fcc $Fe_{40}Mn_{40}Cr_{10}Co_{10}$ single-crystal system and its bi-crystal counterpart. In this context, the experimental results and computational predictions provide insights into the microstructural features attributable to MSRO in the non-equiatomic FeMnCrCo HEA system and other fcc non-equiatomic NiCrCo MEAs (for universality), This is particularly true for the alloys with low SFEs and those subjected to mechanically tensile loading at 77 K.

## Results
### Modelling of CSRO

Figure 1a–c shows the spatial distribution snapshots of four chemical species, i.e. Fe, Mn, Cr and Co in the non-equiatomic $Fe_{40}Mn_{40}Cr_{10}Co_{10}$ system, predicted by MC simulations at elevated temperatures of 1373, 873 and 750 K, respectively. The details of the simulations are elaborated in the 'Methods' section. Although the near-random spatial distribution of each element is observed at 1373 K (Fig. 1a), CSRO formation in the single-crystal system is potentially favourable at 873 and 750 K (Fig. 1b and c, respectively). The associated radial distribution function profiles revealed three features. First, by lowering the temperature, Fe–Co, Mn–Cr, Mn–Co, Fe–Mn, and Fe–Cr, all exhibited a strong tendency to be neighbours, promoting the variability of atomic packing sequence in the first neighbouring shell (Fig. 1d–f). This holds for a general CSRO definition, i.e. 'preference for unlike pairs and avoidance of like pairs'[39]. The results from our MC simulations for the $Fe_{40}Mn_{40}Cr_{10}Co_{10}$ HEA system were consistent with the type of order, reported in a recently reported TEM-based works[19,39], which highlights the Fe–Mn/Co/Cr preference in the first neighbouring shell in an fcc $Fe_{50}Mn_{30}Cr_{10}Co_{10}$ system. Second, of all unlike pairs, Fe–Co exhibited the strongest tendency to be neighbours. Third, a primarily strong tendency toward 'preference for Cr–Cr like pairs' was generated by lowering the temperatures. Besides these results, a large negative mixing enthalpy ($\Delta H_{mix}$) of B with Cr (−61 kJ/mol) and Co (−42 kJ/mol)[40] with respect to a possible interstitial site in grain-interior can drive the B-doped system towards chemical ordering. The solution enthalpy gives an indication that the formation of Cr-enriched CSROs in the B-doped MPEA material subjected to ageing at ~773 K is energetically more preferred, as compared to that in the B-free case subjected to the same thermal treatment. Hence, we anticipate that B ingress can promote the CSROs into the quaternary non-equiatomic MPEA. Based on the findings, we intentionally used the following thermal treatments: homogenisation, hot rolling, cold rolling, and recrystallisation annealing at 1073 K for 60 min, followed by water quenching. Afterward, the alloy was subjected to ageing at ~773 K for 360 min, followed by furnace cooling to 298 K (see the Methods section for details). Additionally, to validate the efficiency of the thermal treatments applied for CSRO generation, we conducted differential thermal analysis (DTA) of the B-doped and B-free reference samples subjected to the ageing and furnace cooling. Results of the DTA profiles are

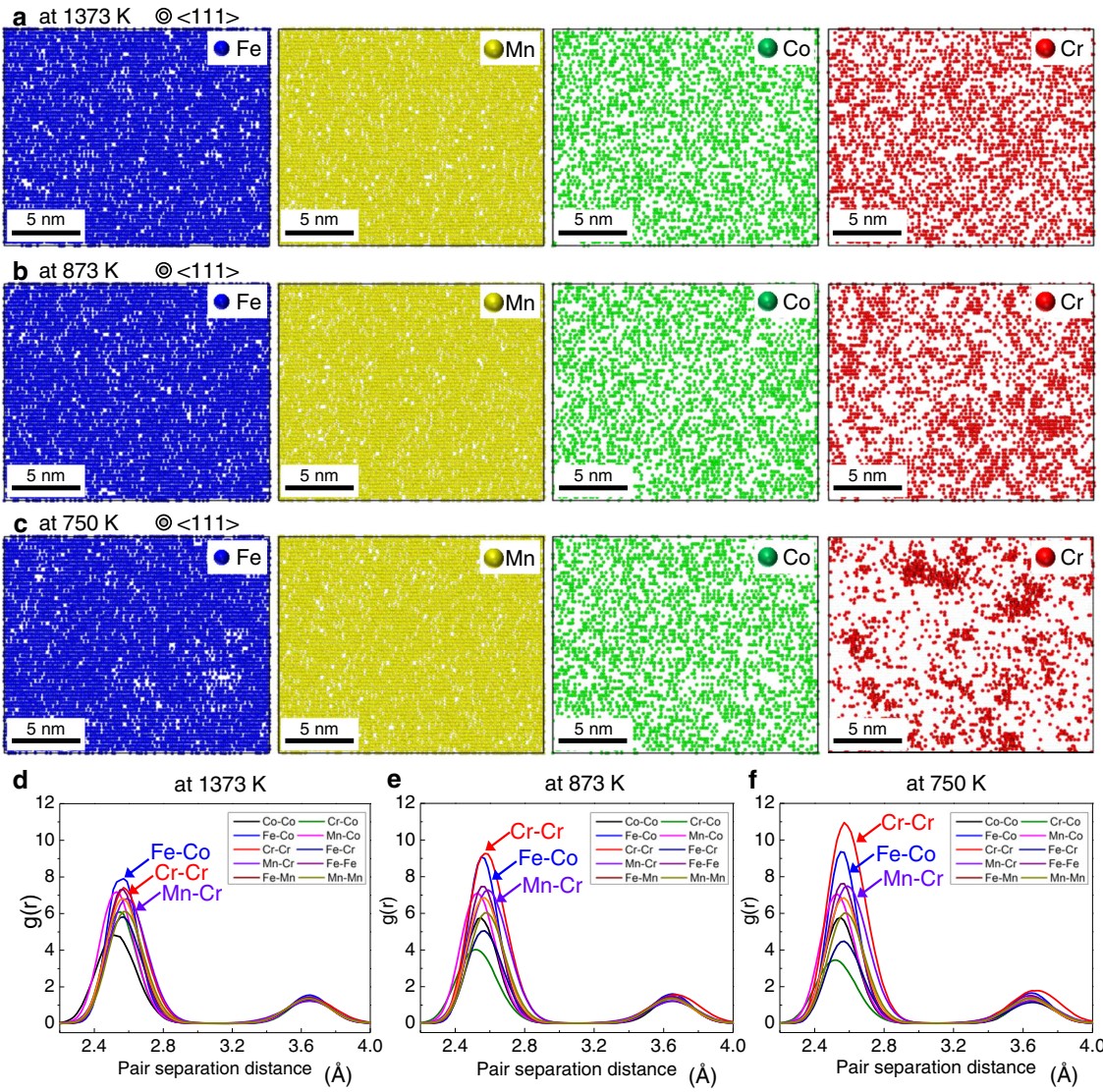

**Fig. 1 | Atomistic MC simulations for CSRO domains in fcc-structured Fe₄₀Mn₄₀Co₁₀Cr₁₀ (at%) HEA system. a** Near-random distribution of four principal elements obtained after the MC simulations at 1373 K. A cell with 216000 atoms (15.7 nm × 13.6 nm × 12.8 nm) was used for the simulation. **b, c** Non-random distribution of principal elements showing thermally activated Cr-enriched like pairs and Fe−Co unlike pairs at 873 and 750 K, respectively, based on the MC simulations.

The captured slices of configurations with a thickness of ~2 nm are shown for clear visualisation. **d, e, f** Corresponding radial distribution function g(r) profiles from the MC simulations, revealing that when the temperature is lowered, both Cr–Cr like pairs (red curves) and Fe–Co unlike pairs (blue curves) are strongly favoured at the first nearest-neighbour distances for producing CSROs in the HEA before straining.

provided in Supplementary Fig. 1, indicating that the thermal treatments employed in this work are feasible for generating CSRO (the details are described in Supplementary Notes 1).

## Mechanical response

The B-doped HEA containing CSROs was tensile-tested at 77 K under quasistatic tensile conditions of moderate and low strain rates ($\dot{\varepsilon}$) of $1.4 \times 10^{-4}$ and $2.8 \times 10^{-5}\,s^{-1}$; the corresponding samples were labelled HEA-M and HEA-L. The $\dot{\varepsilon}$ values and loading temperature were intentionally chosen to minimise deformation-induced heating during tensile tests (see Supplementary Notes 2 for details). The engineering stress−strain curves of HEA-M and HEA-L are shown in Fig. 2. These are compared with those of the same alloy system (in the aged state after recrystallisation) subjected to tensile deformation at 77 K with a higher $\dot{\varepsilon}$ of $1.1 \times 10^{-3}\,s^{-1}$ (denoted as HEA-H). The application of a higher $\dot{\varepsilon}$ increased the maximum tensile strength ($\sigma_{UTS}$) and total elongation ($\varepsilon_{total}$): $\sigma_{UTS}$ and $\varepsilon_{total}$ increased from 1202 MPa and 12% for HEA-L to 1385 MPa and 22% for HEA-H, respectively. The simultaneous increase

in $\sigma_{UTS}$ and $\varepsilon_{total}$ caused by a higher $\dot{\varepsilon}$ is predictable for most fcc-based structural alloys, as a higher deformation rate under quasistatic conditions results in a more uniform distribution of dislocations under conditions of the same amount of strain[41]. Typically, a higher $\dot{\varepsilon}$ increases or decreases the 0.2% offset yield strength ($\sigma_{YS}$) significantly for a given alloy composition, indicating a strong dependence of $\sigma_{YS}$ on the applied $\dot{\varepsilon}$ value[41]. Additionally, nanoscale CSROs and long-range order (LRO) precipitates in HEAs would have enhanced the $\dot{\varepsilon}$ sensitivity of some alloys[42]. However, the current HEA did not follow these typical trends: i.e. the $\dot{\varepsilon}$ dependence of $\sigma_{YS}$ was negligible. From this unusual $\dot{\varepsilon}$ sensitivity, unlike the generally known impact of either CSRO or LRO precipitate that acts as a hardening source[27–29,43], it seems that the sample under study may require additional stress or dragging force to promote the glide of mobile dislocations in the plastic strain regime[29,43].

## Underlying deformation mechanism

All tensile-tested samples were examined by backscattered electron (BSE) imaging and electron backscattered diffraction (EBSD) analyses

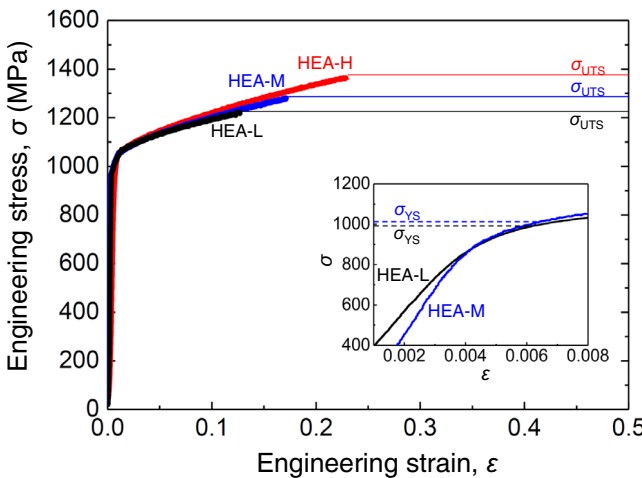

**Fig. 2 | Tensile stress–strain curves of the B-doped Fe₄₀Mn₄₀Co₁₀Cr₁₀ (at%) HEA at 77 K for different strain rates.** The material was finally aged at 773 K for 360 min followed by furnace cooling to room temperature. This fcc single-phase sample was tensile-tested at 77 K, with strain rate $\acute{\varepsilon} = 2.8 \times 10^{-5}\,\mathrm{s^{-1}}$ (black curve; low rate; labelled as HEA-L) and $\acute{\varepsilon} = 1.1 \times 10^{-4}\,\mathrm{s^{-1}}$ (blue curve; moderate rate; labelled as HEA-M), under uniaxial quasistatic conditions. In addition, the tensile curve of the same alloy composition subjected to recrystallising at 1073 K for 60 min followed by water quenching to room temperature is included, along with the uniaxial quasistatic 77 K-straining curve at $\acute{\varepsilon} = 1.2 \times 10^{-3}$ (red curve; high rate; labelled as HEA-H) for comparison[35]. All samples exhibited similar grain sizes ranging from ~3.5–5.1 μm. $\sigma_{YS}$ represents the 0.2% offset yield strength; $\varepsilon_f$, the total elongation; and $\sigma_{UTS}$, the maximum tensile strength. Inset: the magnified curves.

in the regions corresponding to the uniform plastic strain regime and final fracture. The BSE images of the samples deformed at different $\acute{\varepsilon}$ values (HEA-L, HEA-M, and HEA-H) showed profuse planar slip lines and slip bands during tensile testing (Supplementary Fig. 2a–c, respectively). The typical EBSD images of HEA-M showed the post-tensile-testing cell structures resulting from mechanical twins and martensitic transformation from fcc to hexagonal-close-packed (hcp: yellow) phases (Fig. 3a). The EBSD analysis revealed that the applied $\acute{\varepsilon}$ had a negligible influence on the mean area fraction and transformation kinetics of the hcp phase in the fracture and uniform elongation regimes (Supplementary Fig. 2d–f). The bright-field TEM image from one single fcc grain in the HEA-M, viewed from the [011] beam direction, showed the formation of many <110>{111}-type planar slip bands (Fig. 3b). Selected-area EDPs were used to distinguish the mechanical twins from slip bands in the microstructure (Supplementary Fig. 3). The tuneable $\acute{\varepsilon}$ had a negligible effect on the mean thickness of mechanical twins including nano-twins (Fig. 3c), determined from dark-field TEM, high-resolution (HR)TEM, and scanning (S)TEM images recorded the [011] beam direction. Instead, an increase in $\acute{\varepsilon}$ resulted in both more slip bands and band refinement (Supplementary Fig. 3b); the mean widths of slip bands in HEA-H, HEA-M, and HEA-L samples were measured to be 53–114, 88–190, and 210–382 nm, respectively. This experimental observation refers to relatively stronger slip planarity with a higher $\acute{\varepsilon}$.

Next, we experimentally investigated the reason for a higher-$\acute{\varepsilon}$-based refining of the slip bands. It has been established that one slip band formed in the early stages of straining comprises numerous slip planes with high-density dislocations, and the creation of further dislocations in these bands is often prohibited[44,45]. As complex microstructures are present inside the slip bands, TEM observation of slip band substructures that form during deformation is experimentally challenging. To overcome this difficulty, we tried to examine the slip bands in the HEA-M sample using an appropriate zone axis. The choice of the [Ī12] zone axis yielded two advantages. First, the substructure of the bands could be unambiguously observed in the TEM images along

the [Ī12] zone axis compared to that for the [011] one (Supplementary Fig. 4). Second, SRO-generated additional diffuse scattering around normal fcc Bragg reflections was unequivocally detected in {311} directions or at ½{311} locations (halfway between the transmission spot (000) and the {311} spots) in the [Ī12] EDP from the respective fcc grain (see the green arrows in the right corner of Fig. 3d). Using an appropriate zone axis, we found that active slip system was **a/2**<011>{111} (a is the lattice parameter) rather than **a/2**<100>{001}. These preferred slips are observed in most fcc alloys because the shear stress exerted on the slip plane must be the smallest for slip. We showed that the SRO-introduced extra scattering in the EDPs intensified owing to the existence of slip bands in the deformation structure, even in the same TEM specimen and in the same fcc grain. Figure 3e is a representative example of this, obtained by fast Fourier transform (FFT) patterns based on HRTEM. The concrete or spot-like extra diffuse scattering caused by SRO is unambiguously observed at the ½{311} locations in the [Ī12] FFTs acquired from the regions of slip band (red boxes), whereas the diffracted electron signals of the diffuse scattering in the FFT from the band-free regions (grey boxes) are rather limited. This result provides an indication that the SRO is directly correlated with the strain-induced slip bands. Assuming that there was no atomic diffusion of principal species in the HEA during quasistatic deformation at 77 K, it is pertinent to attribute the additional diffuse scattering to the emergence of strain-induced SRO rather than to the initial CSRO prior to the deformation. Again, MSROs are distributed along the slip band. In contrast with the MSROs the distribution of CSROs is highly irregular throughout the individual grain. This was demonstrated by computational studies[14,15,22,26] and experimental images[16–21,33,34].

We further distinguished the MSRO-derived spot-like scattering from common TEM characterisation artefacts associated with the introduction of double diffraction and Moiré fringes from fcc-twinned structures[46,47]. The HRTEM–FFT pattern obtained from the overlapped zone of slip bands and mechanical twins shows replicated extra spots owing to double diffraction (outlined blue boxes in Fig. 3e). The Moiré fringes were thus shown along the <112>-type twin in the high-angle annular dark-field (HAADF) and HRTEM images (Supplementary Fig. 5). The diffracted spots are also introduced when the surface of the thin TEM foils gets oxidised[48]. To differentiate between the MSRO-derived spot-like scattering and the surface oxidation-caused spots, we further performed in-situ TEM heating experiments of the HEA samples for accelerating the oxidation process (see the Methods for details). In the STEM and HRTEM images acquired after the in-situ experiments, heating-introduced surface oxidation (presumably, complex and layered oxide scale including thermodynamically stable $Cr_2O_3$)[48] onto the surface of the TEM foils provokes plentiful extra diffracted scattering in the [Ī12] and [Ī10] STEM-FFTs (Supplementary Fig. 6). Moreover, the locations of the diffracted spots attributable to the deliberate oxidation process can be unequivocally distinguished from the locations of the MSRO-derived spot-like scattering. Hence, the origin of the spot-like scattering shown in the HRTEM-FFT-based profiles (Fig. 3e) can be attributed to the formation of MSRO in the microstructure (particularly, inside slip bands) and not to the surface oxide scale of TEM foils. Eventually, the slip-band refinement with respect to a higher $\acute{\varepsilon}$ can be explained in terms of the MSROs that emerge inside the slip bands.

## Sources of MSRO

Either displacive transformation from fcc to martensite or ordering transition promotes continuous nanometre-scale displacements and resultant crystal-lattice distortion[15,18,19,25,29,49,50]. With these considerations, we conducted STEM analysis on the HEA-M sample along the [011] and [Ī12] zone axes at the same analysed regions. When viewed from the [011] beam direction, only normal fcc diffraction spots were observed in the STEM-FFT pattern. The perfect dislocations with the

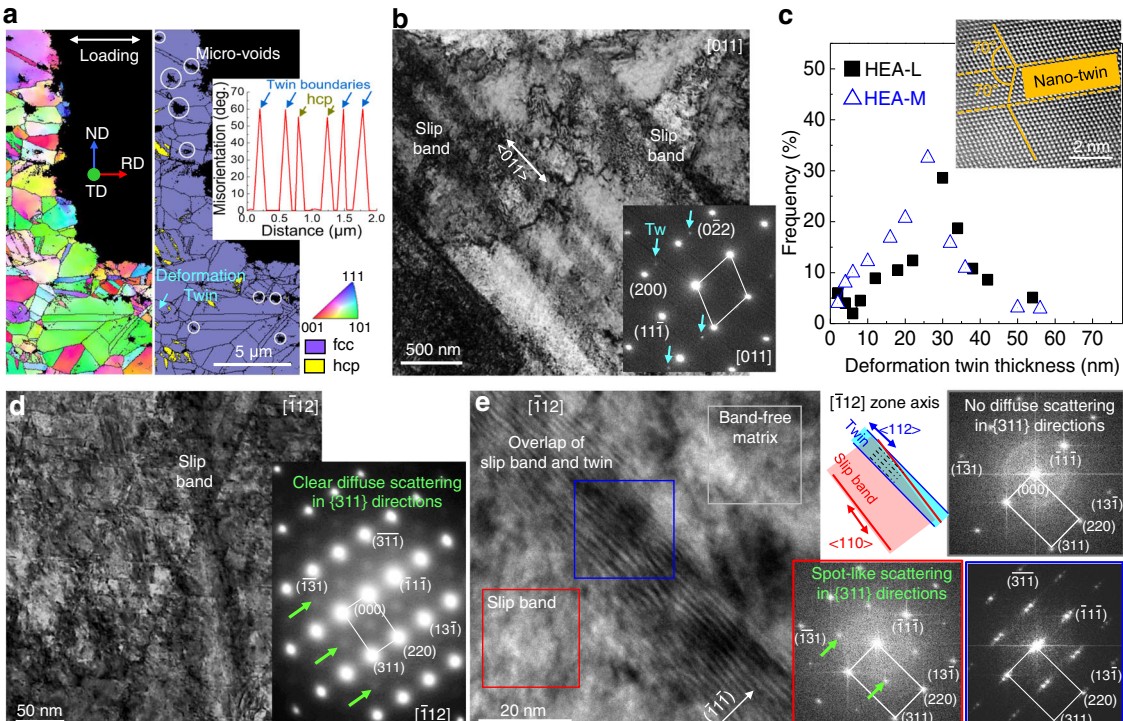

**Fig. 3 | Microstructure of the tensile-tested samples. a** Typical EBSD orientation (left column) and phase identification (middle column) maps, where the tensile loading direction was normal to the plane of view, obtained from the fracture surfaces of HEA-L. Micro-voids were formed at the internal grain and along grain boundaries. Right column top panel: sketch of the sample orientation showing the geometry of normal direction (ND), transverse direction (TD), rolling direction (RD), and the tensile loading direction. fcc: face-centred cubic; hcp: hexagonal closed pack. Middle panel: Representative misorientation angle profile of mechanical twinning boundaries and hcp boundaries. **b** Conventional bright-field TEM image of HEA-L, projected along the [011] zone axis, showing plentiful slip bands and mechanical twins. In the corresponding EDP, extra spots caused by mechanical twins are indicated by 'Tw'. **c** Lamellar twin thickness distribution with respect to the applied strain rates, determined from the TEM images for HEA-M and HEA-L. Inset: a representative STEM image of nano-twin with a mirror-symmetry at an angle of ~70° with respect to the twinning plane. **d** TEM image along [$\bar{1}$12] zone axis for HEA-M. Clear extra diffuse discs at the ½{311} locations in the corresponding EDP (right panel) are indicated by green arrows. **e** HRTEM image (left panel) of slip band passing with mechanical twins in **d** and corresponding FFT patterns (right panel) from specific regions of the slip-band-free matrix (grey box), slip band (red box), and mechanical twins (blue box), based on the [$\bar{1}$12] zone axis. The considerable overlap of slip bands and twins (schematic in the top right corner) resulted in double diffraction in the FFT pattern and Moiré-fringed lattices in the TEM image.

Burgers vector of **b** = <011> were observed with a high density of ~$1.8 \times 10^{10}$/m² (Supplementary Fig. 7). When imaged from the [$\bar{1}$12] beam direction, STEM-annular bright field and its corresponding enlarged dark-field images showed that the glide of mobile partial dislocations parallel to the <011> directions passed through the initially formed slip bands along the <001> directions (Fig. 4a). Additionally, our STEM imaging identified structural features attributable to the diffuse scattering (Fig. 4b–g). No diffuse scattering signals were found in the STEM-FFT pattern acquired from the band-free zone (blue boxes in Fig. 4b and c). In this area, no SFs were thus observed. However, STEM-FFT patterns for the severely lattice-distorted zones in the regions of slip bands (red boxes in Fig. 4b and d–g) showed the MSRO-derived diffuse scattering at the ½{311} locations. One MSRO domain is outlined in the inverse FFT image by superimposing the ordering signals with normal fcc lattice fringes (Fig. 4e). The corresponding magnified lattice fringes reveal the sublattice structure of the domain core, wherein the {311} atomic planes alternate with fcc and MSRO preferentially. This implies that, like the case of CSRO in equiatomic MEAs[17,18], the measured interplanar spacing ($d_{MSRO}$) of MSRO {311} planes (green dotted lines) in the non-equiatomic HEA system doubles the interplanar spacing ($d_{fcc}$) of the {311}$_{fcc}$ planes (white lines). The doubled $d_{fcc}$ accounts for the generation of extra diffuse scattering at the ½{311} locations in the electron reciprocal space. Consistent with the previous TEM-based studies (where considered that CSROs and medium-range-order domains form an interchangeable preferential ordering in a NiCrCo MEA)[17,18,21], our

observations suggest that a period of {311} interplanar spacing in MSRO doubles that of the fcc lattice for the current alloy. This can explain the detection of the diffuse scattering introduced by MSRO at the ½{311} locations in all [$\bar{1}$12] TEM-EDP and HRTEM- or STEM-FFT results. Figure 4f shows severe lattice distortions at the interface between MSRO and fcc matrix. In the corresponding enlarged lattice fringes produced with {111} planes, both the mechanical SFs in {111} directions and immobile Franck dislocations with **b** = 1/3 <111> (marked by 'T' symbols) were observed (Fig. 4g). Both were caused by strain-induced lattice distortion. Particularly, the applied plastic strains led to the renewed 0.1-nm-scale atomic packing mismatch with ~10° bond angle of fcc {111} planes. This distortion in the {111} directions is consistent with the type of bonding preferences suggested in the previous HRTEM result of a NiCrCo MEA[16], in which the alternative contrast was caused by lattice distortion in CSROs along the <111> directions. Hence, we attribute the formation of MSROs to the combination of mechanical SFs and edge dislocations inside slip bands, which are driven by atomic packing displacement during straining. This observation also suggests that most MSROs appear with a rather regular distribution along the slip bands in the individual grain for the deformed structure. This suggests an intrinsic difference between the initial CSROs and the MSROs, even though both generate additional diffuse scattering either along the {311} directions or at ½{311} locations in the TEM-EDPs. In other words, this again gives an indication that the formation mechanism of displacive MSRO differs from that of thermally activated CSRO.

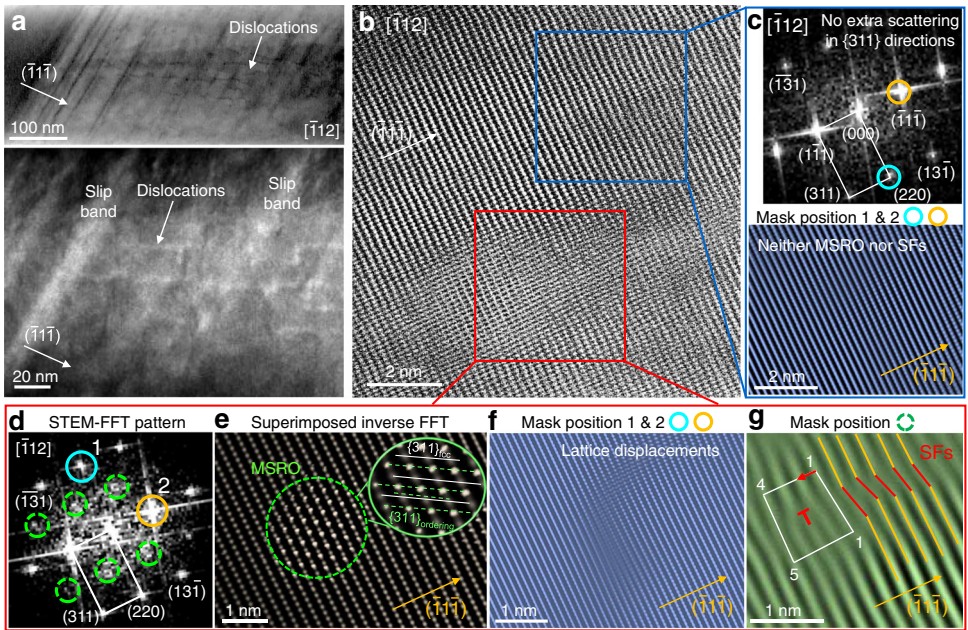

**Fig. 4 | STEM analysis of the origin of mechanically driven SRO in the deformation structure. a** STEM image (top) of HEA-M along the [$\bar{1}12$] zone axis and the corresponding enlarged annular dark-field image (bottom), showing the glide of mobile dislocations across the pre-existing single slip lines and slip bands that are inclined to the <011> directions on the {111} planes. **b** STEM image of one slip band. **c** [$\bar{1}12$] STEM-FFT pattern (top column) and corresponding inverse FFT lattice fringes (bottom column), obtained from the undistorted area (blue box) in **b**, showing neither extra diffuse scattering nor SFs. **d** [$\bar{1}12$] STEM-FFT pattern obtained from the distorted zone of the local lattice in the red box in **b** showing the mechanically driven SRO (MSRO)-derived extra discs at the ½{311} locations (green circles). **e** Inverse FFT image produced by superimposing one single MSRO domain (green dotted circle) with fcc lattice fringes. Corresponding magnified lattice fringes showed the sublattice structure of the domain core, where the {311} atomic planes alternate with normal fcc (white lines) and MSRO (green dotted lines). **f, g** Inverse FFT images from different fcc normal spots, obtained from the interface between MSRO and fcc phase: examples revealing that the structural features attributable to the MSRO-derived diffuse discs at the ½{311} locations in the FFT patterns are associated directly with the synergy of SFs (red lines) on {111} planes and edge dislocations with Burgers vector $b = 1/3 < 111 >$ (marked by 'T').

## Strain-rate dependence of MSRO

We characterised variations in the MSRO density and size as a function of applied $\acute{\varepsilon}$. Selected-area EDPs were performed on the CSRO-containing samples (B-doped HEA and its B-free reference) before and after straining. To support our TEM-EDPs, MD simulation-based virtual EDPs (briefly, MD-EDPs) were carried out (see the Methods section for details). To this end, we assessed how such MD-EDPs could describe our TEM-EDPs for common $L1_2$-structured long-range-order (LRO) precipitates consistently (see Supplementary Fig. 8 and additionally Supplementary Notes 3).

The B-free $Fe_{40}Mn_{40}Cr_{10}Co_{10}$ reference alloy before straining was chemically ordered at 750 K, predicted by MC simulations (especially, Cr–Cr pairs in Fig. 1). However, for the system containing CSROs, our MD-EDPs under the [$\bar{1}12$] zone axis revealed weak or faint diffuse scattering at the ½{311} locations (Supplementary Fig. 9). This indicates that the degree/extent of CSROs in the MC simulations for the reference structure may not yield irrefutable diffuse scattering in the MD-EDPs.

Compared to the B-free system, B-doped $Fe_{40}Mn_{40}Cr_{10}Co_{10}$ HEA prior to straining was expected to have more and larger CSROs owing to a relatively stronger driving force for CSRO formation in the DTA thermal profiles (Supplementary Fig. 1) and a strong negative solution enthalpy by B ingress. However, we experimentally detected weak diffuse scattering at the ½{311} locations in the [$\bar{1}12$] TEM-EDP for the HEA (Fig. 5a). Because our TEM samples include the extremely small CSROs in the real microstructure, a large inelastic background attributed to thermally induced diffuse scattering and plasmon scattering would lower the TEM spatial resolution[37]. This might be the primary cause for the weak diffuse scattering signals from the B-doped sample before straining.

After deformation at a low $\acute{\varepsilon}$, the B-doped sample (HEA-L) showed ordering-induced diffuse discs at the ½{311} locations in the [$\bar{1}12$] TEM-

EDPs (Fig. 5b). For consistency, all [$\bar{1}12$] EDPs were taken from regions of slip bands in individual fcc grains for HEA-L, HEA-M, and HEA-H. On increasing $\acute{\varepsilon}$, the extra diffuse discs gradually become focused at the same ½{311} locations in the TEM-EDPs for HEA-M (middle panel) and spot-like scattering was observed for HEA-H (right panel). A broad-intensity maximum was confirmed by a hump in the electron diffraction intensity profiles[18]. The hump in the TEM-EDPs for the B-doped HEA before straining was even observed in the profiles (gold curve in Fig. 5c). This observation confirms that although faint signals were observed in the TEM-EDP, CSROs were present in the microstructure. The variation in the diffuse scattering intensity as a function of the applied $\acute{\varepsilon}$ can be determined by estimating the ratios of the average peak intensity of the diffuse scattering to the normal fcc disorder reflections. The resulting ratios were 0.38 and 0.12 for HEA-M and HEA-L, respectively. The high intensity of the diffuse scattering owing to the deformation at higher $\acute{\varepsilon}$ could be correlated with the increased structural factor $F_{hkl}$, where $h$, $k$, and $l$ denote the Miller indices[47,51].

At this point, we differentiate between the MSRO-derived spot-like scattering in TEM-EDPs or HRTEM-FFTs and the typical LRO-caused superlattice reflections. Even in the TEM-EDPs along the [$\bar{1}12$] zone axis, the locations (½{311}) of the spot-like scattering for the current HEA (Fig. 5b) were distinct from those ({201} and {110}) of LRO-introduced reflections for other fcc structures (Supplementary Fig. 8). These LRO-caused spots are also found in another fcc-based FeNiCr-CoCuAl$_{0.5}$ HEA[52], where $L1_2$ precipitates generate extra superlattice reflections at the {201} and {110} locations in the [$\bar{1}12$] EDPs. Hence, deciphering where diffracted spots or diffused discs are located in the EDPs, for a given fcc structure, provides a feasible route to demarcate the boundary between nanosized LRO precipitation and high-degree MSRO. Apart from this approach, the transition from SROs to LRO structures during quasistatic tensile deformation at 77 K would be

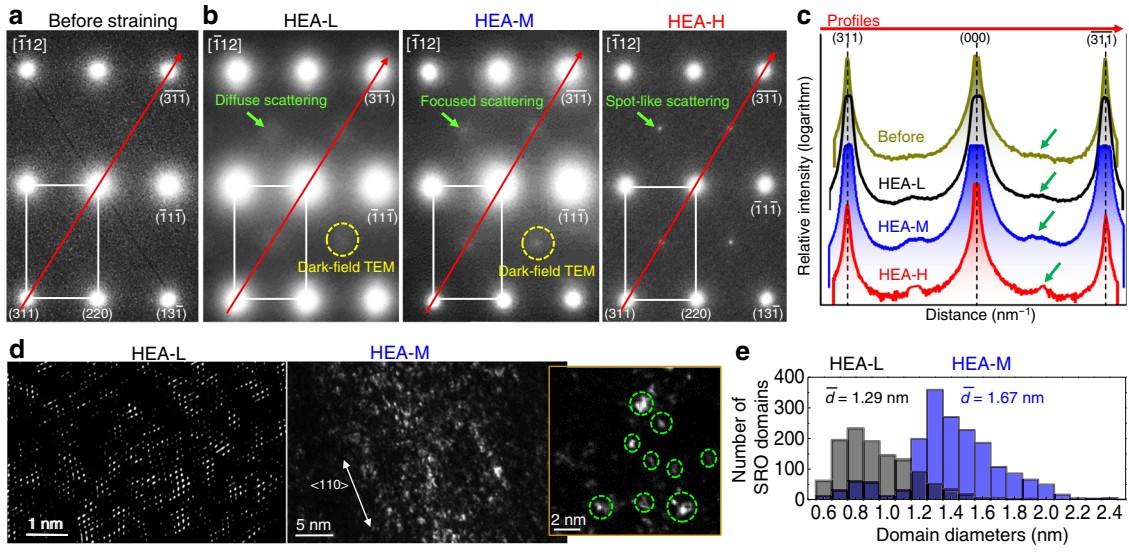

**Fig. 5 | TEM diffraction patterns, intensity of MSRO-derived extra diffuse discs, and the associated STEM images for different strain rates.** Selected-area [$\bar{1}12$] TEM-EDPs taken from individual HEA grains: **a** before and **b** after tensile straining at different strain rates. These EDPs were acquired from the region of slip bands in each deformed specimen. Green arrows indicate periodic diffuse scattering along the {311} directions caused by MSRO. **c** Logarithmically scaled diffraction intensity along the red lines in **a**, **b**: that is, the diagonal lines across normal fcc {311} spots in the EDPs from HEA-L (black) and HEA-M (blue). The hump owing to MSRO-generated diffuse scattering in **b** is outlined by green arrows in the profiles. The measured diffraction intensity for the same alloy composition subjected to 77 K-tensile straining at $\dot{\varepsilon} = 10^{-3}$ (red; HEA-H) is also included. **d** Inverse STEM-FFT (left panel) and annular dark-field (right panels) images formed with the MSRO-introduced extra diffuse discs in the FFT or diffraction patterns (yellow dashed circles) in **b**. The bright contrast in the real microstructure shows the MSRO domains (green dashed circles) in the slip bands that form during straining. The magnified region (yellow box) of the STEM image revealed no shearing of the MSROs in the slip bands inclined to the <110> directions. e Histograms of identified MSROs diameters for HEA-L (grey) and HEA-M (blue) samples, showing the mean value $\bar{d}$.

energetically unfavourable. Earlier studies suggested that clustered SROs can self-aggregate into larger medium-range order or even nanosized LRO precipitates, only when thermal treatments are applied at elevated temperatures, i.e. above ~770 K[22,53]. In other words, the SRO-to-LRO transition is accompanied by a change in the solution enthalpy from strongly negative to strongly positive values. However, tensile testing at 77 K, conducted in this study, is relatively deficient in the thermal energy required for LRO formation in fcc-structured HEA. Therefore, our analysis of the transition event (from diffuse scattering to spot-shaped discs in the [$\bar{1}12$] TEM-EDPs) reveals that either extremely high degree/extent of MSROs or large medium-range orders were realised upon loading at high $\dot{\varepsilon}$. Sudden LRO formation during tensile deformation at 77 K was not indicated. This assertion is thus supported by our MD simulations, described in next section.

Figure 5d shows inverse STEM-FFT and TEM-HAADF images, formed with the MSRO-derived scattering in the [$\bar{1}12$] EDPs for HEA-L (left panel) and HEA-M (right panel), respectively. Compared with HEA-L showing the ½{311} diffuse scattering in the EDP, more and larger MSROs were distributed for HEA-M exhibiting the spot-shaped scattering in the EDP. Approximating the clustered MSROs as spheres[16–19,21,34,35] helps characterise the ordering size distribution (Fig. 5e). Deformation at higher $\dot{\varepsilon}$ increased the intensity of diffuse scattering, leading to increased amounts of MSROs with larger sizes. Such MSRO is likely to be stabilised upon loading at higher $\dot{\varepsilon}$, i.e. strain-induced SRO can be stable owing to the competition between bond length and bond angle forces[38]. Direct observations in support of this hypothesis follow.

To support the TEM results showing the $\dot{\varepsilon}$ dependence of MSRO (Fig. 5), we performed MD-EDP analysis with the associated deformation simulations. Figure 6a shows the [$\bar{1}12$] MD-EDPs, predicted from the non-equiatomic FeMnCrCo system before and after straining at 77 K. While faint diffuse scattering at the ½{311} locations in the MD-EDPs was shown for the system before straining (left column in Fig. 6a, and see additional MD-EDPs along other zone axes in Supplementary

Fig. 10a), deformation at a higher $\dot{\varepsilon}$ results in an increase in the intensity of MSRO-introduced extra discs at the same locations (middle column in Fig. 6a). For higher $\dot{\varepsilon}$, spot-shaped ½{311} discs were predicted at the same locations (right column in Fig. 6a). This trend is consistent with the TEM-EDPs for the interstitial-doped version (Fig. 5a–c). If ordering-introduced ½{311} diffuse discs in the [$\bar{1}12$] EDPs (via either TEM or MD simulations) are detected, definite SF-induced streaking along the {111} directions in [110] MD-EDPs (via either TEM or MD simulations) is observed (Supplementary Fig. 10b).

Figure 6b shows the distributions of mechanical SFs and dislocations in the non-equiatomic FeMnCrCo cell structures strained with different $\dot{\varepsilon}$ values at 77 K, as predicted by the MD simulations. Deformation at a higher $\dot{\varepsilon}$ gives rise to more dislocations (or stronger slip planarity) and more SFs (Fig. 6c). This demonstrates that the increased intensity of MSRO-introduced scattering in reciprocal-space MD-EDPs owing to higher $\dot{\varepsilon}$ drives the B-doped system towards stronger slip planarity and more SFs. Moreover, there were no LRO precipitates in all of the cell structures. This confirms that spot-like ½{311} scattering in the MD-EDPs is directly correlated with SFs and dislocations but not with LRO structure. The strained cell structure showed both the hcp-like structure (~19 and ~4 vol% at the low and moderate $\dot{\varepsilon}$, respectively) and SFs (~21 and ~26 vol% at the low and moderate $\dot{\varepsilon}$, respectively). As the initial leading Shockley partials tend to form SFs with trailing partials, the fraction of SF regions increases upon quasistatic deformation at a higher $\dot{\varepsilon}$ value.

## Forerunner of diffuse scattering

The $\dot{\varepsilon}$ dependence of the MSRO in the non-equiatomic FeMnCrCo HEA was similarly tracked in a non-equiatomic $Ni_{60}Cr_{20}Co_{20}$ (at%) MEA via MD-EDPs and MD simulations. For the non-equiatomic NiCrCo MEA with a predicted SFE of 33.7 mJ/m² (Supplementary Table 1), increasing the $\dot{\varepsilon}$ gives rise to a higher intensity of MSRO-derived ½{311} diffuse scattering in the [$\bar{1}12$] MD-EDPs (Fig. 6d). This trend, i.e. increasing the $\dot{\varepsilon}$ values drives the diffuse discs towards spot-like scattering at the same

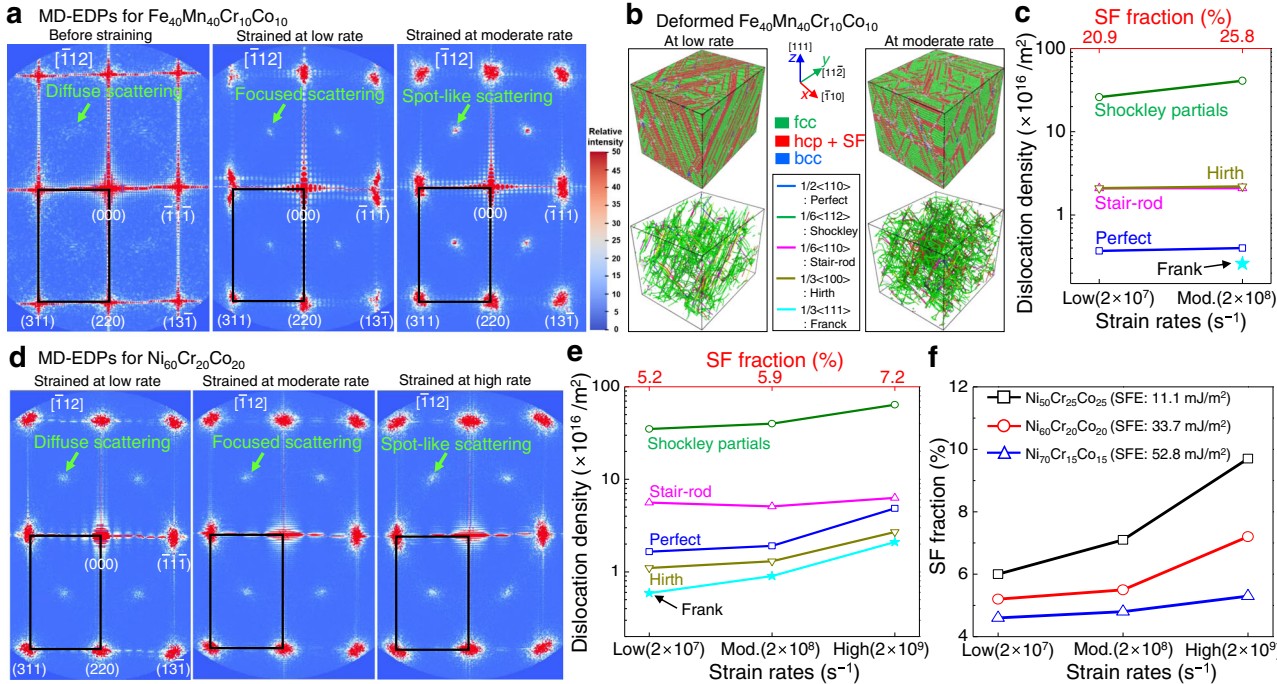

**Fig. 6 | MD simulations for the origin of MSRO-generated extra scattering in non-equiatomic Fe$_{40}$Mn$_{40}$Cr$_{10}$Co$_{10}$ (at%) HEA and non-equiatomic NiCrCo MEAs upon loading at 77 K. a** MD-EDPs for the interstitial-free FeMnCrCo structure before and after straining. The [$\bar{1}$12] diffraction patterns for the structure before straining show MSRO-derived faint diffuse diffraction scattering at the ½{311} locations, while the relative intensity of MSRO-derived diffuse scattering increased with a higher strain rate. The colour scale (bottom right) defines the relative intensity of diffraction spots in arbitrary units (blue: low intensity; red: high intensity). **b** Corresponding cell structures strained at 77 K under conditions of different strain rates, showing the distribution of deformation-induced SFs (stacking faults: top panel) and dislocations with different Burgers vectors (bottom panel). The dislocation distribution reveals that strong slip planarity was achieved at higher strain rates. There were no LRO precipitates in all cell structures. This affirms that spot-like scattering at the ½{311} locations in the MD-EDPs directly correlates with SFs and dislocations, but not with LRO structure. The strained cell

structures at slow and moderate $\dot{\varepsilon}$ were included in this work for comparison, as the highest $\dot{\varepsilon}$ (2 × 10$^9$ s$^{-1}$) caused severe amorphisation of the system and restricted the observation of the $\dot{\varepsilon}$ effect under equivalent conditions. **c** Volume fraction of the SFs and dislocations as a function of the strain rate for the non-equiatomic HEA. **d** MD-EDPs for the strained Ni$_{60}$Cr$_{20}$Co$_{20}$ (at%) MEA, confirming that the relative intensity of MSRO-derived diffuse scattering increased with increasing strain rate. In the MD-EDP for the MEA system before straining, there were no diffuse discs. The strained cell structures, showing the distribution of mechanical SFs and dislocations, are shown in Supplementary Fig. 11. **e** Volume fraction of the mechanical SFs and dislocations as a function of $\dot{\varepsilon}$ for the MEA. **f** Increased SF volume fraction with a higher $\dot{\varepsilon}$ for other NiCrCo-based non-equiatomic MEAs with different SFEs. The corresponding strained cell structures and MD-EDPs are shown in Supplementary Fig. 12. The SFEs were determined from the current molecular static simulations at 0 K (see Supplementary Table 1).

locations, is concurrent with the results (Figs. 5b and 6a) for the B-doped non-equiatomic FeMnCrCo. Figure 6e shows the detailed variation in SF fraction and dislocation density in the non-equiatomic NiCrCo cell structures with respect to the applied strain $\dot{\varepsilon}$, as revealed by MD simulations. This confirms that applying high $\dot{\varepsilon}$ increases the MSRO-derived scattering intensity, SF fraction, and dislocation density in the non-equiatomic NiCrCo MEA. The corresponding cell structures showing the evolution of SFs and dislocations are provided in Supplementary Fig. 11. This scenario is also found in other Ni$_{50}$Cr$_{25}$Co$_{25}$ and Ni$_{70}$Cr$_{15}$Co$_{15}$ (all in at%) MEAs (with the predicted SFEs of 11.1 and 52.8 mJ/m$^2$, respectively). Upon loading with different $\dot{\varepsilon}$, the variation of SF fraction, dislocation distribution, and MSRO-derived electron scattering for the non-equiatomic NiCrCo systems is shown in Fig. 6f and Supplementary Fig. 12. Similar to the MD-EDPs for the strained non-equiatomic FeMnCrCo system (Supplementary Fig. 10b), streaking in the {111} directions between normal fcc spots in the [$\bar{1}$10] MD-EDPs was unambiguously visible, while extra ½{311} diffuse discs in the [$\bar{1}$12] MD-EDPs were visible for the strained Ni$_{50}$Cr$_{25}$Co$_{25}$ MEA (Supplementary Fig. 12b).

## Discussion
We found that thermally induced or diffusion-mediated CSRO forms in the Fe$_{40}$Mn$_{40}$Cr$_{10}$Co$_{10}$ furnace-cooled HEA system with a low SFE (prior to deformation at 77 K). It coexists with the strain-induced or

diffusionless MSRO in the deformation structure at 77 K. The MSRO is primarily formed inside slip bands, and its origin can be attributed by strain-induced crystalline defects (e.g. SFs and dislocations). The MSRO formation may obey the lattice defect model of SRO and not the statistical Warren–Cowley SRO parameter that suits specifically for a binary AB alloy[22,54]. Based on the previous report[55], we also defined the MSRO described in this study as ordering at short ranges. It can form as a result of the diffusionless redistribution of principal elements under mechanically loading conditions, particularly at cryogenic temperatures in the absence of external heating effects. This is a prime cause for the differences in the CSROs. For a detailed understanding of MSRO, we examined the atomistic structures with and without MSRO. Even though the atomic distribution obtained from MC simulations of a structure with MSRO was similar to that obtained from MC simulations of a structure without MSRO, results from MD simulations revealed the differences in the cell structures and MD-EDPs associated with the two different systems (Supplementary Fig. 15). Based on the results obtained using the MD simulation and TEM techniques (Fig. 4), we have presented details on the possible MSRO atomic structure motif and the corresponding unit cell provided in Supplementary Fig. 16 and Supplementary Discussion 4. However, to determine which MSRO motif exists in the real-deformation structure, further TEM-energy dispersive spectroscopy characterisation[17,18] is required.

CSRO in a few MPEAs can be described in terms of 'preference for unlike species and avoidance of like pairs'[17,18,21,39]. Our MC simulations revealed that lowering the temperature of the $Fe_{40}Mn_{40}Cr_{10}Co_{10}$ structure led to a relatively strong tendency for Fe–Co, Mn–Cr, Mn–Co, Fe–Mn, and Fe–Cr, promoting the variability of atomic packing. This trend is also demonstrated experimentally in fcc-structured $Fe_{50}Mn_{30}Cr_{10}Co_{10}$ (at%) HEA[19,39]. Interestingly, a strong tendency of 'preference for Cr–Cr like pairs' is observed for the $Fe_{40}Mn_{40}Cr_{10}Co_{10}$ HEA system, where the atomic sizes of the four species (Fe, Mn, Cr, and Co) are similar[39]. Cr–Cr pairs were previously demonstrated for equiatomic NiCrCo MEA system. The results were arrived by conducting MC simulations[14] and analysing experimental results[16,18]. This indicates that although the alloy compositions differ, the formation of Cr–Cr-related CSROs would be favourable for the MPEAs including a high amount of Cr. This is potentially attributed to the complex mechanisms of CSRO formation[19]. A further in-depth energy computation method or SRO cluster-plus-glue-atom model[22,28], suited for MPEAs (not for binary AB alloys), should be used to theoretically clarify this point in the future. Rather than reiterating the recent efforts on the CSRO or medium-range order, the results reported herein fills the missing gaps existing in the field of strain-induced short-range ordering in MPEAs.

The diffuse scattering at the ½{311} locations in the [$\bar{1}$12]-indexed EDPs came into focus at specific regions (strain-induced slip bands) in the deformation fcc structure. More SROs in the 77 K-deformed HEA were formed than those in the undeformed sample with the same alloy composition, and they were distributed along the slip bands. Applying higher $\dot{\varepsilon}$ under quasistatic conditions increased the intensity of MSRO-derived ½{311}$_{fcc}$ diffuse scattering from faint signals to spot-like ones, which in turn yielded more and larger SROs[21] in the slip bands. We again highlight that the spot-like scattering monitored in the TEM-EDPs, HRTEM- or STEM-FFTs, and MD-EDPs is explicitly derived by a high degree of MSRO, but not by LRO precipitates. The precursor of MSRO-derived diffuse discs and spot-like scattering for the strained MPEAs was directly correlated with strain-induced crystal imperfections (e.g. dislocations and mechanical SFs). Furthermore, from the viewpoint of mechanical response, the non-equiatomic quaternary alloy system exhibited no $\sigma_{YS}$ dependence on the $\dot{\varepsilon}$ values. According to the findings, we herein regarded the localised SROs as mechanically derived SROs (MSROs) rather than diffusion-mediated CSROs that would be destroyed by planar dislocations during deformation.

Our STEM images further revealed the microstructural features responsible for such ordering in the non-equiatomic quaternary HEA, i.e. MSROs in fcc grains were attributed to strain-induced structural deviations (specifically, dislocations and SFs) from a periodic lattice. These results were validated by MD simulations and relevant MD-EDPs. However, three concerns can be raised: (i) a vast difference in the applied $\dot{\varepsilon}$ values between the bulk samples (in the range of $10^{-5}$–$10^{-3}$ s$^{-1}$) and the simulations ($10^7$–$10^9$ s$^{-1}$); (ii) the simulated $Fe_{40}Mn_{40}Cr_{10}Co_{10}$ system had a single-crystalline structure, while experimental B-doped $Fe_{40}Mn_{40}Cr_{10}Co_{10}$ sample was polycrystalline; (iii) the accuracy of the potential used in the present simulation. The first issue is explained in Supplementary Discussion 1, while additional MD simulations are conducted using a bi-crystal model with grain boundary for the second issue (see Supplementary Fig. 13 and Supplementary Discussion 2). The relevant discussion has been presented in Supplementary Discussion 3 and Supplementary Fig. 14.

We showed that for the $Fe_{40}Mn_{40}Cr_{10}Co_{10}$ HEA system strained at 77 K, the detection of the SF-induced streaking along the {111} directions in [110] MD-EDPs corresponded to the existence of MSRO-derived ½{311} diffuse discs in [$\bar{1}$12] MD-EDPs. This is even true for other 77 K-strained non-equiatomic NiCrCo structures. An earlier TEM study demonstrated that highly intensified streaking along the {111} directions in [110] EDPs was directly correlated with high-degree CSROs in a NiCrCo MEA[16]. Based on our findings and those reported

previously, it is highly plausible that an increase in the diffuse-disc intensity due to a high $\dot{\varepsilon}$ reflects a high degree of MSROs, more SFs, and more dislocations in the deformation structures for the non-equiatomic MPEAs with lower SFEs. To sum up, displacive MSRO belongs to isostructural disorder–order transition that occurs in slip bands during 77 K-tensile deformation (specifically, in the plastic strain regime). We assume that MSRO might be defined to be the degree of strain-induced local deviation from the average local-scale ordering in terms of either chemical occupation or structural defects, particularly for the MPEAs with low SFEs and upon loading at 77 K. The process of low-temperature deformation can result in the lowering of the SFEs of fcc-based alloys, leading to an increase in the activities of dislocations and stacking faults. Unfortunately, a kinetics investigation of MSRO evolution has still not been performed, and it is unclear whether MSRO does involve the elemental enrichment of specific constituent species[55] in the case of the alloy under study. Further investigation to figure out which elements are associated with the MSROs and how MSROs are mechanically stabilised under conditions of high $\dot{\varepsilon}$, for example, at low-temperature conditions and different levels of applied plastic strain, would be interesting.

The increased degree/extent of MSROs with a higher $\dot{\varepsilon}$ had a propensity to refine the slip bands in the current HEA, enhancing the $\sigma_{UTS}$ and $\varepsilon_{total}$ simultaneously. This implies that MSROs can act as additional stress or dragging force that allows for mobile dislocations to percolate a glide plane after the onset of yielding, particularly in the plastic strain regime[29,43]. If the observed slip bands are completely developed in the large-strain regime, the associated MSROs will cause glide plane hardening[44,45], based on our observations of non-shearable MSROs (Figs. 4 and 5d, and Supplementary Fig. 5a). However, in-depth characterisation of non-shearable MSROs is still an open issue. A higher $\dot{\varepsilon}$-induced hardening, after the onset of yielding, arises from a high degree/extent of MSRO, and this is potentially promoted by the significantly high lattice distortion effects[28,34,35,56] in the large-strain regime. Although twinning-induced plasticity (TWIP) or transformation-induced plasticity (TRIP) effects can be active in the large-strain regime and arrest the strain localisation[2,40], the current multilength-scale characterisation revealed that both were independent of the applied $\dot{\varepsilon}$ values. The TWIP or TRIP effects are less affected by MSRO formation, as the ordering emerges with the slip bands.

As the evolution of individual slip bands is suppressed by dislocation-source exhaustion in fcc-based metals[57], the denser bands at higher $\dot{\varepsilon}$ are associated with the high density of MSRO. More and larger MSROs are likely to generate a large amount of localised heat inside the slip bands, as the SRO phenomenon (either CSRO or MSRO) is a heat generation process. This MSRO-induced heat will stimulate the slip-band refinement, i.e. additional slip bands can be generated near pre-existing ones, thereby lowering severe strain localisation in a single slip band[45]. Accordingly, additional slip bands can be refined with further straining at a given $\dot{\varepsilon}$. This is phenomenologically analogous to the so-called dynamic slip-band refinement[44]. We anticipate that this slip-band refinement effect is caused by a higher $\dot{\varepsilon}$ and governs the $\varepsilon_{total}$ dependence on the $\dot{\varepsilon}$. Thus, the current study can provide important insights into the fundamental, practical, and mechanistic understanding of many concentrated solid solutions that are designed to take advantage of the disorder–order transition during deformation at low temperatures.

## Methods
### Materials and sample preparation
The interstitially hardened non-equiatomic $Fe_{40}Mn_{40}Cr_{10}Co_{10}$ (at%) single-phase HEA with 30 ppm boron was cast in a vacuum induction furnace (MC100V, Indutherm, Walzbachtal-Wossingen, Germany) using pure metals (purity higher than 99.9 wt%) and a high-purity FeB lump. An interstitial-free version containing the same principal

elements was also cast for comparison. Casting was carried out under an Ar protective atmosphere. The raw ingots of the interstitial-doped and -free HEA specimens, with dimensions of 100 mm × 35 mm × 8 mm, were homogenised at 1478 K for 360 min, pickled in a 20% HCl solution, milled to a thickness of 7 mm, and then water-quenched to room temperature. The isotropic texture had a grain size of ~100 μm for the interstitial-doped case, as determined by EBSD. Grain refinement was achieved through cold rolling to a thickness reduction ratio of 75%, changing the thickness from 7.0 to 1.5 mm. The cold-rolled sheets were annealed at 1073 K for 60 min followed by water quenching to 298 K[35] to achieve full recrystallisation. The uniform texture for all specimens contained fcc single phase, with a grain size range of ~3.5–5.1 μm for the interstitial-doped case as determined by EBSD (Supplementary Fig. 3). The current MC simulations predicted that the interstitial-free HEA showed a transition from a disordered structure to a chemically ordered structure upon the reduction of temperature from 1378 to 750 K (Fig. 1). The recrystallised interstitial-free sample was further aged at ~773 K for 360 min followed by furnace cooling to 298 K to achieve thermally activated CSRO domains. According to this result, we expected that the interstitial-doped alloy would have a high potential to possess CSRO domains owing to a strong negative $\Delta H_{mix}$ of B with Cr (−61 kJ/mol) and Co (−42 kJ/mol)[40].

As demonstrated by microstructure characterisation using X-ray diffraction (XRD) and EBSD, the interstitially hardened HEA had a fcc single-phase structure in all grain-refined states, i.e. recrystallised and aged states[35]. The aged samples were then sliced and thinned via mechanical polishing. The nominal chemical compositions of the aged samples were measured by inductively coupled Ar plasma atomic-emission spectrometry, obtaining the chemical composition of $Fe_{41}Mn_{37}Cr_{11}Co_{11}$ (at%) for the HEA. Upon loading at room temperature, the interstitial HEA showed a single fcc phase, including mechanical twins, but no phase transformation, as deduced from XRD and EBSD[35], as did the interstitial-free sample[58]. Therefore, irrespective of B doping, the current non-equiatomic HEA system was treated as a single-phase TWIP-assisted HEA at 298 K, as reported in the literature[58].

The following rationale explains the reason for selecting $Fe_{40}Mn_{40}Cr_{10}Co_{10}$ HEA with 30 ppm of B in this work: (i) upon tensile loading at 77 K, both deformation twinning and martensitic phase transformation from fcc to hcp phases were observed[35]; and (ii) the alloy system is likely to form MSROs upon loading at an initial strain rate of $10^{-3}$ s$^{-1}$ at 77 K[35]. Hence, the alloy compositions are indispensable for elucidating the mechanism of MSRO formation and its impact on the HEA. Although the concentrated solid solutions investigated in this study are the same as those in our previous work[35], different thermal treatments of low-temperature ageing and furnace cooling were employed here to infuse the CSROs into the alloy compositions before deformation.

## Tensile testing

Tensile specimens with a gauge length of 6.4 mm, a diameter of 2.5 mm, and a thickness of 1.5 mm were cut from a quarter through-thickness position of the fully recrystallised HEA sheets along the rolling direction. Uniaxial tensile testing was performed at 77 K using an Instron machine (Model 1361, Instron Corp., Canton, USA). A 100-kN load cell operating at strain rates of $1.1 \times 10^{-4}$ and $2.8 \times 10^{-5}$ s$^{-1}$ was used under quasistatic conditions. The $\dot{\varepsilon}$ values and loading temperature were intentionally selected to minimise the deformation-induced heating effect on the deformation mechanism and dislocation glide mode under quasistatic tensile conditions. Data were averaged from three or four tests in all specimens deformed at each strain rate. The mechanical properties of all specimens were characterised by yield strength, ultimate tensile strength, and strain to fracture (total elongation). The yield strength was determined from the 0.2% offset stress.

## Backscattered electron microscopy experiments

Multiple characterisation techniques were used to examine the microstructures of the tensile-tested samples through backscattered electron (BSE) and EBSD analyses. The fracture surfaces of the tensile-tested samples were electro-plated with Ni to protect the surface prior to polishing. Then, the samples for BSE and EBSD measurements were mechanically polished (down to ~1 μm), followed by electrochemical polishing using a Lectropol 5 instrument. For electrochemical polishing, a solution of 95% acetic acid and 5% perchloric acid was used. Electrochemical polishing was conducted with 30 V and 50 mA at room temperature for 1.5 min. EBSD measurements were obtained for the fractured samples. An EBSD observation was acquired through field-emission SEM (Quanta 3d FEG, FEI Company, USA) equipped with an OPTIMUS$^{TM}$ TKD detector head (eFlash$^{HR}$, ARGUS$^{TM}$ electron detection system, Bruker, Germany). EBSD scans were obtained with an acceleration voltage, beam current, working distance, and specimen tilting of 20 kV, 22 nA, ~7 mm, and 70°, respectively. Over 350,000 data points were obtained at a rate of 186.2 frames per second. The EBSD data were interpreted using the TSL OIM data collection software (Analysis 7). The minimum confidence index was 0.12.

## Conventional TEM, high-resolution TEM, and scanning TEM imaging

Electron-transparent samples for TEM were prepared by mechanical polishing of thin foils down to a thickness of less than ~80 μm, then punched to discs 3 mm in diameter. Subsequently, electropolishing was carried out using a twin-jet machine with a mixed solution of 92 vol % alcohol and 2 vol% perchloric acid at −25 °C and 28 mA, and finally thinned by an Ar$^+$ beam in a Precision Ion Polishing II (GATAN) system. The bright-field low-magnified TEM images, HRTEM images, and HAADF images were obtained using a JEOL 2010F microscope with an aberration corrector (by a parallel incident electron beam with an accelerating voltage of 200 kV). The probe size was set to 0.1 nm with a convergence semi-angle of 22.5 mrad. Inner and outer collection angles of the HAADF imaging detector (Cs-corrected JEOL 2100F) were 50 mrad and 150–180 mrad, respectively.

Furthermore, for bend-contour-free dislocation imaging, annular bright-field (ABF) and annular dark-field (ADF) techniques in STEM mode were applied to analyse the deformed structures[59,60]. The STEM-ABF and -ADF images were obtained using a cold field-emission gun JEOL NEO-ARM microscope with a spherical aberration correction system (operated at 200 kV and 35 pA) and a cold field-emission gun JEOL−ARM200F with double Cs corrections (operated at 200 kV). The STEM images were obtained at a camera length of 20 or 40 cm, and the corresponding collection angle ranged from 35 to 93 mrad or 17.5 to 46.7 mrad, respectively.

As the convergent electron beams in the STEM mode illuminate the thin TEM samples, the dislocation contrast is often enhanced and the elastic strain contrast in the image background can be reduced. The STEM specimens were tilted using a double-tilt sample holder to satisfy either the systematic excitation condition or the two-beam excitation condition that is appropriate for observing the dislocations. Two different zone axes indexed as [011] and [1̄12] were selected in this study. The former was used to distinguish the planar slip lines or their bands from the deformation-induced mechanical twins or deformation-induced phase transformation. The latter was used for direct observation of SRO-derived electron diffuse scattering as well as for better visualisation of the slip-band substructure (see Supplementary Fig. 4). To observe the crystalline lattice of the fcc phase in the deformed samples, FFTs were conducted to filter out noise from the images, yielding FFT-filtered lattice fringes of the target area. More than three selected-area EDPs and TEM images under two or three different zone axes were obtained from more than three fcc grains for each strain-rate condition to ensure experimental reproducibility.

## MSRO size distribution

An alternative approach to determine the resolution limit is to directly measure the semi-angle of the aperture used in the literature[16,17], i.e. $r_{Airy} \approx \frac{1.2\lambda}{D} = \frac{1.2\lambda}{\alpha} = 1.2d\prime$, where $r_{Airy}$ is the radius of the Airy disc used in this study; $\lambda$ is the electron wavelength (0.02507 Å for 200 kV TEM); $f$ is the focal length of the objective lens (-2.3 mm for the JEOL 2100F and JEOL NEO-ARM); $D$ is the diameter of the objective aperture (-20 μm in this study); $\alpha$ is the measured semi-angle of the aperture; and $d\prime$ is the measured size of the aperture in the reciprocal space. Here, $r_{Airy}$ was -3.5 Å, which was below the sizes of the MSROs in all specimens. This method has a resolution limit of 2.5 Å, reaffirming that our HRTEM and STEM resolution was sufficiently high to measure the MSRO density. A -5 eV-energy slit was used to eliminate the contrast from inelastic scattering. A Gatan US1000CCD camera was used to obtain the EDPs and dark-field images.

We obtained the TEM-EDPs, taken from the regions of SBs for all deformed structures. For MSRO recognition, we then detected periodic diffuse scattering along the {311} directions in the diffraction patterns along the $[\bar{1}12]$ zone axis. Subsequently, one of the MSRO-derived diffuse or focused scattering points was selected for inverse FFT and STEM-HAADF imaging of MSROs. Based on a previously reported procedure[16–18,35], we determined the identified MSRO size distribution by applying a Gaussian template fitting algorithm along with optimised identification parameters to observe MSROs by dark-field imaging. The identified MSROs inside the slip bands can be regarded as having a spherical shape, which is consistent with the circular nature of the diffuse scattering that indicates a lack of preferred orientation[16,18,29].

## In situ heating TEM for intentionally accelerating the surface oxidisation of the TEM foils

We performed in situ TEM heating experiments with the HEA samples under under study to intentionally accelerate the surface oxidation of thin TEM foils. The in situ TEM heating experiments were performed using a Gatan heating holder equipped with water connection on a conventional TEM (JEOL JEM-2010 without any correctors) equipment at 500 °C for 20 min. Afterwards, the surface oxidation of the TEM lamellar was examined using HRTEM and STEM modes in an atomic-resolution TEM (JEOL NEO-ARM) to accurately distinguish the SRO-derived diffuse scattering from the oxidation-induced reflections in the EDPs.

## Measurement of exothermic reactions

To determine the thermal reaction for both the B-doped HEA and its undoped version subjected to ageing at 773 K followed by furnace cooling, DTA tests were conducted. The sample volume for the DTA was -3 × 3 × 1 mm$^3$. The temperature difference between the reference specimen ($Al_2O_3$) and the analysed samples during heating at rates of 0.1 and 0.5 °C/s was recorded as a function of temperature.

## Dislocation analysis

Because the convergent electron beams in the STEM mode illuminate the thin TEM samples, the dislocation contrast is often enhanced and the elastic strain contrast in the image background can be reduced[60]. Viewing the dislocation configuration is easier under the simple STEM operation than those of conventional TEM[59,60]. To determine the Burgers vectors of the perfect and immobile dislocations, typical **g·b** analysis (**g** and **b** denote a diffraction vector and Burgers vector, respectively) was conducted on the deformed samples using low-angle ADF via the JEOL NEO-ARM microscope. The dislocation-caused contrasts are maximised or minimised by using a diffraction condition normal to the Burgers vector, such as the dislocation invisible criterion (**g·b** ≠ 0) or invisible criterion (**g·b** = 0)[16,47].

## Molecular dynamic simulations and associated electron diffraction analysis

The MD simulations were performed using the open-source programme LAMMPS[61] with the second nearest-neighbour modified embedded-atom method (2NN MEAM) potential[62] for the FeMnNiCrCo HEA system developed by ref. 63. For all the simulations, temperature and pressure were controlled by the Nosé–Hoover thermostat and barostat[64,65], respectively, using a time step of 4 fs. Initially, cube-shaped fcc single-crystal cells with $1.317 \times 10^6$ atoms were prepared for the orientation of $x$-[$\bar{2}10$], $y$-[$11\bar{2}$], and $z$-[111]. The initial cells were relaxed at 77 K by applying periodic boundary conditions for all directions in an isobaric-isothermal (NPT) ensemble at zero external pressure. A uniaxial compression was then applied along the z-[111] direction of the relaxed cell at constant strain rates of $2 \times 10^7$ (low), $2 \times 10^8$ (moderate), and $2 \times 10^9$ (high) s$^{-1}$. To visualise the evolution of the microstructure during the loading, local atomic arrangements were identified using the polyhedral template matching (PTM) method[66], and information on dislocations was extracted using the dislocation extraction algorithm (DXA)[67], as implemented in the OVITO programme[68]. The total strain was set to 0.08 for the strained $Fe_{40}Mn_{40}Cr_{10}Co_{10}$ (at%) cell structure, while 0.24 was used for the Ni-based MEA cell structures. A smaller strain was used for the FeMnCrCo structure because amorphisation was observed at higher strain level. To provide further information on the microstructure, the virtual selected-area EDP method developed by Coleman et al.[69,70]. was used to make direct comparisons between the present experimental and simulation data. Simulated 200-kV electron radiation ($\lambda = 0.0251$ Å) was used to create the virtual EDPs using a 0.005 Å$^{-1}$ spacing in each direction of the reciprocal lattice meshes. The patterns were produced limiting a reciprocal lattice vector (K) from $0 \leq K \leq 0.85$ Å$^{-1}$ and restricting mesh points to a 0.01 Å$^{-1}$ hemispherical slice from the associated Ewald sphere in the intensity calculation.

## Monte Carlo simulations

The MC simulations were conducted using the open-source programme LAMMPS[61] with a 2NN MEAM potential for the FeMnNiCrCo HEA system[63]. A cell with 216,000 atoms (15.7 nm × 13.6 nm × 12.8 nm) was used for the simulation. To obtain the equilibrium atomic configuration at the temperatures of interest, isothermal-isobaric ensemble (NPT) hybrid MC plus MD simulations were performed for 200 MC steps. After each MC step, cell dimensions and atomic positions were further relaxed by MD simulations for 10 ps by applying periodic boundary conditions for all directions at zero external pressure.

## Data availability

The data that support the findings of this study are available from the corresponding authors upon request.

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

## Acknowledgements

J.B.S. acknowledges financial support from the National Research Foundation of Korea (NRF) grant funded by the Korea government (NRF–2021R1A2C4002622), funded by the Korea government (MSIT) (NRF–2022R1A5A1030054), and by Technology Innovation Program (Alchemist Project, 1415180672, Al-based supercritical materials discovery) funded by the Ministry of Trade, Industry & Energy, Korea. W.S.K. is financially supported by the Future Material Discovery Program of the Korea (No. 2019M3D1A1079214). S.S.S. is supported by the National Research Foundation of Korea (NRF–2020R1C1C1003554) and by the Korea Institute for Advancement of Technology (KIAT) grant funded by the Korean Government (MOTIE, P0002019, The Competency Development Program for Industry Specialist). M.Y.N., and H.J.C. are supported by the Ministry of Trade, Industry, and Energy (MOTIE) of Korea through the project No. N0002598 supervised by the Korea Institute for Advancement of Technology (KIAT). Z.L. acknowledges financial support from the Natural Science Foundation of Hunan Province in China (Grant No. 2021JJ10056). H.S.K. acknowledges financial support from the National Research Foundation of Korea (NRF) grant funded by the Korea government (MSIT) (NRF–2022R1A5A1030054).

## Author contributions

J.B.S. designed the research project. J.B.S. and H.S.K. supervised the study. J.B.S. fabricated the alloy materials. W.S.K. performed the atomistic simulations. J.B.S., M.Y.N, H.J.C., and Y.U.H. worked on the TEM characterisation and data interpretation. J.B.S. prepared TEM samples and conducted the BSE and EBSD experiments. J.G.K., H.S., and S.S.S. performed the mechanical characterisation. J.B.S., Z.L., E.P., and H.S.K. analysed the data and wrote the manuscript. All authors discussed the results, had input and commented on the manuscript.

## Competing interests

The authors declare no competing interests.
