## [Peer Review File · Nature Communications]

Mechanically derived short-range order and its impact on the multi-principal-element alloysREVIEWER COMMENTS

Reviewer #1 (Remarks to the Author):

In this work, the authors studied the mechanically stimulated SRO in a Fe₄₀Mn₄₀Cr₁₀Co₁₀ HEA using TEM observations and MD/MC simulations. The results demonstrate that, unlike chemical SRO, mechanically derived SRO evolves with higher strain rate at 77K. Besides, this SRO, which is attributed to stacking-faults and dislocations, plays a twofold role: hardening by dislocation glide and heat generation by slip-band refinement. This is an interesting work and opens new perspectives to the mechanically-driven SRO. I could recommend for publication after the following comments/suggestions are addressed.

1. MC simulations indicate that there are Cr-Cr SROs in a Fe₄₀Mn₄₀Cr₁₀Co₁₀ HEA and the mean size ranges from 0.7 nm to 14 nm when the temperature increases from 900K to 750K (as shown in Figure 1). Is there any experimental observation to support these simulations (include SRO type and size)? I notice that the authors cite Ref. 17 but it is CoCrNi MEA, which is quite different from the one (Fe₄₀Mn₄₀Cr₁₀Co₁₀ HEA) in this work. Besides, is it possible to plot the size distribution of SROs at 900K and 750K (like Figure 4e) instead of only mention the mean size?
2. Both NiCrCo MEA and Ag are deformed under low, moderate and fast strain rate (as shown in Figure 5d and 6), while Fe₄₀Mn₄₀Cr₁₀Co₁₀ HEA is only deformed under low and moderate strain rate (as plotted in Figure 5b). Is there any reason why the results of Fe₄₀Mn₄₀Cr₁₀Co₁₀ HEA under fast strain rate are not included? Besides, this paper claims that the deformation is highly dependent on the strain rate. However, the simulation strain rate ($10^7 \sim 10^9 \text{ s}^{-1}$) is several order of magnitude higher than the experimental one ($10^{-5} \sim 10^{-3} \text{ s}^{-1}$). In that case, how to prove that the simulation predictions are able to support the experimental observations?
3. The microstructural discrepancies between the experimental sample and the simulation model. The experimental Fe₄₀Mn₄₀Cr₁₀Co₁₀ sample is polycrystalline with a grain size from 3.5 to 5.1 μm , while the simulated Fe₄₀Mn₄₀Cr₁₀Co₁₀ sample is single crystalline. A bi-/poly-crystal model with grain boundaries is highly recommended to add for comparison.
4. The Discussion section is not sufficient. Please add some discussions on the effect of temperature on TSRO (such as 140K/300K vs. 77K), comparisons between mechanically stimulated TSRO and thermally activated CSRO, etc.
5. It seems that there are several typos: i) label in Figure 1c "at 750K"; ii) label of red curve in Figure 4c "HEA-F". Please double check.

Reviewer #2 (Remarks to the Author):

Dear Editor,

The manuscript talks about the mechanically driven SRO in HEA/MEA. If I understand correctly, the concept originates from the 1986 PRB paper by Prof. Littlewood. At that time, the Ge-Si alloy, which was

considered as a model random alloy, showed some SRO when epitaxially grown on Si substrate AND annealed at 450 C. The PRB paper then rationalized the mechanism beneath the observation.

After reading this manuscript, I am overall confused. First of all, what is the definition of the TSRO here? The authors claimed that TSRO appears even in pure Ag metal, indicating a completely different definition than the one in the PRB paper. The authors should give their definition of TSRO both linguistically and mathematically.

Then how does the TSRO appear? The kinetics? In the PRB paper, the mechanically driven SRO appears by annealing, similar to the chemical SRO. How about the TSRO here? Intuitively, the slip would destroy the ordering and make the system approach ideal random.

Technical side, a lot of MD and MC simulations are performed. How accurate is the potential? The prediction of the phase diagram is a non-trivial task. Does the ultrahigh loading strain rate in MD affect the conclusions? I am curious about the accuracy of these simulations. More validations and discussions are needed.

Best wishes

Reviewer #3 (Remarks to the Author):

Chemical short-range ordering recently received significant interest, especially within multi-component alloys due to its important role in tailoring the properties of materials. Seol, JB et al. have presented a detailed and systematic manuscript reporting a mechanically driven “topological short-range order” (TSRO) and its effects in a B-doped Fe₄₀Mn₄₀Cr₁₀Co₁₀ high entropy alloy (HEA), which is novel and interesting.

The authors performed careful characterisation experiments, especially TEM analysis, on HEA samples deformed with strain rates of $\sim 10^{-3}$, $\sim 10^{-4}$ and $\sim 10^{-5}$ at liquid nitrogen temperature. Results indicate that the thermal-induced chemical short-range ordering already pre-existed in the undeformed samples but with a low density. Tensile tests show strain-to-failure increases with strain rate while yield strength remains strain rate-independent. Supported by experimental observations and molecular dynamics simulation results, the authors claimed that the improving ductility with a high strain rate is due to the abundant and homogeneous slip band activities, correlated with the TSRO zones observed within the slip bands. Regardless of the intensive partial dislocation activities, the authors claimed they are minor contributors when compared to slip banding. The TSRO was argued to introduce a “glide plane hardening” effect, i.e., to obstruct dislocations gliding in the slip bands, consequently contributing to improved ductility at high strain rate levels. The authors provided a massive amount of high quality experimental and numerical simulation data; while some of the conclusions are not fully supported by the evidence, they should be addressed before considering for publication.

My major comments are given below for reference,

The authors concretely attributed the super-lattice diffraction spots in the SAED patterns to diffusive scattering, thus evaluating short-range ordering. Yet no sign of diffusive scattering was seen in the diffraction pattern, such as streaks (except SFs), blurry backgrounds. I strongly suggest the authors explain why using the sign of long-range order, i.e. superlattice spots, to characterise short-range ordering.

As presented by the authors, topological short-range ordering, that usually observed in amorphous materials, can be induced by deformation in the selected high entropy alloy, even in pure fcc Ag. This is somehow confusing as the evidence is not sufficient to show the topological nature of the ordering. The cited literature by Littlewood (page 24, line 108) reported a potential strain-induced chemical short-range order without mentioning any topological characteristic. I strongly suggest the authors be extremely careful using the term “topological short-range ordering” unless experimental evidence on the short-range ordering in deformed Ag is provided. Obviously, there is no chemical short-range order in pure Ag, but maybe it is not the case for TSRO.

A few things need to be clarified in the introduction to improve the readability of the manuscript, though some of them are given later in the main text.

The authors did not clarify the reason for conducting tensile tests at different strain rates and at a low temperature of 77 K.

The authors did not clarify the concept of topological SRO.

Some minor comments are given below.

Page 3, line 78. “Localised planar slip ... destroy the pre-existing CSRO domains ...causing glide plane softening^{19,20}.” As reported in both literature (19), and (20), localised planar slip is caused by glide plane softening. In addition, there is a typo in reference 20, it should be “diffraction” instead of “diffractio;n”.

Page 6, line 146. As mentioned by the authors, SRO already exited before straining.

Page 7, line 160. “Final strain to sample failure” is a strain not an “elongation”, as provided in the brackets.

Page 8, line 180. Due to the limited resolution of EBSD, the high density of nanotwins can sometimes be mis-indexed as HCP phase, as presented in Figure 2 and Extended Data Figure 3. This confusion directly undermines the argument that phase transformation was observed in the material. High-resolution TEM on the HCP stacking sequence along $\langle 110 \rangle$ zone axis is essential.

Extended data figure 4. The TEM images and the SAED patterns are not correlated.

Page 8, line 196. If the authors were to identify the slip system by using the Burgers vector of the dislocation slipping in that system, then it should be $a/2\langle 011 \rangle\{111\}$ instead of “ $a\langle 011 \rangle\{111\}$ ”.

Figure 2. The arrangement of the sub-figures is to be improved. In addition, the measurement on the Moiré-fringe was wrong, which should be deleted though, as it shows no practically useful information.

Page 33, line 847. It is necessary to explain why using in-situ heating test to introduce oxidation to the TEM lamellar when one can simply remove the sample from TEM and wait for it. Is it for the purpose of accelerating the oxidation process? If so, the piece of information should be included in the methods.

Lastly, I would like to appreciate the vast amount of work the authors have done and included in the extended data.

AUTHORS' POINT-BY-POINT RESPONSE TO THE REVIEWERS' COMMENTS
Manuscript NCOMMS-22-06614A or NCOMMS-22-06614-T

Dear Reviewers

Here, we reply to the reviewers' comments on our manuscript (NCOMMS-22-06614A or NCOMMS-22-06614-T) entitled *Mechanically derived short-range order and its impact on the multi-principal-element alloys*. All the amendments are outlined below in more detail and highlighted in the revised version of our paper with **blue colour highlighting**.

Reviewer 1:

In this work, the authors studied the mechanically stimulated SRO in a $\text{Fe}_{40}\text{Mn}_{40}\text{Cr}_{10}\text{Co}_{10}$ HEA using TEM observations and MD/MC simulations. The results demonstrate that, unlike chemical SRO, mechanically derived SRO evolves with higher strain rate at 77K. Besides, this SRO, which is attributed to stacking-faults and dislocations, plays a twofold role: hardening by dislocation glide and heat generation by slip-band refinement. This is an interesting work and opens new perspectives to the mechanically-driven SRO. I could recommend for publication after the following comments/suggestions are addressed.

Comment # 1

MC simulations indicate that there are Cr-Cr SROs in a $\text{Fe}_{40}\text{Mn}_{40}\text{Cr}_{10}\text{Co}_{10}$ HEA and the mean size ranges from 0.7 nm to 1.4 nm when the temperature increases from 900K to 750K (as shown in Figure 1). Is there any experimental observation to support these simulations (include SRO type and size)? I notice that the authors cite Ref. 17 but it is CoCrNi MEA, which is quite different from the one ($\text{Fe}_{40}\text{Mn}_{40}\text{Cr}_{10}\text{Co}_{10}$ HEA) in this work. Besides, is it possible to plot the size distribution of SROs at 900K and 750K (like Figure 4e) instead of only mention the mean size?

Author reply to comment #1

We thank the reviewer for the kind support and valuable comments. We fully concur with the suggestion. Rather than citing ref. 17, we newly added one more reference [Liu, D. et al. Chemical short-range

order in $\text{Fe}_{50}\text{Mn}_{30}\text{Co}_{10}\text{Cr}_{10}$ high-entropy alloy. *Mater. Today Nano* **16**, 100139 (2021)] providing the experimental results of chemical SRO (CSRO) in non-equiatomic quaternary $\text{Fe}_{50}\text{Mn}_{30}\text{Co}_{10}\text{Cr}_{10}$ (at%) HEA. They reported that even though Cr, Mn, Fe, and Co are close in atomic size, unusual or particular CSRO does exist in the $\text{Fe}_{50}\text{Mn}_{30}\text{Co}_{10}\text{Cr}_{10}$ HEA, unlike equiatomic ternary VCoNi MEA. In particular, their previous TEM results demonstrated a general tendency toward ‘preference for unlike pairs and avoidance of like pairs’, *i.e.*, Fe–Mn, Fe–Co, Fe–Cr and Mn–Co all exhibited a relatively strong tendency to be neighbours, elevating the variability of atomic packing in the non-equiatomic $\text{Fe}_{50}\text{Mn}_{30}\text{Co}_{10}\text{Cr}_{10}$ HEA. A similar trend was also shown in our $\text{Fe}_{40}\text{Mn}_{40}\text{Cr}_{10}\text{Co}_{10}$ (at%) HEA. In addition to this similarity, our MC simulations showed a ‘preference for Cr–Cr like pairs’ in the $\text{Fe}_{40}\text{Mn}_{40}\text{Cr}_{10}\text{Co}_{10}$ system at 750 K. This intriguing feature of Cr–Cr like pairs was even demonstrated in a ternary equiatomic NiCrCo MEA, suggested in the earlier MC simulations and seen experimentally. This indicates that even though concentrated solid solutions including Cr had different chemical compositions, ‘preference for Cr–Cr like pairs’ would be favourable. Reason for this is presumably due to the complex thermodynamic driving force for generating the CSRO. In fact, thermodynamic driving force for forming the CSRO in a few M/HEAs can stem from diverse paths, for example, fluctuation in the local strain, bonding state, electronic and magnetic interaction. A further in-depth understanding of this issue (determination of CSRO type) demands careful energy computations. However, this additional work to reveal why there is a tendency of ‘preference for Cr–Cr like pairs’ in the non-equiatomic quaternary is beyond the scope of this study, because we primarily focus on mechanically stimulated SRO.

In addition to the type of CSRO, we fully agree that reviewer’s suggestion about plotting the statistical size distribution of CSRO domains via atomistic MC simulations would be beneficial for strengthening our work. Unfortunately, quantitative measurements on the exact size distribution of CSROs, based on the MC simulations, is still an open issue on account of the limitation of the simulation methodology. In other words, it would be impossible to define clear boundaries between the CSROs and the matrix region in the captured snapshot-slices of atomic species configurations with a thickness of 2 or 3 nm, given by the MC simulations. Instead, we could provide the qualitative trend in CSROs with respect to the temperatures of the current HEA system. Furthermore, based on our TEM diffraction patterns, the intensity of CSRO-stimulated diffuse scattering signals for boron-free $\text{Fe}_{40}\text{Mn}_{40}\text{Cr}_{10}\text{Co}_{10}$ HEA (before tensile deformation) was limited to quantitatively determine the degree/extent of the extremely small CSROs. In turn, TEM foils taken from real bulk samples of the $\text{Fe}_{40}\text{Mn}_{40}\text{Cr}_{10}\text{Co}_{10}$ HEA before straining would have possessed the limited degree/extent of chemical SRO domains. All authors, therefore, conclude that even though a comparison of the SRO size distribution between atomistic MC simulations and TEM results is important but challenging, it seems to be beyond the scope of this study.

With consideration of the length of this manuscript, we addressed the statistical size distribution of mechanically derived SRO, determined from dark-filed TEM imaging.

The following changes were made:

Please see the Introduction section on lines between 74 and 106, pages #4~5,

“Extensive efforts to monitor the ordering transition in MPEAs have provided insights into the thermally induced CSROs via simulations^{14,15,26} and X-ray absorption³⁰. However, X-ray-based measurements provide rather low analytical capabilities, restricting the experimental observation of CSROs in the individual grains of the materials, as they are averaged over a comparatively large volume of material¹⁶. Later, transmission electron microscopy (TEM) and atom probe tomography went through a revolution to record the extremely small CSROs within individual fcc grains for metallic alloys¹⁶⁻²¹. For instance, TEM dark-field images, formed with CSRO-induced additional diffuse scattering in reciprocal-space selected-area electron diffraction patterns (EDPs) with specific zone axes, enable quantitative measurements of the CSRO degree/extent in fcc-structured NiCrCo and VNiCo MEAs¹⁶⁻¹⁸. This approach is even applicable for binary Ti-6Al concentrated solution³³ and high-Mn (25 wt%) steel sample³⁴. Overall, CSRO sizes ranged from ~0.7 to ~2.0 nm for a given fcc-structured alloy composition^{34,35}.

In contrast to the superlattice reflections caused by long-range-ordered nanoprecipitates, elastic diffuse scattering between normal fcc Bragg reflections in TEM-EDPs is ascribed to structural deviations (*e.g.*, vacancies, dislocations, stacking faults, and SROs) from a periodic crystalline lattice^{36,37}. Among these defects, the detection of extra diffuse discs at $\frac{1}{2}\{311\}$ locations in the [112] TEM-EDPs are often used to evidence the presence of irregularly distributed CSROs within the individual grains in most fcc-structured alloys, including MPEAs and high-Mn steel. Thus, stacking faults (SFs) in most fcc structures often cause clear streaking along the {111} directions between common fcc Bragg reflections in the EDPs under the [110] zone axis. Recently, Zhang et al.¹⁶ made an intriguing discovery that TEM dark-field imaging, produced with the 'SF-caused streaking', provided the direct evidence of CSROs in a furnace-cooled NiCrCo MEA. They further revealed that increasing the amount of CSRO increased the stacking fault energy (SFE) and hardness. The experimental observations of the work were consistent with those of the previous m

odelling-based studies, where tuning the local chemical order at the nanoscale altered the SFEs and mechanical properties of some MPEAs^{14,26}. However, there have also been points of disagreement about the role of CSROs in the SFEs of MPEAs. Other TEM-based studies suggested that SFEs of MPEAs do not influence the degree of chemical ordering^{24,25}. Therefore, the following fundamental questions still remain: (i) whether SFs are phenomenologically intertwined with the clustered ordering; (ii) whether ordering can affect mechanical twinning and deformation-induced martensitic transformation¹²; and (iii) the mechanism and reason for the ‘deformation-derived disorder-to-SRO transition at low temperatures’, suggested based on experimental observations³⁵ and theoretical calculations³⁸.”

“Hereafter, such strain-induced ordering is denoted as mechanically derived SRO (MSRO) for convenience.”

Please see Results on lines between 130 and 142, page #6,

“.....in the Methods section. Although the near-random spatial distribution of each individual element is observed at 1373 K (Fig. 1a), CSRO formation in the single-crystal system is potentially favourable at 873 and 750 K (Fig. 1b and c, respectively). The associated radial distribution function profiles revealed three features. First, by lowering the temperature, Fe–Co, Mn–Cr, Mn–Co, Fe–Mn, and Fe–Cr pairs, all showed a relatively stronger tendency to be neighbours, promoting the variability of atomic packing sequence in the first neighbouring shell (Fig. 1d–f). This holds for a general CSRO definition, *i.e.*, ‘preference for unlike pairs and avoidance of like pairs’³⁹. Our MC simulations for the non-equiatomic Fe₄₀Mn₄₀Cr₁₀Co₁₀ HEA were consistent with the type of ordering, reported by a recent TEM work³⁹ highlighting Fe–Mn/Co/Cr preference in the first neighbouring shell in an fcc Fe₅₀Mn₃₀Cr₁₀Co₁₀ (at%) system. Second, of all unlike pairs, Fe–Co exhibited the strongest tendency to be neighbors. Third, a primarily strong tendency toward ‘preference for Cr–Cr like pairs’ was generated by lowering the temperatures. Besides these results”

Please see lines 268 to 272 on page #12,

“Concurrent with previous TEM-based studies in which CSROs and chemical medium-range-order domains form an interchangeable preferential ordering in a

NiCrCo MEA^{17,18,21}, our observations suggest that a period of {311} inter-planar spacing in MSRO doubles that of the fcc lattice for the current alloy. This can explain the detection of the diffuse scattering introduced by MSRO at the $\frac{1}{2}\{311\}$ locations in all [112] TEM-EDP and HRTEM- or STEM-FFT results.”

Please see lines 297 to 311 on page #13,

“The B-free Fe₄₀Mn₄₀Cr₁₀Co₁₀ reference alloy before straining was chemically ordered at 750 K, predicted by MC simulations (especially, Cr–Cr pairs in Fig. 1). However, for the system containing CSROs, our MD-EDPs ([112] zone axis) revealed weak or faint diffuse scattering at the $\frac{1}{2}\{311\}$ locations (Supplementary Fig. 9). This indicates that the degree/extent of CSROs in the MC simulations for the reference structure may not yield clear diffuse scattering in the MD-EDPs.

Compared to the B-free system, B-doped Fe₄₀Mn₄₀Cr₁₀Co₁₀ HEA prior to straining was expected to have more and larger CSROs owing to a relatively stronger driving force for CSRO formation in the DTA thermal profiles (Supplementary Fig. 1) and to a strong negative solution enthalpy by B ingress. However, we experimentally detected weak diffuse scattering at the $\frac{1}{2}\{311\}$ locations in the [112] TEM-EDP for the HEA (Fig. 5a). Because our TEM samples include the extremely small CSROs in the real microstructure, a large inelastic background attributed to thermally induced diffuse scattering and plasmon scattering would lower the TEM spatial resolution³⁷. This might be the primary cause for the weak diffuse scattering signals from the B-doped sample before straining.”

Please also see Discussion on lines between 404 and 418, pages #17~18

“Short-range diffusion-mediated CSRO in a few MPEAs can be described in terms of ‘preference for unlike species and avoidance of like pairs’^{17,18,21,39}. Our MC simulations revealed that lowering the temperature of Fe₄₀Mn₄₀Cr₁₀Co₁₀ structure led to a relatively strong tendency for Fe–Co, Mn–Cr, Mn–Co, Fe–Mn, and Fe–Cr pairs, promoting the variability of atomic packing. This trend is also demonstrated experimentally in fcc-structured Fe₅₀Mn₃₀Cr₁₀Co₁₀ (at%) HEA³⁹. What is more interesting is that there is even a strong tendency of ‘preference for Cr–Cr like pairs’ in the current Fe₄₀Mn₄₀Cr₁₀Co₁₀ HEA, where atomic sizes of four species (Fe, Mn, Cr, and Co) are similar³⁹. Likewise, Cr–Cr like pairs were previously demonstrated

in equiatomic NiCrCo MEA, suggested in the MC simulations¹⁴ and seen experimentally^{16,18}. This gives a strong indication that despite different alloy compositions, formation of Cr–Cr-related CSROs would be favourable in the MPEAs. Reason for this is presumably attributed to the complex mechanisms of CSRO formation¹⁹. A further in-depth energy computation is needed to clarify this issue, which is beyond the scope of this study. Rather than reiterating the recent efforts in the CSRO or medium-range order, the current study fills the missing gap on the strain-induced short-range ordering in the MPEAs.”

Please see the revised reference list,

19. Wu, Y. *et al.* Short-range ordering and its effects on mechanical properties of high-entropy alloys. *J. Mater. Sci. Technol.* **62**, 214–220 (2021).
20. Inoue, K., Yoshida, S. & Tsuji, N. Direct observation of local chemical ordering in a few nanometer range in CoCrNi medium-entropy alloy by atom probe tomography and its impact on mechanical properties. *Phys. Rev. Mater.* **5**, 085007 (2021).
21. Wang, J., Jiang, P., Yuan F. & Wu, X. Chemical medium-range order in a medium-entropy alloy. *Nat. Commun.* **13**, 1021 (2022).
36. Cowley, J. M. *Diffraction Physics*, Elsevier, Amsterdam (1981).
37. Zuo, J. M., Pacaud, J. Hoier, R. & Spence, J. C. H. Experimental measurement of electron diffuse scattering in magnetite using energy-filter and imaging plates. *Micron* **31**, 527–532 (2000)
39. Liu, D. *et al.* Chemical short-range order in Fe₅₀Mn₃₀Co₁₀Cr₁₀ high-entropy alloy, *Mater. Today Nano.* **16**, 100139 (2021).

Comment # 2

Both NiCrCo MEA and Ag are deformed under low, moderate and fast strain rate (as shown in Figure 5d and 6), while Fe₄₀Mn₄₀Cr₁₀Co₁₀ HEA is only deformed under low and moderate strain rate (as plotted in Figure 5b). Is there any reason why the results of Fe₄₀Mn₄₀Cr₁₀Co₁₀ HEA under fast strain rate are not included? Besides, this paper claims that the deformation is highly dependent on the strain rate. However, the simulation strain rate ($10^7 \sim 10^9 \text{ s}^{-1}$) is several order of magnitude higher than the

experimental one ($10^{-5} \sim 10^{-3} \text{ s}^{-1}$). In that case, how to prove that the simulation predictions are able to support the experimental observations?

Author reply to comment #2

We fully agree with these issues related to the strain rates. First, we considered here the visualisation of $\text{Fe}_{40}\text{Mn}_{40}\text{Cr}_{10}\text{Co}_{10}$ HEA results for only low ($2 \times 10^7 \text{ s}^{-1}$) and moderate ($2 \times 10^8 \text{ s}^{-1}$) strain rates owing to the limitation of interatomic potential used for the non-equiatomic quaternary HEA system. Smaller strain (0.08) was used for the HEA structure, as amorphisation was observed at a fast strain rate ($2 \times 10^9 \text{ s}^{-1}$). This amorphisation (grey colour in the figure below) becomes more severe for deformation with a fast strain rate, restricting the observation of the effect of strain rate under equivalent conditions. Therefore, we regrettably should present results for only low and moderate strain rates because the presence of result for the fast rate and discussion of the amorphisation would go beyond the scope of the manuscript and, from a more technical perspective, would make the presentation of the results far more complex and non-transparent. Instead, we briefly explained the reason for this in Figure 6b caption in the newly revised manuscript.

$\text{Fe}_{40}\text{Mn}_{40}\text{Cr}_{10}\text{Co}_{10}$ (Fast strain rate)

Figure caption. Evolution of non-equiatomic FeMnCrCo cell structures upon different strain levels with fast strain rate ($2 \times 10^9 \text{ s}^{-1}$) at 77 K, showing severe amorphisation (grey colour) by applying fast strain rate.

The second issue is related to the inherent limitation of atomistic simulations. As the reviewer stated, the experimental time scale would not be accessible with the current MD simulations. The extremely high strain rate of common MD simulations can result in a higher yield and flow stress values in the

stress-strain response and a higher density of dislocations (more than several order of magnitude) compared to the experimental results as reported by previous works [Ko, W. -S. *et al.* Atomistic deformation behavior of single and twin crystalline Cu nanopillars with preexisting dislocations, *Acta Materialia* **197**, 54-68 (2020)]. For example, a discrepancy in the dislocation density between the present MD simulations ($10^{16} \sim 10^{17} / \text{m}^2$) for $\text{Fe}_{40}\text{Mn}_{40}\text{Cr}_{10}\text{Co}_{10}$ HEA and the present TEM experiments ($\sim 10^{10} / \text{m}^2$) for boron-doped $\text{Fe}_{40}\text{Mn}_{40}\text{Cr}_{10}\text{Co}_{10}$ HEA may raise a question how the simulation predictions are able to support the experimental observations. Therefore, we could hardly say that there is a quantitative agreement between experiments and simulations. Instead, we could provide a possible insight into stronger slip planarity due to a higher loading rate, proved by experiments and simulations qualitatively. In addition, the use of a much smaller sample and narrower selected area could compensate the higher dislocation density for the analysis of diffraction. In this regard, an important point to compare the experimental and simulation results is not on the absolute size and loading rates but on whether the simulation can reproduce the slip mechanism shown in the experiments phenomenologically. As shown in the figure, the present MD simulation can reproduce well the slip mechanism via partial dislocations of usual fcc metals depending on specific values of SFE. To clarify this issue, we added the following sentence in the revised manuscript.

The following changes were made:

Please see lines between 118 and 123 on pages #5~6,

“To phenomenologically underpin the TEM results obtained from respective fcc single-grains, we further conducted atomistic Monte Carlo (MC) simulations, molecular dynamics (MD) simulations, and MD-based virtual diffraction analyses in an fcc $\text{Fe}_{40}\text{Mn}_{40}\text{Cr}_{10}\text{Co}_{10}$ single-crystal system. In this context, the experimental results and computational predictions provide insights into the microstructural features attributable to MSRO in the non-equiatomeric FeMnCrCo HEA and other fcc non-equiatomeric NiCrCo MEAs.”

Please see lines between 224 and 229, page #10

“Assuming that there was no atomic diffusion of principal species in the HEA during quasistatic deformation at 77 K, it is pertinent to attribute the additional diffuse scattering to the emergence of strain-induced SRO rather than to the initial CSRO prior to the deformation. Again, MSROs are distributed along the slip band. In contrast with MSRO, the distribution of initial CSROs is highly regular throughout

the individual grain, suggested computationally^{14,15,22,26} and imaged experimentally^{16-21,33,34}.

Please see lines between 371 and 375, page #16

“Deformation at a higher $\dot{\epsilon}$ gives rise to more dislocations (or stronger slip planarity) and more SFs (Fig. 6c). This demonstrates that the increased intensity of MSRO-introduced scattering in reciprocal-space MD-EDPs due to higher $\dot{\epsilon}$ drives the B-doped system towards stronger slip planarity and more SFs. Moreover, there were no LRO precipitates in all of the cell structures. This confirms that spot-like $\frac{1}{2}\{311\}$ scattering in the MD-EDPs is directly correlated with SFs and dislocations, but not with LRO structure.”

Please see Discussion on lines between 439 and 447, page #19

“These results were consistently supported by MD simulations and relevant MD-EDPs. However, three concerns can be raised: (i) a vast difference in the applied $\dot{\epsilon}$ values between the bulk samples (in range of $10^{-5} - 10^{-3} \text{ s}^{-1}$) and the simulations ($10^7 - 10^9 \text{ s}^{-1}$); (ii) the simulated $\text{Fe}_{40}\text{Mn}_{40}\text{Cr}_{10}\text{Co}_{10}$ system had a single-crystalline structure, while experimental B-doped $\text{Fe}_{40}\text{Mn}_{40}\text{Cr}_{10}\text{Co}_{10}$ sample was polycrystalline; (iii) the accuracy of the potential used in the present simulation. The first issue is explained in Supplementary Discussion 1, while additional MD simulations are conducted using a bi-crystal model with grain boundary for the second issue (see Supplementary Fig. 13 and Supplementary Discussion 2). The last issue is elaborated in Supplementary Discussion 3 and Supplementary Fig. 14.”

Please see Figure 6b caption, on page #43,

“The strained cell structures at slow and moderate $\dot{\epsilon}$ were included in this work for comparison, as the highest $\dot{\epsilon}$ ($2 \times 10^9 \text{ s}^{-1}$) caused a severe amorphisation in the system and restricted the observation of the $\dot{\epsilon}$ effect under equivalent conditions.”

Please also see Supplementary Discussion 1 in Supplementary Information

1. The loading rate issue is related to the inherent limitation of atomistic simulations in principle. The extremely high $\dot{\epsilon}$ for usual MD simulation can result in a higher yield and flow stress values in the stress-strain response, leading to higher density of dislocations (more than several order of magnitude) compared to the experimental result as reported by a previous modelling⁴. For example, in the current work, the density of dislocations predicted by the present MD simulations was $10^{16} - 10^{17} / \text{m}^2$, while the density of $\sim 10^{10} / \text{m}^2$ was observed by the current TEM experiments. Additionally, for the MD simulations, amorphisation was observed at the highest $\dot{\epsilon}$ level for the $\text{Fe}_{40}\text{Mn}_{40}\text{Cr}_{10}\text{Co}_{10}$ HEA structure, restricting the observation of $\dot{\epsilon}$ effect on the simulated cell structure under equivalent conditions. However, the use of a much smaller sample and narrower selected area in the present MD simulations could negate the effect of the very high dislocation density in the analysis of diffraction. In this regard, an important point is to compare the experimental and simulation results not based on the absolute size and rate of loading but on whether the simulation can reproduce the slip mechanism of the experiment. As shown in the figure, the present MD simulation can well reproduce the slip mechanism via partial dislocations of usual fcc metals depending on specific values of SFE. With this consideration, our observation is used to give a plausible insight into a qualitative agreement between experiments and simulations.

Comment # 3

The microstructural discrepancies between the experimental sample and the simulation model. The experimental $\text{Fe}_{40}\text{Mn}_{40}\text{Cr}_{10}\text{Co}_{10}$ sample is polycrystalline with a grain size from 3.5 to 5.1 μm , while the simulated $\text{Fe}_{40}\text{Mn}_{40}\text{Cr}_{10}\text{Co}_{10}$ sample is single crystalline. A bi-/poly-crystal model with grain boundaries is highly recommended to add for comparison.

Author reply to comment #3

Many thanks for this helpful comment, we fully agree. In principle, all TEM results in this work were taken from each grain in the B-doped $\text{Fe}_{40}\text{Mn}_{40}\text{Cr}_{10}\text{Co}_{10}$ polycrystalline samples. As the reviewer and general readers recognized, selected-area diffraction patterns and associated images via TEM are typically obtained from one single respective grain in most materials. Similarly, we here determined the degree and spatial extent of SROs by measuring those that exist within each fcc grain or within an individual grain of the HEA. Specifically speaking, dark-field TEM images formed with the extra

scattering appearance in specific beam direction EDPs enable quantitative visualisation of the extremely small CSROs in the respective grain for NiCrCo and VNiCo MEAs¹⁶⁻¹⁸ as well as binary Ti-6Al concentrated solution³³. If selected-area TEM analysis includes two or three grains with different orientations at the same time, then the associated diffraction pattern cannot be interpreted, and thereby, complex spots are displayed in the diffraction patterns. Hence, selected-area diffraction patterns of TEM for the SRO-stimulated diffuse scattering must be taken from each fcc grain or individual grains of the M/HEA materials. This rationalizes that similar to the TEM diffraction patterns obtained from each grain in the polycrystalline samples, MD simulation-based virtual diffractions were acquired from a single crystalline structure in our original manuscript.

Nevertheless, the reviewer's point would be on different conditions of the dislocation nucleation between experiments and simulations, *i.e.*, grain boundary (GB) can act as a heterogeneous site for the dislocation nucleation. With this consideration, we further performed the MD simulations and its virtual diffraction analysis for the deformed Fe₄₀Mn₄₀Cr₁₀Co₁₀ system with bi-crystal model following the reviewer's recommendation. The poly-crystal cannot be applied, as our MD simulation-based virtual diffraction cannot distinguish diffraction in a specific direction from the averaged diffraction from multiple grains, and it was difficult to find and apply the diffraction simulation on the specific crystal direction. In other words, a diffraction pattern can be obtained only in the crystal orientation initially specified for the simulation cell. Even for the bi-crystal, there are not many choices of initial crystal orientation appropriate for the analysis of the [112] zone axis. This is even true for TEM analysis, as reviewer recognized. In our additional simulations, we selected a tilt grain boundary with a boundary plane of {112}, rotation axis of <110>, and misorientation angle of 70.53° (*i.e.*, {112}<110> incoherent twin boundary). The additional simulation now considers the effect of grain boundaries on the dislocation activity. After the MD simulations run, a half of the bi-crystal cell was used for the diffraction analysis. The atomic configurations and resultant MD-EDPs are now presented in Supplementary Fig. 14. The result of bi-crystal cell exhibits the similar EDPs with those of single crystal cell (*i.e.*, the focused scattering along the [112] zone axis), demonstrating that the key outcome of the current simulations is independent of the presence or absence of GBs.

The following changes in Supplementary Information were made:

Please see newly added Supplementary Fig. 13,

Supplementary Fig. 13. MD simulations for the origin of MSRO-generated extra scattering in bi-crystal non-equiatomic FeMnCrCo HEA with the grain boundary upon loading at 77 K. **a**, MD-EDPs for an interstitial-free $\text{Fe}_{40}\text{Mn}_{40}\text{Cr}_{10}\text{Co}_{10}$ (at%) structure before and after straining. The extra diffuse discs at the $\frac{1}{2}\{311\}$ locations in the $[112]$ MD-EDPs become more clear by applying the plastic strain at moderate rate for a given structure system. This is consistent with the result of single-crystal non-equiatomic FeMnCrCo HEA (Fig. 6a). After the MD run, a half of the bi-crystal cell was used for the diffraction analysis. **b**, Corresponding bi-crystal cell structures including a grain boundary (GB).

Please see Supplementary Discussion 2 in Supplementary Information

2. The most of present MD simulations were performed using an initial cell with the single crystal fcc structure. This pristine condition would not take into account the situation in which dislocations are nucleated by interaction with grain boundaries. To ascertain the possibility of this effect, we performed additional MD simulations and virtual diffraction analysis for the deformation of $\text{Fe}_{40}\text{Mn}_{40}\text{Cr}_{10}\text{Co}_{10}$ system with a bi-crystal model. For the analysis of diffraction along the $[112]$ zone axis, we selected a tilt grain boundary with a boundary plane of $\{112\}$, rotation axis of $\langle 110 \rangle$, and misorientation angle of 70.53° (i.e., $\{112\}\langle 110 \rangle$ incoherent twin boundary). After the MD run, a half of the bi-crystal cell was used for the diffraction analysis. The atomic configurations and resultant MD-EDPs are presented in Supplementary Fig. 14. The result of bi-crystal cell exhibiting the similar EDPs with that of single crystal cell (i.e., the focused scattering along the $[112]$ zone axis) demonstrates that the key outcome of the present MD simulations is independent of the presence or absence of GBs.

Comment #4

The Discussion section is not sufficient. Please add some discussions on the effect of temperature on TSRO (such as 140K/300K vs. 77K), comparisons between mechanically stimulated TSRO and thermally activated CSRO, etc.

Author reply to comment #4

We thank the reviewer for the suggestion. We revised the corresponding paragraph according to the reviewer's comment. As well as such amendments, particular attention has been paid to the thorough overhaul throughout the manuscript to improve the readability of this manuscript. Furthermore, the arrangement of the main text was newly improved.

The following changes were made:

Please see Discussion section on lines between 419 and 438 on pages #18~19,

“To reiterate our TEM findings from B-doped $\text{Fe}_{40}\text{Mn}_{40}\text{Cr}_{10}\text{Co}_{10}$ HEA after tensile deformation with different $\dot{\epsilon}$ at 77 K, diffuse scattering at the $\frac{1}{2}\{311\}$ locations in the [112]-indexed EDPs came into focus at specific regions (strain-induced slip bands) in the deformation fcc structure. More SROs in the 77 K-deformed HEA were formed than those in the undeformed sample with the same alloy composition, and they were distributed along the slip bands. Particularly, applying higher $\dot{\epsilon}$ under quasistatic conditions increased the intensity of MSRO-derived $\frac{1}{2}\{311\}_{\text{fcc}}$ diffuse scattering from faint signals to spot-like ones, which in turn yielded more and larger SROs in the slip bands. We again highlight that the spot-like scattering monitored in the TEM-EDPs, HRTEM- or STEM-FFTs, and MD-EDPs is explicitly derived by a high degree of MSRO, but not by fcc-based LRO precipitates. The precursor of MSRO-derived diffuse discs and spot-like scattering for the strained MPEAs was directly correlated with strain-induced crystal imperfections (dislocations and mechanical SFs), which were originally driven by a strain-induced atomic packing mismatch. Furthermore, from the viewpoint of mechanical response, the non-equiatomically quaternary alloy system exhibited no σ_{YS} dependence on the $\dot{\epsilon}$ values. On account of these unique features, we herein regarded the localised SROs as mechanically derived

SROs (MSROs) rather than diffusion-mediated CSROs that would be destroyed by planar dislocations during deformation.

Our STEM images further revealed the microstructural features responsible for such ordering in the non-equiatomic quaternary HEA, *i.e.*, MSROs in fcc grains were attributed to strain-induced structural deviations (specifically, dislocations and SFs) from a periodic lattice.”

Also, see lines 448 to 463, on pages #19~20

“We showed that for the 77 K-strained Fe₄₀Mn₄₀Cr₁₀Co₁₀ HEA, the detection of the SF-induced streaking along the {111} directions in [110] MD-EDPs corresponded to the existence of MSRO-derived $\frac{1}{2}\{311\}$ diffuse discs in [112] MD-EDPs. This is even true for other 77 K-strained non-equiatomic NiCrCo structures. An earlier TEM study demonstrated that highly intensified streaking along the {111} directions in [110] EDPs was directly correlated with high-degree CSROs in a NiCrCo MEA¹⁶. According to our findings and those previously reported, it is highly plausible that an increase in the diffuse-disc intensity due to a higher $\dot{\epsilon}$ reflects a higher degree of MSROs, more SFs, and more dislocations in the deformation structures for the non-equiatomic MPEAs. To sum up, displacive MSRO belongs to isostructural disorder–order transition that occurs in slip bands during 77 K-tensile deformation (specifically, in the plastic strain regime). We assume that MSRO might be defined to be the degree of strain-induced local deviation from the average local-scale ordering in terms of either chemical or structural occupation. Unfortunately, a kinetics investigation of MSRO evolution has still not been performed. Further investigation to figure out how MSROs are mechanically stabilised with a higher $\dot{\epsilon}$, for example, at different low temperatures and levels of applied plastic strain, would be interesting.”

On lines between 467~487, pages #20~21

“If the observed slip bands are completely developed in the large strain regime, the associated MSROs will cause glide plane hardening^{44,45}, based on our observations of non-shearable MSROs (Figs. 4 and 5d, and Supplementary Fig. 5a). However, in-depth characterisation of non-shearable MSROs is still an open issue. A higher $\dot{\epsilon}$ -induced hardening after the onset of yielding arises from high degree/extent of MSRO, elevating severe lattice distortion effect^{28,34,35,55}. Although twinning-induced

plasticity (TWIP) or transformation-induced plasticity (TRIP) effects can be active in the large-strain regime and arrest the strain localisation^{2,40,56}, the current multilength-scale characterisation revealed that both were independent of the applied $\dot{\epsilon}$ values. The TWIP or TRIP effects are less affected by MSRO formation, as the ordering emerges with the slip bands.

As the evolution of individual slip bands is suppressed by dislocation-source exhaustion in fcc-based metals^{57,58}, the denser bands at higher $\dot{\epsilon}$ are associated with the high density of MSRO. More and larger MSROs are likely to generate large amount of localised heat inside the slip bands, as the SRO phenomenon (either CSRO or MSRO) is a heat generation process. This MSRO-induced heat will stimulate the slip-band refinement, *i.e.*, new slip bands can be generated near pre-existing ones, thereby lowering severe strain localisation in a single slip band⁴⁵. Accordingly, new slip bands can be refined with further straining at a given $\dot{\epsilon}$. This is phenomenologically analogous to the so-called dynamic slip-band refinement⁴⁴. We anticipate that this slip-band refinement effect is caused by a higher $\dot{\epsilon}$ and governs the ϵ_{total} dependence on the $\dot{\epsilon}$. Thus, the current study can provide important insights into the mechanical properties of many fcc-structured alloys that are designed to take advantage of the disorder–order transition.”

Comment #5

It seems that there are several typos: i) label in Figure 1c “at 750K”; ii) label of red curve in Figure 4c “HEA-F”. Please double check.

Author reply to comment #5

We appreciate the hint and corrected the typos. The terminology of “HEA-F” is revised as “HEA-H” (for high strain rate).

The following changes in figures were made:

Please see Figure 1c, on page #32,

Please also see Figure 5c (formerly, Figure 4c),

Reviewer 2:

Comment #1

The manuscript talks about the mechanically driven SRO in HEA/MEA. If I understand correctly, the concept originates from the 1986 PRB paper by Prof. Littlewood. At that time, the Ge-Si alloy, which was considered as a model random alloy, showed some SRO when epitaxially grown on Si substrate AND annealed at 450 C. The PRB paper then rationalized the mechanism beneath the observation. After reading this manuscript, I am overall confused. First of all, what is the definition of the TSRO here? The authors claimed that TSRO appears even in pure Ag metal, indicating a completely different definition than the one in the PRB paper. The authors should give their definition of TSRO both linguistically and mathematically.

Then how does the TSRO appear? The kinetics? In the PRB paper, the mechanically driven SRO appears by annealing, similar to the chemical SRO. How about the TSRO here? Intuitively, the slip would destroy the ordering and make the system approach ideal random.

Author reply to comment #1

We would like to express our sincere appreciation to the reviewer for these immensely helpful and detailed comments. We would like to apologise for the overall confusing. We fully agree that the former version of the manuscript did cause a misunderstanding, particularly at “TSRO appears even in pure Ag metal”. We fully agree that this statement in the former version of our manuscript was confusing. After authors’ careful discussions on the systematic results, we concluded that Ag perfect crystal should be disordered irrespective of loading. With this consideration, we deleted the corresponding paragraph about Ag simulations to avoid the confusion. Furthermore, please see the authors’ response on the comment #4 raised by referee #1 for the revised Discussion section.

The following changes were made:

Please see updated Abstract, lines between 28 and 40 on page #2,

“Chemical short-range order in disordered solid solutions often emerges with specific heat treatments. Here, we demonstrate that unlike the thermally activated ordering, mechanically derived short-range order (MSRO) in a multi-principal-element $\text{Fe}_{40}\text{Mn}_{40}\text{Cr}_{10}\text{Co}_{10}$ (at%) alloy originates from tensile deformation at liquid- N_2 temperature, and its degree/extent can be tailored by tuning loading rates under quasistatic conditions. The mechanical response and multi-length-scale characterisation consistently pointed to the minor contribution of MSRO formation to yield strength, mechanical twinning, and deformation-induced displacive transformation. Scanning and high-resolution transmission electron microscopy together with diffraction patterns revealed both the microstructural features responsible for such ordering and the dependence of the ordering degree/extent on the applied strain rates. Underpinned by the molecular dynamics simulations, we show that the MSRO is driven by strain-induced structural deviations (dislocations and stacking faults) from a periodic lattice, offering new perspectives on the ordering transition and mechanistic understanding of multi-principal-element alloys at low temperatures.”

Please see Introduction section, lines between 59 and 64 on page #3,

“When compositionally homogeneous structures have specific low-coordination-number clusters, the preferential local ordering of principal elements generally dominates over the spatial order of a few nearest-neighbour spacings, *i.e.*, short-range

order (SRO), often called chemical SRO (CSRO)^{22,23}.

Formation of diffusion-mediated CSRO domains in fully disordered alloys belongs to thermally activated isostructural disorder-to-order transition at short ranges.”

“It has been widely conjectured that localised planar slip and leading dislocations would destroy the pre-existing CSRO domains in a face-centred-cubic (fcc) phase upon loading, which corresponds to the so-called glide plane softening^{31,32}. This effect drives the non-random system towards ideal random, referred as the ‘deformation-derived CSRO-to-disorder transition’.”

Please see lines between 86 and 106 on pages #4~5,

“In contrast to the superlattice reflections caused by long-range-ordered nanoprecipitates, elastic diffuse scattering between normal fcc Bragg reflections in TEM-EDPs is ascribed to structural deviations (*e.g.*, vacancies, dislocations, stacking faults, and SROs) from a periodic crystalline lattice^{36,37}. Among these defects, the detection of extra diffuse discs at $\frac{1}{2}\{311\}$ locations in the [112] TEM-EDPs are often used to evidence the presence of irregularly distributed CSROs at individual grains in most fcc-structured alloys, including MPEAs and high-Mn steel. Thus, stacking faults (SFs) in most fcc structures often cause clear streaking along the {111} directions between common fcc Bragg reflections in the EDPs under the [110] zone axis. Recently, Zhang et al.¹⁶ made an intriguing discovery that TEM dark-field imaging, produced with the 'SF-caused streaking', provided the direct evidence of CSROs in a furnace-cooled NiCrCo MEA. They further revealed that increasing the amount of CSRO increased the stacking fault energy (SFE) and hardness. The experimental observations of the work were consistent with those of the previous modelling-based studies, where tuning the local chemical order at the nanoscale altered the SFEs and mechanical properties of some MPEAs^{14,26}. However, there have also been points of disagreement about the role of CSROs in the SFEs of MPEAs. Other TEM-based studies suggested that SFEs of MPEAs do not influence the degree of chemical ordering^{24,25}. Therefore, the following fundamental questions still remain: (i) whether SFs are phenomenologically intertwined with the clustered ordering; (ii) whether ordering can affect mechanical twinning and deformation-induced martensitic transformation¹²; and (iii) the mechanism and reason for the ‘deformation-derived disorder-to-SRO transition at low temperatures’, suggested

based on experimental observations³⁵ and theoretical calculations³⁸.”

Please see lines between 114 and 115 on page #5,

“Hereafter, such strain-induced ordering is denoted as mechanically derived SRO (MSRO) for convenience.”

Please see the Results section, lines between 130 and 142 on page #6,

“Although the near-random spatial distribution of each individual element is observed at 1373 K (Fig. 1a), CSRO formation in the single-crystal system is potentially favourable at 873 and 750 K (Fig. 1b and c, respectively). The associated radial distribution function profiles revealed three features. First, by lowering the temperature, Fe–Co, Mn–Cr, Mn–Co, Fe–Mn, and Fe–Cr pairs, all showed a relatively stronger tendency to be neighbours, promoting the variability of atomic packing sequence in the first neighbouring shell (Fig. 1d–f). This holds for a general CSRO definition, *i.e.*, ‘preference for unlike pairs and avoidance of like pairs’³⁹. Our MC simulations for the non-equiatomic Fe₄₀Mn₄₀Cr₁₀Co₁₀ HEA were consistent with the type of ordering, reported by a recent TEM work³⁹ highlighting Fe–Mn/Co/Cr preference in the first neighbouring shell in an fcc Fe₅₀Mn₃₀Cr₁₀Co₁₀ (at%) system. Second, of all unlike pairs, Fe–Co exhibited the strongest tendency to be neighbors. Third, a primarily strong tendency toward ‘preference for Cr–Cr like pairs’ was generated by lowering the temperatures”

Please see the lines between 152 and 156 on page #7,

“Additionally, to validate the efficiency of the thermal treatments applied for CSRO generation, we conducted differential thermal analysis (DTA) of the B-doped and B-free reference samples subjected to the ageing and furnace cooling. Results of the DTA profiles are provided in Supplementary Fig. 1, indicating that the thermal treatments employed in this work are feasible for generating CSRO (the details are described in Supplementary Notes 1).”

Please see the Supplementary Notes 1 and 2 in Supplementary Information

“1. In principle, CSRO formation is heat-generation process. The advent of

exothermic peaks in DTA thermal profiles gives the concrete hint the CSRO formation in most solid solutions owing to its strong negative ΔH_{mix} (heat generation). As shown in Supplementary Fig. 1, prominent exothermic peaks in the heating process of both samples were observed. We found that there was a shift in the exothermic peak to lower temperature due to B doping: 797 K for the B-doped HEA, while 806 K for the B-free case. Moreover, the peak height increases with B ingress for a given continuous heating rate. These results can imply a strong thermodynamic driving force for forming CSRO owing to the B ingress. This rationalizes our aforementioned anticipation that high degree or large extent of Cr rich CSROs can be introduced by both B doping and ageing plus furnace cooling employed herein.

2. In this study, we conducted tensile tests at different strain rates and at 77 K. Reasons for this are as follows. First, our previous TEM result suggested that lowering of tensile testing temperatures from 298 to 77 K leads to both deformation-induced order transition and resultant planar dislocation glide in boron-doped $\text{Fe}_{40}\text{Mn}_{40}\text{Cr}_{10}\text{Co}_{10}$ (at%) HEA³. However, it failed to explore the origin of strain-driven SRO in the alloy. Second, reducing the tensile testing temperatures from 298 to 77 K can affect the SFE values of the alloy substantially, as SFE is a function of temperature. In other words, varying the loading temperatures cannot decipher whether planar dislocation glide in fcc metallic alloys is either due to SRO or due to SFE. In fact, consensus has not been reached on the origin of planar dislocation glide, mainly due to the fact that both SRO and SFE have the strong influence on the glide mode of dislocations and associated hardening in glide softening- or hardening-dominated fcc solid solutions. Lastly, chemical SRO is a thermally activated or diffusion-mediated process. This implies that the formation of chemical SRO in M/HEAs is sensitive to loading rates and loading temperatures owing to deformation-introduced heating. Hence, we intentionally selected the tensile testing parameters, to minimise the loading temperatures on the SFE value and to minimise the deformation-induced heating during tensile tests.

Please see the lines between 224 and 229 on page #10,

“Assuming that there was no atomic diffusion of principal species in the HEA during quasistatic deformation at 77 K, it is pertinent to attribute the additional diffuse scattering to the emergence of strain-induced SRO rather than to the initial CSRO

prior to the deformation. Again, MSROs are distributed along the slip band. In contrast with MSRO, the distribution of initial CSROs is highly regular throughout the individual grain, suggested computationally^{14,15,22,26} and imaged experimentally^{16-21,33,34}.

Also, see the Discussion section, lines between 404 and 463 on pages #17~20,

“Short-range diffusion-mediated CSRO in a few MPEAs can be described in terms of ‘preference for unlike species and avoidance of like pairs’^{17,18,21,39}. Our MC simulations revealed that lowering the temperature of Fe₄₀Mn₄₀Cr₁₀Co₁₀ structure led to a relatively strong tendency for Fe–Co, Mn–Cr, Mn–Co, Fe–Mn, and Fe–Cr pairs, promoting the variability of atomic packing. This trend is also demonstrated in fcc-structured Fe₅₀Mn₃₀Cr₁₀Co₁₀ (at%) HEA, experimentally³⁹. What is more interesting is that there is even a strong tendency of ‘preference for Cr–Cr like pairs’ in the current Fe₄₀Mn₄₀Cr₁₀Co₁₀ HEA, where atomic sizes of four species (Fe, Mn, Cr, and Co) are similar³⁹. Likewise, Cr–Cr like pairs were previously demonstrated in equiatomic NiCrCo MEA, suggested in the MC simulations¹⁴ and seen experimentally^{16,18}. This gives a strong indication that despite different alloy compositions, formation of Cr–Cr-related CSROs would be favourable in the MPEAs. Reason for this is presumably attributed to the complex mechanisms of CSRO formation¹⁹. A further in-depth energy computation requires space for clarifying this issue, which is beyond the scope of this study. Rather than reiterating the recent efforts in the CSRO or medium-range order, the current study fills the missing gap on the strain-induced short-range ordering in the MPEAs.

To recapitulate our TEM findings from B-doped Fe₄₀Mn₄₀Cr₁₀Co₁₀ HEA after tensile deformation with different ϵ at 77 K, diffuse scattering at the $\frac{1}{2}\{311\}$ locations in the [112]-indexed EDPs came into focus at specific regions (strain-induced slip bands) in the deformation fcc structure. More SROs in the 77 K-deformed HEA were formed than those in the undeformed sample with the same alloy composition, and they were distributed along the slip bands. Particularly, applying higher ϵ under quasistatic conditions increased the intensity of MSRO-derived $\frac{1}{2}\{311\}_{\text{fcc}}$ diffuse scattering from faint signals to spot-like ones, which in turn yielded more and larger SROs in the slip bands. We again highlight that the spot-like scattering monitored in the TEM-EDPs, HRTEM- or STEM-FFTs, and MD-EDPs is explicitly derived by a high degree of MSRO, but not by fcc-based LRO precipitates. The precursor of

MSRO-derived diffuse discs and spot-like scattering for the strained MPEAs was directly correlated with strain-induced crystal imperfections (dislocations and mechanical SFs), which were originally driven by a strain-induced atomic packing mismatch. Furthermore, from the viewpoint of mechanical response, the non-equiatom quaternary alloy system exhibited no σ_{YS} dependence on the $\dot{\epsilon}$ values. On account of these unique features, we herein regarded the localised SROs as mechanically derived SROs (MSROs) rather than diffusion-mediated CSROs that would be destroyed by planar dislocations during deformation.

Our STEM images further revealed the microstructural features responsible for such ordering in the non-equiatom quaternary HEA, *i.e.*, MSROs in fcc grains were attributed to strain-induced structural deviations (specifically, dislocations and SFs) from a periodic lattice. These results were supported by MD simulations and relevant MD-EDPs, consistently. However, three concerns can be raised: (i) a vast difference in the applied $\dot{\epsilon}$ values between the bulk samples (in range of $10^{-5} - 10^{-3} \text{ s}^{-1}$) and the simulations ($10^7 - 10^9 \text{ s}^{-1}$); (ii) the simulated $\text{Fe}_{40}\text{Mn}_{40}\text{Cr}_{10}\text{Co}_{10}$ system had a single-crystalline structure, while experimental B-doped $\text{Fe}_{40}\text{Mn}_{40}\text{Cr}_{10}\text{Co}_{10}$ sample was polycrystalline; (iii) the accuracy of the potential used in the present simulation. The first issue is explained in Supplementary Discussion 1, while additional MD simulations are conducted using a bi-crystal model with grain boundary for the second issue (see Supplementary Fig. 13 and Supplementary Discussion 2). The last issue is elaborated in Supplementary Discussion 3 and Supplementary Fig. 14.

We showed that for the 77 K-strained $\text{Fe}_{40}\text{Mn}_{40}\text{Cr}_{10}\text{Co}_{10}$ HEA, the detection of the SF-induced streaking along the $\{111\}$ directions in $[110]$ MD-EDPs corresponded to the existence of MSRO-derived $\frac{1}{2}\{311\}$ diffuse discs in $[112]$ MD-EDPs. This is even true for other 77 K-strained non-equiatom NiCrCo structures. An earlier TEM study demonstrated that highly intensified streaking along the $\{111\}$ directions in $[110]$ EDPs was directly correlated with high-degree CSROs in a NiCrCo MEA¹⁶. According to our findings and those previously reported, it is highly plausible that an increase in the diffuse-disc intensity due to a higher $\dot{\epsilon}$ reflects a higher degree of MSROs, more SFs, and more dislocations in the deformation structures for the non-equiatom MPEAs. To sum up, displacive MSRO belongs to isostructural disorder–order transition that occurs in slip bands during 77 K-tensile deformation (specifically, in the plastic strain regime). We assume that MSRO might be defined to be the degree of strain-induced local deviation from the average local-

scale ordering in terms of either chemical or structural occupation. Unfortunately, a kinetics investigation of MSRO evolution has still not been performed. Further investigation to figure out how MSROs are mechanically stabilised with a higher $\dot{\epsilon}$, for example, at different low temperatures and levels of applied plastic strain, would be interesting..”

Comment #2

Technical side, a lot of MD and MC simulations are performed. How accurate is the potential? The prediction of the phase diagram is a non-trivial task. Does the ultrahigh loading strain rate in MD affect the conclusions? I am curious about the accuracy of these simulations. More validations and discussions are needed.

Author reply to comment #2

We thank the reviewer for the suggestion, we fully concur. According to the reviewer’s comment, critical concerns can be raised: (i) the accuracy of interatomic potential is questionable, and (ii) there was a vast difference in the applied $\dot{\epsilon}$ values between the bulk samples (in range of $10^{-5} - 10^{-3} \text{ s}^{-1}$) and the simulations ($10^7 \sim 10^9 \text{ s}^{-1}$).

Regarding the first potential accuracy issue, the accuracy and transferability of such interatomic potential (2NN MEAM) was detailed at the Methods section in the original paper, focusing on fundamental physical properties and properties related to the deformation (e.g., compositional dependence of the yielding tendency and the sluggish diffusion) in comparison with experiments. We must admit that the interatomic potential for HEAs used in the present study is not perfect and exhibits some weaknesses. We already confirmed the problem of unexpected amorphization at too high strain level (here we kindly refer to comment #2 of the first referee). However, we would like to clarify that the key outcome of the present MD simulations is independent of the selection of interatomic potential. To evaluate the dependency of MD results on the selection of interatomic potential, we have performed additional simulations based on another interatomic potential for HEAs [Gröger, R., Vitek, V. & Dlouhý, A. Effective pair potential for random fcc CoCrFeMnNi alloys. *Modelling and Simulation in Materials Science and Engineering*, **28**, 075006 (2020)] based on a pairwise Lennard-Jones (briefly, LJ) model. This potential leads to higher stacking fault energies than the 2NN MEAM potential [Choi, W. M., Jo, Y. H., Sohn, S. S., Lee, S. & Lee, B. J. Understanding the physical metallurgy of the CoCrFeMnNi high-entropy alloy: an atomistic simulation study. *Npj Comput. Mater.* **4**, 1–9 (2018).], as listed in Supplementary Table 1. To see the possible dependence of deformation behaviour on the

interatomic potential, we have focused on the evolution of MD-EDPs for [112] zone axis which is the essential result supporting our main conclusion. As shown in Supplementary Fig. 14, newly added results by the simple pairwise LJ potential are well consistent with the results by the 2NN MEAM potential used in our original manuscript (e.g., the advent of diffuse scattering, the loading rate dependence of the scattering intensity, and a strong slip planarity upon a higher loading rate).

The second issue is related to the inherent limitation of atomistic simulations. As the reviewer stated, the experimental time scale would be not accessible with the current MD simulations. The extremely high strain rate of usual MD simulation can result in a higher yield and flow stress values in the stress-strain response and higher density of dislocations (more than several order of magnitude) compared to the experimental result as reported by previous works [Ko, W. -S. *et al.* Atomistic deformation behavior of single and twin crystalline Cu nanopillars with preexisting dislocations, *Acta Materialia* **197**, 54-68 (2020)]. Therefore, we can hardly say that there is a quantitative agreement between experiment and simulation, but we can only provide a possible insight which can be derived by the qualitative agreement. For example, a discrepancy in the density of dislocation observed by the present MD simulations ($10^{16} \sim 10^{17} / \text{m}^2$) and the present experiment ($\sim 10^{10} / \text{m}^2$) may raise a question why there is a good agreement between observed TEM-EDPs and MD-EDPs. However, the use of very smaller sample and narrower selected area can compensate the very higher dislocation density for the analysis of diffraction. In this regard, an important point to compare the experimental and simulation results is not on the absolute size and rate of loading but on whether the simulation can reproduce the slip mechanism of the experiment. As shown in the figure, the present MD simulation can well reproduce the slip mechanism via partial dislocations of usual fcc metals depending on specific values of SFE. This critical issue is additionally described in Supplementary Discussion 3. Lastly, in the revised manuscript we newly suggest that future work to accurately decipher how the MSROs are mechanically stable, and whether the MSRO is either non-shearable or shearable would be necessary.

The following changes in supplementary information were made:

Please see newly added Supplementary Fig. 14

b Strained at low rate

c Strained at moderate rate

Supplementary Fig. 14. MD simulations, based on the Lennard-Jones potential model, for single-crystal non-equiatomic FeMnCrCo HEA upon loading at 77 K. **a**, Evolution of MD-EDPs along the [112] zone axis for the HEA structure after straining at low (left panel) and moderate (right panel) loading rates. The diffuse scattering at the $1/2\{311\}$ locations is shown, while the relative intensity of the diffuse scattering increases with a higher strain rate. These two features obtained from the Lennard-

Jones (LJ) potential model are well consistent with those acquired from different potential model (2NN MEAM) (Fig. 6a). **b, c**, Corresponding cell structure strained at low and moderate loading rates, respectively, showing the distribution of deformation-induced SFs (middle panel) and dislocations with different Burgers vectors (right panel). Applying a higher loading rate elevated both stronger slip planarity and more SFs, which was well matched with those acquired from different potential model (2NN MEAM) (Fig. 6b).

Please see the revised reference list,

21. Wang, J., Jiang, P., Yuan F. & Wu, X. Chemical medium-range order in a medium-entropy alloy. *Nat. Commun.* **13**, 1021 (2022).
36. Cowley, J. M. *Diffraction Physics*, Elsevier, Amsterdam (1981).
37. Zuo, J. M., Pacaud, J. Hoier, R. & Spence, J. C. H. Experimental measurement of electron diffuse scattering in magnetite using energy-filter and imaging plates. *Micron* **31**, 527–532 (2000).
39. Liu, D. *et al.* Chemical short-range order in Fe₅₀Mn₃₀Co₁₀Cr₁₀ high-entropy alloy, *Mater. Today Nano.* **16**, 100139 (2021).
42. Ma, E. Unusual dislocation behavior in high-entropy alloys. *Scr Mater.* **181**, 127–133 (2020).

Please see the revised Supplementary Table 1,

Supplementary Table 1. **Stacking fault energies (SFEs) of fcc-based MPEAs.** SFEs were determined from the molecular statics simulations at 0 K using the second nearest-neighbor modified embedded-atom method (2NN MEAM) and Lennard-Jones (LJ) interatomic potentials. The calculations were performed considering random fcc solid solutions at concerned compositions below.

System	Composition (at%)	SFEs (mJ/m ²)	Interatomic potential	Reference
FeMnCrCo HEA	Fe ₄₀ Mn ₄₀ Cr ₁₀ Co ₁₀	-83.2	2NN MEAM	1

NiCoCr MEA	Ni ₅₀ Co ₂₅ Cr ₂₅	11.1		
	Ni ₆₀ Co ₂₀ Cr ₂₀	33.7		
	Ni ₇₀ Co ₁₅ Cr ₁₅	52.8		
FeMnCrCo HEA	Fe ₄₀ Mn ₄₀ Cr ₁₀ Co ₁₀	25.3	LJ	2

Please see newly added Supplementary Discussion 3 in Supplementary Information

3. To evaluate the dependency of MD results on the selection of interatomic potential, we have performed additional simulations based on another interatomic potential for the current HEA system based on a pairwise LJ model². This potential leads to higher stacking fault energies than the 2NN MEAM potential¹, as listed in Supplementary Table S1. To see the possible dependence of deformation behaviour on the interatomic potential, we have focused on the evolution of MD-EDPs for [112] zone axis which is the essential result supporting our main conclusion. As shown in Supplementary Fig. 14, newly added results obtained from the simple pairwise LJ potential are well consistent with the results from the 2NN MEAM potential (e.g., the advent of diffuse scattering, the loading rate dependence of the scattering intensity, and a strong slip planarity upon a higher loading rate). Hence, we demonstrate that the important findings from the present MD simulations (by 2NN MEAM potential) are independent of the choice of interatomic potential.

Supplementary References

1. Choi, W. M., Jo, Y. H., Sohn, S. S., Lee, S. & Lee, B. J. Understanding the physical metallurgy of the CoCrFeMnNi high-entropy alloy: an atomistic simulation study. *Npj Comput. Mater.* **4**, 1–9 (2018).
2. Gröger, R., Vitek, V. & Dlouhý, A. Effective pair potential for random fcc CoCrFeMnNi alloys. *Model. Simul. Mater. Sci. Eng.*, **28**, 075006 (2020).
3. Seol, J. B. *et al.* Short-range order strengthening in boron-doped high-entropy alloys for cryogenic applications. *Acta Mater.* **194**, 366–377 (2020).
4. Ko, W. -S. *et al.* Atomistic deformation behavior of single and twin crystalline Cu nanopillars with preexisting dislocations, *Acta Mater.* **197**, 54–68 (2020).

Reviewer 3:

Chemical short-range ordering recently received significant interest, especially within multi-component alloys due to its important role in tailoring the properties of materials. Seol, JB et al. have presented a detailed and systematic manuscript reporting a mechanically driven “topological short-range order” (TSRO) and its effects in a B-doped Fe₄₀Mn₄₀Cr₁₀Co₁₀ high entropy alloy (HEA), which is novel and interesting.

The authors performed careful characterisation experiments, especially TEM analysis, on HEA samples deformed with strain rates of $\sim 10^{-3}$, $\sim 10^{-4}$ and $\sim 10^{-5}$ at liquid nitrogen temperature. Results indicate that the thermal-induced chemical short-range ordering already pre-existed in the undeformed samples but with a low density. Tensile tests show strain-to-failure increases with strain rate while yield strength remains strain rate-independent. Supported by experimental observations and molecular dynamics simulation results, the authors claimed that the improving ductility with a high strain rate is due to the abundant and homogeneous slip band activities, correlated with the TSRO zones observed within the slip bands. Regardless of the intensive partial dislocation activities, the authors claimed they are minor contributors when compared to slip banding. The TSRO was argued to introduce a “glide plane hardening” effect, i.e., to obstruct dislocations gliding in the slip bands, consequently contributing to improved ductility at high strain rate levels. The authors provided a massive amount of high quality experimental and numerical simulation data; while some of the conclusions are not fully supported by the evidence, they should be addressed before considering for publication. My major comments are given below for reference,

Major comment # 1

The authors concretely attributed the super-lattice diffraction spots in the SAED patterns to diffusive scattering, thus evaluating short-range ordering. Yet no sign of diffusive scattering was seen in the diffraction pattern, such as streaks (except SFs), blurry backgrounds. I strongly suggest the authors explain why using the sign of long-range order, i.e. superlattice spots, to characterise short-range ordering.

Author reply to comment #1

We thank the reviewer for this critical comment. We fully concur the suggestion. We revise the text

accordingly. At this point, we would like differentiate the diffracted signs in electron diffraction patterns between long-range-order (LRO) precipitates and SROs. First of all, common $L1_2$ -structured fcc-ordered LRO precipitate, for example Ni_3Al structure, provides extra superlattice reflections at the $\{201\}$ locations, *i.e.*, halfway between the transmission spot (000) and the $\{402\}$ spots, and at $\{110\}$ locations in the TEM-EDPs along the $[112]$ zone axis. This experimental observation is also predicted in our MD-EDPs (Supplementary Fig. 8a). As another example, $L1_2$ -structured LRO precipitate in high-Mn fcc-austenitic steel provides the diffraction spots at the same locations with those of LRO- Ni_3Al . In other words, our analysis of the spot-like scattering at the $\frac{1}{2}\{311\}_{\text{fcc}}$ in the TEM-EDPs, HRTEM- or STEM-FFTs, and MD-EDPs is explicitly derived by a high degree of MSRO, but not by fcc-based LRO precipitates. In addition to this EDP analysis, other features obtained from the current mechanical response and DTA profiles provided strong indications that during quasistatic tensile deformation at 77 K, the current HEA system exhibited the MSRO formation and no sudden LRO formation. Lastly, in the MD simulations revealing the evolution of cell structures with respect to loading rates, there was no LRO structure in the strained cell structures.

The following changes were made:

Please see the Introduction section on lines between 78~106, pages #4~5

“Later, transmission electron microscopy (TEM) and atom probe tomography went through a revolution to record the extremely small CSROs within individual fcc grains for metallic alloys^{16–21}. For instance, TEM dark-field images, formed with CSRO-induced additional diffuse scattering in reciprocal-space selected-area electron diffraction patterns (EDPs) with specific zone axes, enable quantitative measurements of the CSRO degree/extent in fcc-structured NiCrCo and VNiCo MEAs^{16–18}. This approach is even applicable for binary Ti–6Al concentrated solution³³ and high-Mn (25 wt%) steel sample³⁴. Overall, CSRO sizes ranged from ~0.7 to ~2.0 nm for a given fcc-structured alloy composition^{34,35}.”

In contrast to the superlattice reflections caused by long-range-ordered nanoprecipitates, elastic diffuse scattering between normal fcc Bragg reflections in TEM-EDPs is ascribed to structural deviations (*e.g.*, vacancies, dislocations, stacking faults, and SROs) from a periodic crystalline lattice^{36,37}. Among these defects, the detection of extra diffuse discs at $\frac{1}{2}\{311\}$ locations in the $[112]$ TEM-EDPs are often used to evidence the presence of irregularly distributed CSROs at individual grains in most fcc-structured alloys, including MPEAs and high-Mn steel. Thus, stacking

faults (SFs) in most fcc structures often cause clear streaking along the {111} directions between common fcc Bragg reflections in the EDPs under the [110] zone axis. Recently, Zhang et al.¹⁶ made an intriguing discovery that TEM dark-field imaging, produced with the 'SF-caused streaking', provided the direct evidence of CSROs in a furnace-cooled NiCrCo MEA. They further revealed that increasing the amount of CSRO increased the stacking fault energy (SFE) and hardness. The experimental observations of the work were consistent with those of the previous modelling-based studies, where tuning the local chemical order at the nanoscale altered the SFEs and mechanical properties of some MPEAs^{14,26}. However, there have also been points of disagreement about the role of CSROs in the SFEs of MPEAs. Other TEM-based studies suggested that SFEs of MPEAs do not influence the degree of chemical ordering^{24,25}. Therefore, the following fundamental questions still remain: (i) whether SFs are phenomenologically intertwined with the clustered ordering; (ii) whether ordering can affect mechanical twinning and deformation-induced martensitic transformation¹²; and (iii) the mechanism and reason for the 'deformation-derived disorder-to-SRO transition at low temperatures', suggested based on experimental observations³⁵ and theoretical calculations³⁸."

Please see the Results section on lines between 152~157, page #7

"Additionally, to validate the efficiency of the thermal treatments applied for CSRO generation, we conducted differential thermal analysis (DTA) of the B-doped and B-free reference samples subjected to the ageing and furnace cooling. Results of the DTA profiles are provided in Supplementary Fig. 1, indicating that the thermal treatments employed in this work are feasible for generating CSRO (the details are described in Supplementary Notes 1)."

Also, see the newly added Supplementary Notes 1 in Supplementary Information file

1. In principle, CSRO formation is heat-generation process. The advent of exothermic peaks in DTA thermal profiles gives the concrete hint that the CSRO formation in most solid solutions is due to the strong negative ΔH_{mix} (heat generation). As shown in Supplementary Fig. 1, prominent exothermic peaks in the heating process of both samples were observed. We found that there was a shift in the exothermic peak to lower temperature due to B doping: 797 K for the B-doped HEA, while 806 K for the B-free case. Moreover, the peak height increases with B ingress for a given

continuous heating rate. These results can imply a strong thermodynamic driving force for forming CSRO owing to the B ingress. This rationalizes our aforementioned anticipation that high degree or large extent of Cr rich CSROs can be introduced by both B doping and ageing employed herein.

Please see the Results section on lines between 173~179, page #8

“Additionally, nanoscale CSROs and long-range order (LRO) precipitates in HEAs would have enhanced the ϵ sensitivity of some alloys⁴². Yet, the current HEA did not follow these typical trends: *i.e.*, the ϵ dependence of σ_{YS} was negligible. From this unusual ϵ sensitivity, unlike the generally known impact of either CSRO or LRO precipitate that acts as a hardening source^{27-29,43}, it seems that our case may require an additional stress or dragging force to promote the glide of mobile dislocations in the plastic strain regime^{29,43}.”

Please see the Results section on lines between 224~229, page #10

“Assuming that there was no atomic diffusion of principal species in the HEA during quasistatic deformation at 77 K, it is pertinent to attribute the additional diffuse scattering to the emergence of strain-induced SRO rather than to the initial CSRO prior to the deformation. Again, MSROs are distributed along the slip band. In contrast with MSRO, the distribution of initial CSROs is highly regular throughout the individual grain, suggested computationally^{14,15,22,26} and imaged experimentally^{16-21,33,34}.”

Please see the Results section on lines between 268~272, page #12

“Concurrent with previous TEM-based studies in which CSROs and chemical medium-range-order domains form an interchangeable preferential ordering in a NiCrCo MEA^{17,18,21}, our observations suggest that a period of {311} inter-planar spacing in MSRO doubles that of the fcc lattice for the current alloy. This can explain the detection of the diffuse scattering introduced by MSRO at the $\frac{1}{2}\{311\}$ locations in all [112] TEM-EDP and HRTEM- or STEM-FFT results.”

Also, see the Results section on lines between 326 and 346 on pages #14~15

“At this point, we differentiate between the MSRO-derived spot-like scattering in

TEM-EDPs or HRTEM-FFTs and the typical LRO-caused superlattice reflections. Even in the TEM-EDPs along the [112] zone axis, the locations ($\frac{1}{2}\{311\}$) of the spot-like scattering for the current HEA (Fig. 5b) were distinct from those ($\{201\}$ and $\{110\}$) of LRO-introduced reflections for other fcc structures (Supplementary Fig. 8). These LRO-caused spots are also found in another fcc-based FeNiCrCoCuAl_{0.5} HEA⁵², where $L1_2$ precipitates generate extra superlattice reflections at the $\{201\}$ and $\{110\}$ locations in the [112] EDPs. Hence, deciphering where diffracted spots or diffused discs are located in the EDPs, for a given fcc structure, provides a feasible route to demarcate the boundary between nano-sized LRO precipitation and high-degree MSRO. Apart from this approach, the transition from SROs to LRO structures during quasistatic tensile deformation at 77 K would be energetically unfavourable. Earlier studies suggested that clustered SROs can self-aggregate into larger medium-range-order domains or even nano-sized LRO precipitates, only when thermal treatments are applied at elevated temperatures, *i.e.*, above ~ 770 K^{22,53}. In other words, the SRO-to-LRO transition is accompanied by a change in the solution enthalpy from strongly negative to strongly positive values. However, tensile testing at 77 K, conducted in this study, is relatively deficient in the thermal energy required for LRO formation in fcc-structured HEA. Therefore, our analysis of the transition from $\frac{1}{2}\{311\}$ diffuse scattering (for HEA-M) to $\frac{1}{2}\{311\}$ spot-shaped discs (for HEA-H) in the [112] TEM-EDPs reveals extremely high degree/extent of MSROs upon loading at a higher $\dot{\epsilon}$, but not sudden LRO formation during tensile deformation at 77 K. This assertion is thus supported by our MD simulations, described in next section.”

Also, see lines between 371 and 375 on page #16

“This demonstrates that the increased intensity of MSRO-introduced scattering in reciprocal-space MD-EDPs due to higher $\dot{\epsilon}$ drives the B-doped system towards stronger slip planarity and more SFs. Moreover, there were no LRO precipitates in all of the cell structures. This confirms that spot-like $\frac{1}{2}\{311\}$ scattering in the MD-EDPs is directly correlated with SFs and dislocations, but not with LRO structure.”

Also, see the Figure 6b caption

“There were no LRO precipitates in all cell structures. This affirms that spot-like scattering at the $\frac{1}{2}\{311\}$ locations in the MD-EDPs is directly correlated with SFs and dislocations, but not with LRO structure.”

Also, see the newly added Supplementary Notes 3 in Supplementary Information file

3. Our MD-EDPs along the [001], [112], [110], and [111] zone axes for common $L1_2$ - Ni_3Al structure are shown in Supplementary Fig. 8a. The predicted diffraction spots, caused by ordering in the structure, were highlighted by the arrows in all MD-EDPs. Especially, in the [112] MD-EDP, whose zone axis is suitable for directly imaging SRO (CSRO or MSRO) in the microstructure, chemically $L1_2$ -structured LRO provides extra superlattice reflections at $\{201\}$ locations, *i.e.*, halfway between the transmission spot (000) and the $\{402\}$ spots, and at $\{110\}$ locations. The locations of the LRO-generated extra diffraction spots, determined by MD-EDPs, are consistently detected in the TEM-EDPs. This is true even for the [112] zone axis. The TEM-EDPs along the [001], $[112]^3$, and [110] zone axes for a common $L1_2$ or $L'1_2$ LRO precipitate, $(\text{Fe,Mn})_{3-x}\text{AlC}_x$ in high-Mn fcc-austenitic steel, are shown in Supplementary Fig. 8b. With this comparison, we suggest that locations of MSRO-derived diffuse scattering in the TEM-EDPs are well predicted by the MD-EDPs.”

Major comment # 2

As presented by the authors, topological short-range ordering, that usually observed in amorphous materials, can be induced by deformation in the selected high entropy alloy, even in pure fcc Ag. This is somehow confusing as the evidence is not sufficient to show the topological nature of the ordering. The cited literature by Littlewood (page 24, line 108) reported a potential strain-induced chemical short-range order without mentioning any topological characteristic. I strongly suggest the authors be extremely careful using the term “topological short-range ordering” unless experimental evidence on the short-range ordering in deformed Ag is provided. Obviously, there is no chemical short-range order in pure Ag, but maybe it is not the case for TSRO.

Author reply to comment #2

Many thanks for this helpful comment, we fully agree. First of all, we would like to apologise for the overall confusing. We fully agree that our original manuscript did cause a misunderstanding, particularly at “TSRO appears even in pure Ag metal”. We fully agree that this statement is confusing.

After authors' careful discussions on the systematic results, we concluded that Ag perfect crystal should be disordered irrespective of loading, and deleted the corresponding paragraph. Hence, please kindly look at the Author reply to comment #1 raised by the Reviewer #2.

Major comment # 3

A few things need to be clarified in the introduction to improve the readability of the manuscript, though some of them are given later in the main text.

Author reply to comment #3

We thank the reviewer for the comment. Particular attention has been paid to the thorough overhaul throughout the manuscript to improve the readability of this manuscript. Furthermore, the arrangement of the main text was newly improved.

The following changes were made:

Please see the Introduction section on lines between 59~64 on page #3,

“When compositionally homogeneous structures have specific low-coordination-number clusters, the preferential local ordering of principal elements generally dominates over the spatial order of a few nearest-neighbour spacings, *i.e.*, short-range order (SRO), often called chemical SRO (CSRO)^{22,23}.

Formation of diffusion-mediated CSRO domains in fully disordered alloys belongs to thermally activated isostructural disorder-to-order transition at short ranges.”

“ It has been widely conjectured that localised planar slip and leading dislocations would destroy the pre-existing CSRO domains in a face-centred-cubic (fcc) phase upon loading, which corresponds to the so-called glide plane softening^{31,32}. This effect drives the non-random system towards ideal random, referred as the ‘deformation-derived CSRO-to-disorder transition’. Meanwhile, increasing the degree and spatial extent of the CSROs can strengthen fcc solid solutions by

activating the planarity of dislocation slip and by modifying the strain hardening capacity^{16,23,27}.”

Also, see lines between 74~106 on pages #4~5,

“Extensive efforts to monitor the ordering transition in MPEAs have provided insights into the thermally induced CSROs via simulations^{14,15,26} and X-ray absorption³⁰. However, X-ray-based measurements provide rather low analytical capabilities, restricting the experimental observation of CSROs in the individual grains of the materials, as they are averaged over a comparatively large volume of material¹⁶. Later, transmission electron microscopy (TEM) and atom probe tomography went through a revolution to record the extremely small CSROs within individual fcc grains for metallic alloys¹⁶⁻²¹. For instance, TEM dark-field images, formed with CSRO-induced additional diffuse scattering in reciprocal-space selected-area electron diffraction patterns (EDPs) with specific zone axes, enable quantitative measurements of the CSRO degree/extent in fcc-structured NiCrCo and VNiCo MEAs¹⁶⁻¹⁸. This approach is even applicable for binary Ti-6Al concentrated solution³³ and high-Mn (25 wt%) steel sample³⁴. Overall, CSRO sizes ranged from ~0.7 to ~2.0 nm for a given fcc-structured alloy composition^{34,35}.

In contrast to the superlattice reflections caused by long-range-ordered nanoprecipitates, elastic diffuse scattering between normal fcc Bragg reflections in TEM-EDPs is ascribed to structural deviations (*e.g.*, vacancies, dislocations, stacking faults, and SROs) from a periodic crystalline lattice^{36,37}. Among these defects, the detection of extra diffuse discs at $\frac{1}{2}\{311\}$ locations in the [112] TEM-EDPs are often used to evidence the presence of irregularly distributed CSROs at individual grains in most fcc-structured alloys, including MPEAs and high-Mn steel. Thus, stacking faults (SFs) in most fcc structures often cause clear streaking along the {111} directions between common fcc Bragg reflections in the EDPs under the [110] zone axis. Recently, Zhang et al.¹⁶ made an intriguing discovery that TEM dark-field imaging, produced with the 'SF-caused streaking', provided the direct evidence of CSROs in a furnace-cooled NiCrCo MEA. They further revealed that increasing the amount of CSRO increased the stacking fault energy (SFE) and hardness. The experimental observations of the work were consistent with those of the previous modelling-based studies, where tuning the local chemical order at the nanoscale

altered the SFEs and mechanical properties of some MPEAs^{14,26}. However, there have also been points of disagreement about the role of CSROs in the SFEs of MPEAs. Other TEM-based studies suggested that SFEs of MPEAs do not influence the degree of chemical ordering^{24,25}. Therefore, the following fundamental questions still remain: (i) whether SFs are phenomenologically intertwined with the clustered ordering; (ii) whether ordering can affect mechanical twinning and deformation-induced martensitic transformation¹²; and (iii) the mechanism and reason for the ‘deformation-derived disorder-to-SRO transition at low temperatures’, suggested based on experimental observations³⁵ and theoretical calculations³⁸. ”

“The reasons for choosing this non-equiatomic HEA composition are explained in the Methods section”

“Hereafter, such strain-induced ordering is denoted as mechanically derived SRO (MSRO) for convenience.”

Also, see the Methods section between 536~543 on pages #22~23,

“The following rationale explains the reason for selecting Fe₄₀Mn₄₀Cr₁₀Co₁₀ HEA with 30 ppm of B in this work: (i) upon tensile loading at 77 K, both deformation twinning and martensitic phase transformation from fcc to hcp phases were observed³⁵; and (ii) the alloy system is likely to form MSROs upon loading at an initial strain rate of 10⁻³ s⁻¹ at 77 K³⁵. Hence, the alloy compositions are indispensable for elucidating the mechanism of MSRO formation and its impact on the HEA. Although the concentrated solid solutions investigated in this study are the same as those in our previous work³⁵, different thermal treatments of low-temperature ageing and furnace cooling were employed here to infuse the CSROs into the alloy compositions before deformation.”

Major comment # 4

The authors did not clarify the reason for conducting tensile tests at different strain rates and at a low temperature of 77 K.

Author reply to comment #4

We thank the reviewer for the suggestion, we fully concur. We added the reasons for conducting tensile tests at different strain rates and at a low temperature of 77 K in the revised manuscript.

The following changes were made:

Please see the Results section on lines 162~163, page #7

“The $\dot{\epsilon}$ values and loading temperature were intentionally chosen to minimise deformation-induced heating during tensile tests (see the Supplementary Notes 2 for details).”

Also, see the newly added Supplementary Notes 2 in Supplementary Information

2. In this study, we conducted tensile tests at different strain rates and at 77 K. Reasons for this are as follows. First, our previous TEM result suggested that lowering of tensile testing temperatures from 298 to 77 K leads to both deformation-induced order transition and resultant planar dislocation glide in boron-doped $\text{Fe}_{40}\text{Mn}_{40}\text{Cr}_{10}\text{Co}_{10}$ (at%) HEA³⁵. However, it failed to explore the origin of strain-driven SRO in the alloy. Second, reducing the tensile testing temperatures from 298 to 77 K can affect the SFE values of the alloy substantially, as SFE is a function of temperature. In other words, varying the loading temperatures cannot decipher whether planar dislocation glide in fcc metallic alloys is either due to SRO or due to SFE. In fact, consensus has not been reached on the origin of planar dislocation glide, mainly due to the fact that both SRO and SFE have the strong influence on the glide mode of dislocations and associated hardening in glide softening- or hardening-dominated fcc solid solutions. Lastly, chemical SRO is usually a thermally activated or diffusion-mediated process. This implies that the formation of chemical SRO in M/HEAs is sensitive to loading rates and loading temperatures owing to deformation-introduced heating. Hence, we intentionally

selected the tensile testing parameters, to minimise the loading temperatures on the low SFE value and to minimise the deformation-induced heating during tensile tests.”

Minor comment #5

The authors did not clarify the concept of topological SRO.

Author reply to comment #6

We fully agree. We used the term “mechanically derived SRO (MSRO)” instead of “TSRO” in the revised manuscript. We clarified this term by using a better expression in the revised manuscript. Please kindly see the Author reply to comment #1 raised by Reviewer #2 for details.

The following changes were made:

Please see Discussion section on lines between 428 and 435, pages #18~19

“The precursor of MSRO-derived diffuse discs and spot-like scattering for the strained MPEAs was directly correlated with strain-induced crystal imperfections (dislocations and mechanical SFs), which were originally driven by a strain-induced atomic packing mismatch. Furthermore, from the viewpoint of mechanical response, the non-equiatomic quaternary alloy system exhibited no σ_{YS} dependence on the $\dot{\epsilon}$ values. On account of these unique features, we herein regarded the localised SROs as mechanically derived SROs (MSROs) rather than diffusion-mediated CSROs that would be destroyed by planar dislocations during deformation.”

Please see lines between 456 and 463, pages #19~20

“To sum up, displacive MSRO belongs to isostructural disorder–order transition that occurs in slip bands during 77 K-tensile deformation (specifically, in the plastic strain regime). We assume that MSRO might be defined to be the degree of strain-induced local deviation from the average local-scale ordering in terms of either chemical or structural occupation. Unfortunately, a kinetics investigation of MSRO evolution has still not been performed. Further investigation to figure out how MSROs are mechanically stabilised with a higher $\dot{\epsilon}$, for example, at different low temperatures and levels of applied plastic strain, would be interesting.”

Minor comment #6

Page 3, line 78. “Localised planar slip ... destroy the pre-existing CSRO domains ...causing glide plane softening^{19,20}.” As reported in both literature (19), and (20), localised planar slip is caused by glide plane softening. In addition, there is a typo in reference 20, it should be “diffraction” instead of “diffractio;n”.

Author reply to comment #6

We thank the reviewer for the correction.

The following changes were made:

Please see the revised reference list,

32. Pekin, T. C., Gammer, C., Ciston, J., Ophus, C. & Minor, A. M. In situ nanobeam electron diffraction strain mapping of planar slip in stainless steel. *Scr. Mater.* **146**, 87–90 (2018).

Minor comment #7

Page 6, line 146. As mentioned by the authors, SRO already exited before straining.

Author reply to comment #7

We appreciate the hint and deleted the typos in the revised manuscript.

Minor comment #8

Page 7, line 160. “Final strain to sample failure” is a strain not an “elongation”, as provided in the brackets.

Author reply to comment #8

We appreciate the hint and revised the typos.

The following changes were made:

Please see lines 166~167 on page #7,

“increased the maximum tensile strength (σ_{UTS}) and total elongation (ϵ_{total}).”

Minor comment #9

Page 8, line 180. Due to the limited resolution of EBSD, the high density of nanotwins can sometimes be mis-indexed as HCP phase, as presented in Figure 2 and Extended Data Figure 3. This confusion directly undermines the argument that phase transformation was observed in the material. High-resolution TEM on the HCP stacking sequence along $\langle 110 \rangle$ zone axis is essential.

Author reply to comment #9

We thank the reviewer for the suggestion, and we fully concur that the high density of nanotwins can sometimes be mis-indexed. In fact, for the density measurements on nano-twins, we used HRTEM and STEM images not EBSD. For the thicknesses of mechanical twins with $> \sim 40$ nm, we used EBSD images. We revised the text accordingly.

The following changes were made:

Please see lines 194~200 on page #9,

“The tuneable $\dot{\epsilon}$ had a negligible effect on the mean thickness of mechanical twins including nano-twins (Fig. 3c), determined from dark-field TEM, high-resolution (HR)TEM, and scanning (S)TEM imaging along the [011] beam direction. Instead, increasing $\dot{\epsilon}$ led to both more slip bands and band refinement (Supplementary Fig. 3b); the mean widths of slip bands in HEA-H, HEA-M, and HEA-L samples were measured to be 53–114, 88–190, and 210–382 nm, respectively. This experimental observation refers to a relatively stronger slip planarity with a higher $\dot{\epsilon}$.”

Minor comment #10

Extended data figure 4. The TEM images and the SAED patterns are not correlated.

Author reply to comment #10

In Supplementary Fig. 3 (formerly, Extended Data figure 4), the SAED patterns (upper panels) obtained from regions of mechanical twinning and slip band were representatively displayed for simple comparison of SAED patterns from twinning and slip band. Hence, we revised the Supplementary Fig. 3 caption for better clarity.

The following changes were made:

Please see the Supplementary Fig. 3 caption

“Supplementary Fig. 3. **Comparison of TEM-EDPs from mechanical twinning and slip bands for the tensile-tested samples at 77 K.** **a**, Representative selected-area EDPs (top panel) and corresponding coloured surface plots in three dimensions, taken from mechanical twins (arrows and ‘Tw’ symbol). In the plots, the color scales with spot intensity on the electron diffractions (blue: background, green: low-intensity, yellow: high-intensity). Bottom panels: Representative TEM-bright field and dark-field images of mechanical twins in the deformation structure. **b**, Representative TEM-EDP and corresponding for coloured surface plots. Bottom panels: Representative TEM images of slip bands (SBs) formed in the HEA-S (left panel) and HEA-M (right panel) samples. With increasing the strain rates, the mean width of SBs that were inclined to $\langle 110 \rangle$ directions lying on $\{111\}$ planes was reduced.”

Minor comment #11

Page 8, line 196. If the authors were to identify the slip system by using the Burgers vector of the dislocation slipping in that system, then it should be $a/2\langle 011 \rangle \{111\}$ instead of “ $a\langle 011 \rangle \{111\}$ ”

Author reply to comment #11

Many thanks for this comment and we revised the typos.

The following changes were made:

Please see lines between 213 and 214 on page #9

“Using an appropriate zone axis, we found that active slip system was $\mathbf{a}/2\langle 011 \rangle\{111\}$ (\mathbf{a} is the lattice parameter) rather than $\mathbf{a}/2\langle 100 \rangle\{001\}$ (Fig. 3d).”

Minor comment #12

Figure 2. The arrangement of the sub-figures is to be improved. In addition, the measurement on the Moiré-fringe was wrong, which should be deleted though, as it shows no practically useful information.

Author reply to comment #12

We appreciate the hint and deleted the typos.

The following changes were made:

Please see lines between 234 and 235 on page #10

The Moiré-fringes were thus shown along the $\langle 112 \rangle$ -type twin in the high-angle annular dark-field (HAADF) and bright-field HRTEM images (Supplementary Fig. 5).

Minor comment #13

Page 33, line 847. It is necessary to explain why using in-situ heating test to introduce oxidation to the TEM lamellar when one can simply remove the sample from TEM and wait for it. Is it for the purpose of accelerating the oxidation process? If so, the piece of information should be included in the methods.

Author reply to comment #13

Many thanks for this helpful comment. As the reviewer recognized, diffracted spots are also introduced when the surface of thin TEM foils oxidises⁴⁸. Hence, we should distinguish the MSRO-derived spot-like scattering from the surface oxidation-caused spots. With an attempt to distinguish the MSRO-derived spot-like scattering from the surface oxidation-caused spots, we further performed in-situ TEM heating experiments of the current HEA samples for accelerating the oxidation process.

The following changes were made:

Please see lines between 235~248, on pages #10~11

On occasion, diffracted spots are also introduced when the surface of thin TEM foils oxidises⁴⁸. With an attempt to distinguish the MSRO-derived spot-like scattering from the surface oxidation-caused spots, we further performed in-situ TEM heating experiments of the current HEA samples for accelerating the oxidation process (see the Methods for details). In the STEM and HRTEM images acquired after the in-situ experiments, heating-introduced surface oxidation (presumably, complex and layered oxide scale including thermodynamically stable Cr_2O_3)⁴⁸ onto the surface of the TEM foils provokes plentiful extra diffracted scattering in the [112] and [110] STEM-FFTs (Supplementary Fig. 6). Moreover, the locations of diffracted spots caused by the deliberate oxidation is unequivocally distinguished from those of MSRO-derived spot-like scattering. Hence, the spot-like scattering shown in the HRTEM-FFT (Fig. 3e) stems from the MSRO formation in the microstructure (particularly, inside slip bands) rather than the surface oxide scale of TEM foils. Eventually, the slip-band refinement with respect to a higher ϵ can be explained in terms of the MSROs that emerge inside the slip bands.”

Please see the Methods section

“In-situ heating TEM for intentionally accelerating surface oxidation of the TEM foils

To intentionally accelerate the surface oxidation process of thin TEM foils, we performed in-situ TEM heating experiments of the current HEA samples. The in-situ TEM heating experiments were performed using a Gatan heating holder equipped with water connection on a conventional TEM (JEOL JEM-2010 without any correctors) equipment at 500 °C for 20 min. Afterwards, the surface oxidation of the TEM lamellar was examined using HRTEM and STEM modes in an atomic-resolution TEM (JEOL NEO-ARM) for accurately distinguishing the SRO-derived diffuse scattering from the oxidation-induced reflections in the EDPs.

Minor comment #14

Lastly, I would like to appreciate the vast amount of work the authors have done and included in the extended data.

Author reply to comment #14

We would like to express our sincere appreciation to the reviewer for immensely helpful and detailed reports for improving our manuscript. All fruitful comments raised by the reviewer make our manuscript much stronger.

REVIEWER COMMENTS

Reviewer #1 (Remarks to the Author):

My comments have been addressed properly. Therefore, I would recommend it for publication.

Reviewer #2 (Remarks to the Author):

The clarity of this second version draft has improved significantly. The authors stated that "the MSRO is driven by strain-induced structural deviations (dislocations and stacking faults) from a periodic lattice". But still, I have some questions about the significance and interpretations of the experimental results.

1.

Is this MSRO a unique phenomenon in Fe-Mn-Cr-Co? I assume no. It is well-known that dislocations and stacking faults massively appear when any ductile metal is plastically deformed. So is MSRO an alternative terminology for this well-known phenomenon?

Lattice distortions are well-known in HEAs even for single crystals. Is this lattice distortion MSRO according to your definition?

2.

If the MSRO is unique in FeMnCrCo, then why is it unique? What is the precondition to having MSRO during loading?

3.

It has been proposed that CSRO can be used to tailor the alloy properties in the fabrication process. On the contrary, it seems that MSRO is caused during the plastic loading. Then how to make use of the MSRO?

4.

A detail about the CSRO. In Fig. 1, the authors plotted some purple circles to indicate the CSRO. Then what is the definition of the CSRO here?

According to the Warren-Cowley short-range order parameter, CSRO is an averaged property of a homogeneous system, not the local clustering as indicated by the circles. Do you think there is no CSRO outside the circles?

Reviewer #3 (Remarks to the Author):

The authors did a fantastic job with the revisions. The revised manuscript looks excellent and I have no

further comments.

I now recommend the paper be accepted and published!

AUTHORS' POINT-BY-POINT RESPONSE TO THE REVIEWERS' COMMENTS
Manuscript NCOMMS-22-06614A

Dear Reviewers

Here, we reply to the reviewers' comments on our manuscript (NCOMMS-22-06614A) entitled *Mechanically derived short-range order and its impact on the multi-principal-element alloys*. All the amendments are outlined below in more detail and highlighted in the revised version of our paper with **blue colour highlighting**.

With best regards on behalf of the author team,

Prof. Jae Bok Seol & Prof. Hyung Seop Kim (corresponding authors)

Reviewer 1:

My comments have been addressed properly. Therefore, I would recommend it for publication.

Author reply

We would like to express our sincere appreciation to the reviewer for the clear recommendation for the publication of our work. Once again, thank you so much for your contribution and time.

Reviewer 2:

The clarity of this second version draft has improved significantly. The authors stated that "the MSRO is driven by strain-induced structural deviations (dislocations and stacking faults) from a periodic lattice". But still, I have some questions about the significance and interpretations of the experimental results.

Comment #1

Is this MSRO a unique phenomenon in Fe-Mn-Cr-Co? I assume no. It is well-known that dislocations and stacking faults massively appear when any ductile metal is plastically deformed. So is MSRO an alternative terminology for this well-known phenomenon? Lattice distortions are well-known in HEAs even for single crystals. Is this lattice distortion MSRO according to your definition?

Author reply to comment #1

We thank the reviewer for the kind support and valuable comments. We fully concur that dislocations and stacking faults massively appear when any ductile metal is plastically deformed. We, therefore, deleted “unique” term in the 2nd revised manuscript. Instead, in order to avoid any confusing, we outline that this MSRO can form in multi-principal-element alloys with low stacking-fault energies (SFEs) upon load at 77 K. In other words, in this revision, we addressed that MSRO in the multi-principal-element alloys with low SFEs tends to form upon load at 77 K, which is driven by crystalline lattice defects (dislocations and stacking faults). This is related to the following two facts: i) Dislocations and stacking faults in alloys with lower SFEs are more active compared to that in alloys with higher SFEs; ii) The lower temperature (i.e., 77 K) further reduces the SFEs of FCC-based alloys and hence intensify the activities of dislocations and stacking faults.

As suggested by the reviewer, lattice distortions are also well-known in many HEAs and MEAs. Actually, one can hardly distinguish the lattice distortions from the SROs in HEAs due to the multi-component-induced chemical complexity [Ma, Y. *et al.* Chemical short-range orders and the induced structural transition in high-entropy alloys. *Scr. Mater.* **144**, 64-68 (2018)]. In this second revision, we stated that a higher strain rate hardens the current HEA after the onset of yielding and activates more and larger strain-induced MSRO, possibly promoted by the lattice distortion effect, particularly in the large-strain regime. Namely, we assume that the MSRO formed along slip bands during deformation at 77 K can be promoted by the intrinsic lattice distortion effect.

The following changes were made:

In Abstract on page #2,

“Underpinned by molecular dynamics, we show that MSRO in the alloys with low stacking-fault energies tends to form upon load at 77 K, which is driven by crystalline lattice defects (dislocations and stacking faults), offering new perspectives on the strain-induced ordering transition.”

Please see lines between 86 and 88 on page #4,

“Recently, medium-range order with 1–5 nm in a dual-phase $\text{Al}_{9.5}\text{CrCoNi}$ (at%) MEA has been temporarily defined as next-level order structure beyond CSRO²¹.”

Please see lines between 117 and 121 on page #5,

“Hereafter, such strain-induced ordering is alternatively denoted as mechanically derived SRO (MSRO) for convenience. We then elucidate the evolution of the microstructure with different loading rates under quasistatic conditions across multiple length scales, so as to investigate the vital effect of MSRO on the mechanical twins and deformation-induced martensitic transformation at 77 K.”

In Discussion section on page #18,

“Diffusion-controlled CSRO forms in the $\text{Fe}_{40}\text{Mn}_{40}\text{Cr}_{10}\text{Co}_{10}$ furnace-cooled HEA with a low SFE, and it coexists with the strain-induced MSRO in the deformation structure at 77 K. We found that the MSRO is primarily formed inside slip bands, and it is originated by strain-induced crystalline defects (SFs and dislocations).”

Also, please see the page #20,

“We showed that for the 77 K-strained $\text{Fe}_{40}\text{Mn}_{40}\text{Cr}_{10}\text{Co}_{10}$ HEA with a low SFE,....”

“...deformation structures for the non-equiatomic MPEAs with lower SFEs.”

“We assume that MSRO might be defined to be the degree of strain-induced local deviation from the average local-scale ordering in terms of either chemical occupation or structural defects, particularly for the MPEAs with low SFEs and upon loading at 77 K. Especially, low-temperature deformation, employed here, can lead to a lowering of the SFEs of fcc-based alloys and hence intensify the activities of dislocations and stacking faults. Unfortunately, a kinetics investigation of MSRO

evolution has still not been performed, and it is unclear whether the MSRO does involve the elemental enrichments of specific constituent species⁵⁵ in the current alloy system. Further investigation to figure out which elements actually involve the MSROs and how MSROs are mechanically stabilised with a higher $\dot{\epsilon}$, for example, at different low temperatures and levels of applied plastic strain, would be interesting.”

Also, see the page #21,

“A higher $\dot{\epsilon}$ -induced hardening, after the onset of yielding, arises from high degree/extent of MSRO, possibly promoted by lattice distortion effect^{28,34,35,56} in the large-strain regime.”

Comment #2

If the MSRO is unique in FeMnCrCo, then why is it unique? What is the precondition to having MSRO during loading?

Author reply to comment #2

Many thanks for this valuable comment. According to the last author-reply-to-comment #1, we deleted “unique” term in this revised manuscript. Based on our findings, we assume that there are four preconditions for showing the MSRO in the MPEAs during loading: (i) the material has low SFE; (ii) tensile testing is conducted at liquid-N₂ temperature, which lowers the SFE of most materials; (iii) the material majorly deforms in terms of slip bands; (iv) the material contains the CSRO before straining. In addition, please kindly consider the author-reply-to-comment #1.

The following changes were made:

Please see lines between 125 and 128 on page #6,

“In this context, the experimental results and computational predictions provide insights into the microstructural features attributable to the MSRO in the non-

equiatomic FeMnCrCo HEA and other fcc non-equiatomic NiCrCo MEAs (for universality), particularly in the alloys with low SFEs and under low-SFE conditions (i.e., loading at 77 K).”.

Comment #3

It has been proposed that CSRO can be used to tailor the alloy properties in the fabrication process. On the contrary, it seems that MSRO is caused during the plastic loading. Then how to make use of the MSRO?

Author reply to comment #3

We thank the reviewer for the kind support and valuable comments. In the revised version, we addressed the importance of MSRO.

The following changes were made:

In abstract on page #2,

“Underpinned by molecular dynamics, we show that MSRO in the alloys with low stacking-fault energies tends to form upon load at 77 K, which is driven by crystalline lattice defects (dislocations and stacking faults), offering new perspectives on the strain-induced ordering transition.”

Please see pages #21~22,

“Thus, the current study can provide important insights into the fundamental, practical, and mechanistic understanding of many concentrated solid solutions that are designed to take advantage of the disorder–order transition during deformation at low temperatures.”

Comment #4

A detail about the CSRO. In Fig. 1, the authors plotted some purple circles to indicate

the CSRO. Then what is the definition of the CSRO here? According to the Warren-Cowley short-range order parameter, CSRO is an averaged property of a homogeneous system, not the local clustering as indicated by the circles. Do you think there is no CSRO outside the circles?

Author reply to comment #4

Many thanks for this helpful comment, we fully agree. In principle, Warren-Cowley short-range order (SRO) parameter can be used to describe the occupation in any neighboring shell around a central atom. This parameter reflects the site occupancy for the n^{th} shell of neighbors in a binary AB alloy, defined as $\alpha_n = 1 - P_n^{\text{BA}} / x_A$, where P_n^{BA} is the probability of finding an A atom in the neighborhood of a B atom, and x_A and x_B are the proportions of atoms A and B in the alloy, with $x_A + x_B = 1$. So far solid solutions are best expressed by the statistical SRO parameter owing to the presence of disorders. Although the heterogeneous distribution of solutes is well favored in the binary AB alloys, there has been only one model for SRO in HEA. It is the cluster-plus-glue-atom model, in which the cluster is the nearest-neighbor polyhedron centered by a solute atom having strong interaction with the base solvent atoms to represent the strongest CSRO, and some other solute atoms (i.e., glue atoms) with weak interactions are certainly required to fill the space between the clusters to balance atomic-packing density [Hong, H. L., Wang, Q., Dong, C. & Liaw, P. K. Understanding the Cu-Zn brass alloys using a short-range-order cluster model: significance of specific compositions of industrial alloys. *Sci. Rep.* **4**, 7065 (2014); Ma, Y. *et al.* Chemical short-range orders and the induced structural transition in high-entropy alloys. *Scr. Mater.* **144**, 64-68 (2018)]. In addition, there are four models of SRO in the literature [Schönfeld, B. Local atomic arrangements in binary alloys. *Prog. Mater. Sci.* **44**, 435–543 (1999); Song, Z. Z. *et al.* Room-temperature-deformation-induced chemical short-range ordering in a supersaturated ultrafine-grained Al-Zn alloy. *Scr. Mater.* **210**, 114423 (2022)]:

- (i) The statistical Warren-Cowley model. SRO seems to be homogeneous.
- (ii) The disperse order model. Unlike the statistical model, SRO is heterogeneous.
- (iii) The microdomain model. In this case, well-ordered regions (the microdomains) embed in a random matrix with the same composition.

- (iv) The lattice defect model. Lattice defects such as dislocations and vacancies cause the variation of the degree of order and inhomogeneity of composition.

Considering the diverse CSRO definitions, we deleted the plotted purple circles for indicating the CSRO in our MC simulations (Figure 1).

The following changes in Figure 1 were made:

Figure 1. **Atomistic MC simulations for CSRO domains in fcc-structured Fe₄₀Mn₄₀Co₁₀Cr₁₀ (at%) HEA system.** **a**, Near-random distribution of four principal elements obtained after the MC simulations at 1373 K. A cell with 216000 atoms (15.7 nm × 13.6 nm × 12.8 nm) was used for the simulation. **b, c**, Non-random distribution of principal elements showing thermally activated Cr-enriched like pairs and Fe–Co unlike pairs at 873 and 750 K, respectively, based on the MC simulations. The captured slices of configurations with a thickness of ~3 nm are shown for clear visualisation. **d, e, f**, Corresponding radial distribution function g(r) profiles from the MC simulations, revealing that when lowering the temperatures, both Cr–Cr like pairs (red curves) and Fe–Co unlike pairs (blue curves) are strongly favoured at the first nearest-neighbour distances for producing CSROs in the HEA before straining.

Please also see Discussion section, lines 411 ~ 414 on page #18,

“The CSRO formation may obey the lattice defect model of SRO, rather than statistical Warren-Cowley SRO description that suits specifically for a binary AB alloy^{22,54}.”

Also, see lines 425 ~ 427 on page #18,

“A further in-depth energy computation or SRO cluster-plus-glue-atom model²², suited for MPEAs rather than for binary AB alloys, is needed to theoretically clarify this point in future.”

Please see the revised reference list,

12. Oh, H. S. et al. Engineering atomic-level complexity in high-entropy and complex concentrated alloys. *Nat. Commun.* **10**, 2090 (2019).
22. Hong, H. L., Wang, Q., Dong, C. & Liaw, P. K. Understanding the Cu-Zn brass alloys using a short-range-order cluster model: significance of specific compositions of industrial alloys. *Sci. Rep.* **4**, 7065 (2014).
54. Schönfeld, B. Local atomic arrangements in binary alloys. *Prog. Mater. Sci.* **44**, 435–543 (1999).
55. Song, Z. Z. et al. Room-temperature-deformation-induced chemical short-range ordering in a supersaturated ultrafine-grained Al-Zn alloy. *Scr. Mater.* **210**, 114423 (2022).

Reviewer 3:

The authors did a fantastic job with the revisions. The revised manuscript looks excellent and I have no further comments. I now recommend the paper be accepted and published!

Author reply

We would like to express our sincere appreciation to the reviewer for the clear recommendation for the publication of our work. Once again, thank you so much for your contribution and time.

REVIEWER COMMENTS

Reviewer #2 (Remarks to the Author):

I would like to thank the authors for their great effort in improving the draft. The refs they provided are also valuable.

However, I remain confused about their definition of the MSRO.

I propose a way of clarifying. In Fig. 1, you compared the configurations with and without CSROs. This shows very clear to me what is CSRO. And it is consistent with my understanding based on the WC-SRO definition.

For the MSRO, could you please also show similar comparisons, i.e., the real-space atomistic structures with and without MSRO?

In the ref you mentioned,

<https://www.sciencedirect.com/science/article/pii/S135964622100703X>

they also talked about the mechanically driven SRO. But I can understand their results - the L1₀ ordering forms when the alloy is plastically deformed.

But for your definition of MSRO, I am sorry I remain confused.

AUTHORS' POINT-BY-POINT RESPONSE TO THE REVIEWERS' COMMENTS
Manuscript NCOMMS-22-06614B

Dear Reviewers

Here, we reply to the reviewers' comments on our manuscript (NCOMMS-22-06614B) entitled *Mechanically derived short-range order and its impact on the multi-principal-element alloys*. All the amendments are outlined below in more detail and highlighted in the revised version of our paper with **blue colour highlighting**.

With best regards on behalf of the author team,

Prof. Jae Bok Seol & Prof. Hyung Seop Kim (corresponding authors)

Reviewer 2:

I would like to thank the authors for their great effort in improving the draft. The refs they provided are also valuable. However, I remain confused about their definition of the MSRO. I propose a way of clarifying. In Fig. 1, you compared the configurations with and without CSROs. This shows very clear to me what is CSRO. And it is consistent with my understanding based on the WC-SRO definition.

Comment #1

For the MSRO, could you please also show similar comparisons, i.e., the real-space atomistic structures with and without MSRO?

Author reply to comment #1

We appreciate very much for the valuable comment. Following the reviewer's suggestion, we compared the configurations with and without the mechanical loading (obtained by Molecular Dynamics simulations). In fact, the result exhibits only marginal differences in terms of the

chemical distribution of principal elements before and after straining. This is not surprising, since the MD simulation is only performed for a very short period of time during which atomic diffusion cannot occur (the chemical ordering can be dealt by the prior Monte Carlo (MC) simulations overcoming the time-scale limitation). As already explained in the previous manuscript version, the strain-induced MSRO was observed during the deformation at very low temperature (77 K) and comparably higher strain rates which is not sufficient for redistribution of principal elements. This situation corresponds well to the usual condition of MD simulation.

The following changes in Supplementary Figure 15 were made:

Supplementary Figure 15. **Atomic configurations of fcc-structured Fe₄₀Mn₄₀Co₁₀Cr₁₀ (at%) HEA system before and after straining.** A cell with 216000 atoms (15.7 nm × 13.6 nm × 12.8 nm) prepared by the atomistic MC simulations at 750 K, shown in Fig. 1c, was used for straining at 77 K. **a, b**, Non-random distribution of four principal elements before and after straining, respectively. The atomistic distribution in **b** is obtained at moderate strain rate. All the captured slices of configurations with a thickness of ~2 nm are shown for clear visualisation. **c, e**, Corresponding radial distribution function g(r) profiles. **d, f**, Corresponding cell structures visualized by using the polyhedral template matching method for MD simulations (see Methods).

Please also see Discussion section, lines 415 ~ 418 on page #18,

“For better understanding about the MSRO, we showed the comparison of real-space atomistic structures with and without MSRO (Supplementary Fig. 15). Specifically, the $\text{Fe}_{40}\text{Mn}_{40}\text{Cr}_{10}\text{Co}_{10}$ structure with CSRO (Fig. 1c) was considered as the initial state (without MSRO).”

Comment #2

In the ref you mentioned,

<https://www.sciencedirect.com/science/article/pii/S135964622100703X>

they also talked about the mechanically driven SRO. But I can understand their results - the $L1_0$ ordering forms when the alloy is plastically deformed. But for your definition of MSRO, I am sorry I remain confused.

Author reply to comment #2

Many thanks for this additional comment. We fully agree that the mechanically driven SRO has been found by the recent studies. The previous work by Song *et al.* demonstrated the $L1_0$ ordering promoted by room-temperature-deformation. They also argued that vacancies introduced by severe plastic deformation play an important role in the formation of the CSRO. However, our study stressed that stacking faults and edge dislocations produced by applying relatively higher loading rate at much lower temperature (77 K) are vital for the formation of mechanically derived short-range order (MSRO). Again, the MSRO described in this study is a chemical short-range order that is derived by mechanical deformation at 77 K, while the previous chemical SROs reported in MPEA fields are mostly driven by specific heat treatments. Hence, we expect that the orders observed in this study are chemical ones, even the ordered nanophases in various crystalline alloys are also mainly involving chemical orders. Finally, we cordially hope this definition would be helpful for your understanding.

The following change was made:

Please see Discussion section, lines 413 ~ 415 on page #18,

“Following the previous study⁵⁵, we also define the MSRO described in this study as CSRO that can form under mechanically loading conditions, particularly at cryogenic temperatures without external and internal heating effects.”

REVIEWER COMMENTS

Reviewer #1 (Remarks to the Author):

I agree with reviewer #2 that the comparisons in the real-space atomistic structures with and without MSRO are essential, as the authors done in Fig. 1 for CSRO. Therefore, I suggest the authors to plot atomistic structures of the domains with MSRO (green dotted circle in Fig. 4e) and without MSRO in a straightforward way.

AUTHORS' POINT-BY-POINT RESPONSE TO THE REVIEWERS' COMMENTS
Manuscript NCOMMS-22-06614C

Dear Reviewers

Here, we reply to the reviewers' comments on our manuscript (NCOMMS-22-06614C) entitled *Mechanically derived short-range order and its impact on the multi-principal-element alloys*. All the amendments are outlined below in more detail and highlighted in the revised version of our paper with **blue colour highlighting**.

With best regards on behalf of the author team,

Prof. Jae Bok Seol & Prof. Hyong Seop Kim (corresponding authors)

Reviewer 1 (Remarks to the Author):

I agree with reviewer #2 that the comparisons in the real-space atomistic structures with and without MSRO are essential, as the authors done in Fig. 1 for CSRO. Therefore, I suggest the authors to plot atomistic structures of the domains with MSRO (green dotted circle in Fig. 4e) and without MSRO in a straightforward way.

Author reply to comment #1

We appreciate very much for the valuable comment. Following the reviewer's suggestion, we again compared the atomic configurations with and without the MSRO. In this version, we revised Supplementary Fig. 15, added Supplementary Fig. 16, and Supplementary Discussion 4 in order to explain the atomic configurations with and without the MSRO in a straightforward way.

Based on our high-resolution and scanning TEM images obtained from the deformation structure at 77 K shown in Fig. 4c-f, the atomic configurations were considered the atomic configurations with and without the MSRO via atomistic Monte Carlo (MC) simulations and

Molecular Dynamics (MD) simulations. Unfortunately, according to the MC simulation, non-random distribution of four principal elements (specifically, Cr-enriched and Fe-Co-enriched features) in the fcc HEA structure without deformation is much similarly to that of the fcc HEA structure with deformation, as shown in Supplementary Fig. 15a and d, respectively. This is owing to a paucity of diffusion for solutes or vacancies during mechanical loading at 77 K, implying the MSRO is mainly originated from a diffusionless process. This is a prime cause for difference in comparison with the CSRO (i.e., CSRO generally derives from the diffusion-mediated redistribution of principal elements). Hence, we assume that despite mechanical deformation loading at 77 K, atomic configurations of MSRO are almost identical to those of CSRO.

According to our TEM results that showed both the MSRO-induced $\frac{1}{2}\{311\}$ diffuse scattering in electron reciprocal-space EDPs under the $[112]$ zone axis. (Fig. 3d, e, and Fig. 4d) and the MSRO atomic structure (Fig. 4e), the MSRO was formed by the redistribution of principal elements, which derived from the mechanical stacking faults inside slip bands (Fig. 4g). This implies that like the CSRO in equiatomic MEAs^{17,18}, the measured interplanar spacing (d_{MSRO}) of the MSRO $\{311\}$ planes (green dotted lines) in the current non-equiatomic HEA doubles the interplanar spacing (d_{fcc}) of the $\{311\}_{\text{fcc}}$ planes (white lines). The doubled d_{fcc} by the MSRO accounts for the origin of extra diffuse scattering at $\frac{1}{2}\{311\}$ locations in the electron reciprocal space. This TEM observation is intriguingly supported by the MD simulations (after MC simulation at 750 K) and corresponding MD-EDPs (see Supplementary Fig. 15b, c, e, and f, respectively). Although the MC simulations in this study are failed to decipher the difference in real-space atomistic structures without and with the MSRO, our MD simulations reveal clear differences in the cell structures before and after deformation at 77 K. The MD cell structure of fcc non-equiatomic HEA before deformation, *i.e.*, without MSRO, exhibit only normal fcc spots in the MD-EDPs (Supplementary Fig. 15b and c).

Meanwhile, after deformation, the MD cell structure with MSRO exhibits profuse mechanical stacking faults (red colour), shown in Supplementary Fig. 15e. Thus, in the corresponding MD-EDPs, we found not only clear streaking along fcc slip planes $\{111\}$ under the $[110]$ zone axis but also clear extra spot-like $\frac{1}{2}\{311\}$ diffuse scattering under the $[112]$ zone axis. These simulation results agree well with our TEM-EDPs of the current FeMnCrCo HEA with the MSRO. More interestingly, the TEM- and MD-EDPs described in this study are interestingly

much similarly to the previously reported TEM results of CSRO structure in VCoNi and CrCoNi MEAs. Based on this similar location of the $\frac{1}{2}\{311\}$ diffuse scattering by MSRO, it is highly plausible that real-space atomistic structure with the MSRO in the FeMnCrCo HEA structure is identical to that of CSRO in VCoNi and CrCoNi MEAs. We note that the CSRO is constructed by the L1₁-type structure motif in the fcc MEA structures [5]. In turn, the MSRO unit cell is likely to have L1₁-type structure motif, and the atomic occupation in MSRO during mechanical loading at 77 K is re-arranged as a result of diffusionless process, *i.e.*, stacking faults and edge dislocations inside slip band. Hence, according to the concept of CSRO structure motif, we herein provide a schematic of the MSRO atomistic structure motif (Supplementary Fig. 16). Based on the TEM- and MD-EDPs along different zone axes, we sketch the diffraction patterns caused by normal fcc spots and extra MSRO scattering (Supplementary Fig. 16a). These EDPs described here agree well with those by fcc plus CSRO shown in the literature. Next, we constructed atomic projections of MSRO along different zone axes. Based on the TEM observations (Fig. 4e), where the measured interplanar spacing (d_{MSRO}) of the MSRO $\{311\}$ planes (green dotted lines) in the current non-equiatom HEA doubled the interplanar spacing (d_{fcc}) of the $\{311\}_{\text{fcc}}$ planes (white lines), like the CSRO in equiatom MEAs^{17,18}, the unit cells of MSRO plus fcc structure were sketched. Supplementary Fig. 16b show the schematic of atomic structure motif exhibiting MSRO (green circles), viewed from the $\langle 1\ 1\ 2 \rangle$ directions. Based on this motif, we accordingly present simplified three-dimensional (3D) unit cell of MSRO in the fcc crystalline lattice along the $\langle 1\ 1\ 2 \rangle$ zone axis in Supplementary Fig. 16c. Furthermore, the possible 2D unit cell of MSRO on the $\{111\}_{\text{fcc}}$ slip planes under $\langle 112 \rangle_{\text{fcc}}$ directions (Supplementary Fig. 16d) and on the $(112)_{\text{fcc}}$ plane under $\langle 111 \rangle_{\text{fcc}}$ directions are schematically illustrated in Supplementary Fig. 16e.

The following changes in revised manuscript were made:

On lines 271~276, page 12,

“This implies that like the CSRO in equiatom MEAs^{17,18}, the measured interplanar spacing (d_{MSRO}) of the MSRO $\{311\}$ planes (green dotted lines) in the current non-equiatom HEA doubles the interplanar spacing (d_{fcc}) of the $\{311\}_{\text{fcc}}$ planes (white lines). The doubled d_{fcc} by the MSRO accounts for the origin of extra diffuse scattering at $\frac{1}{2}\{311\}$ locations in the electron reciprocal space.”

On lines 417~428, page 18,

“Following the previous study⁵⁵ we also define the MSRO described in this study as ordering at short ranges: namely, it can form as a result of the diffusionless redistribution of principal elements under mechanically loading conditions, particularly at cryogenic temperatures without external and internal heating effects. This is a prime cause for difference in comparison with the CSRO. To figure out the MSRO concisely, we examined the atomistic structures with and without MSRO. Even though MC simulation of a structure with MSRO showed the similar atomic distribution as a structure without MSRO, MD simulation revealed the difference in cell structures and MD-EDPs between with and without the MSRO (Supplementary Fig. 15). Based on our MD simulation and TEM observations (Fig. 4), more details of the possible MSRO atomic structure motif and corresponding unit cell shown in Supplementary Fig. 16 are described in Supplementary Discussion 4. However, to determine which MSRO motif exists in the real-deformation structure, further TEM-energy dispersive spectroscopy characterisation^{17,18} still requires.”

The following changes in Supplementary Figure 15 were made:

Supplementary Figure 15. Atomic configurations, cell structures, and corresponding MD-EPDs under different zone axes of fcc-structured $\text{Fe}_{40}\text{Mn}_{40}\text{Co}_{10}\text{Cr}_{10}$ (at%) HEA system before and after straining at 77 K. a–c, before straining (without MSRO). d–f, after straining at 77 K (with MSRO). Details are explained in Supplementary Discussion 4. **a, d**, MC simulation-provided radial distribution function $g(r)$ profiles of non-random distribution of four principal elements in the HEA without and with the MSRO, respectively. This result showed Cr-enriched and Fe-Co-enriched features, revealing that atomic configurations of MSRO are almost identical to those of CSRO. A cell with 216000 atoms ($15.7 \text{ nm} \times 13.6 \text{ nm} \times 12.8 \text{ nm}$) prepared by the atomistic MC simulations at 750 K, shown in Fig. 1c, was used for straining at 77 K. The HEA structure including the CSRO (Fig. 1c) was considered as the initial state (without MSRO). **b, e**, Corresponding MD simulation-provided cell structures of the HEA without and with the MSRO, respectively, visualized by using the polyhedral template matching method for MD simulations (see Methods). **c, f**, Corresponding MD-EDPs, viewed from different zone axes, of the HEA without and with the MSRO, respectively. Owing to the presence of MSRO in the MD cell structure, we observed not only streaking in $\{111\}$ slip planes under the $[110]$ zone axis but also clear extra spot-like $\frac{1}{2}\{311\}$ diffuse scattering under the $[112]$ zone axis in **f**.

The following changes in Supplementary Figure 16 were made:

Supplementary Figure 16. Schematic illustration of diffraction patterns and possible atomic configurations with MSRO in fcc-structured Fe₄₀Mn₄₀Co₁₀Cr₁₀ (at%) HEA system after straining at 77 K. **a**, Sketch of normal fcc reflections (gray circles) and MSRO scattering (green color), revealed by our TEM- and MD-EDPs under different zone axes. These EDPs described here agreed well with those in the fcc plus CSRO shown in the literature^{16-19,34,39}. **b**, Atomic structure motif exhibiting MSRO (green circles) in the fcc structure, viewed from the $\langle 1\ 1\ 2 \rangle$ directions. Details are explained in Supplementary Discussion 4. **c**, Possible 3D unit cells of the MSRO along the $\langle 1\ 1\ 2 \rangle$ zone axis. **d, e**, Corresponding possible 2D unit cell of the MSRO on the $\{111\}_{\text{fcc}}$ slip planes under $\langle 112 \rangle_{\text{fcc}}$ directions, and on the $(112)_{\text{fcc}}$ plane under $\langle 111 \rangle_{\text{fcc}}$ directions, respectively. Based on the TEM observations (Fig. 4e), where the measured interplanar spacing (d_{MSRO}) of the MSRO $\{311\}$ planes (green dotted lines) in the current non-equiatomic HEA doubled the interplanar spacing (d_{fcc}) of the $\{311\}_{\text{fcc}}$ planes (white lines), like the CSRO in equiatomic MEAs^{17,18}, the unit cells were sketched.

REVIEWERS' COMMENTS

Reviewer #1 (Remarks to the Author):

My comments have been addressed properly. Therefore, I would recommend it for publication.